



# Technical Note: Uncovering the influence of methodological variations on the extractability of iron bound organic carbon.

Ben J. Fisher[1*], Johan C. Faust[1], Oliver W. Moore[1], Caroline L. Peacock[1], Christian März[1]

[1]School of Earth and Environment, University of Leeds, Leeds, LS2 9JT, UK
[*]*Current address: School of GeoSciences, University of Edinburgh, EH9 3FE, UK*

*Correspondence to*: Ben J. Fisher (Ben.Fisher@ed.ac.uk)

**Abstract.**

Association of organic carbon (OC) with reactive iron ($Fe_R$) represents an important mechanism by which OC is protected against remineralisation in soils and marine sediments. Recent studies indicate that the molecular structure of organic
compounds and/or the identity of associated $Fe_R$ phases exerts a control on the ability of an OC-$Fe_R$ complex to be extracted by the citrate-bicarbonate-dithionite (CBD) method. While many variations of this method exist in the literature, these are often uncalibrated to each other, rendering comparisons of OC-$Fe_R$ values extracted by different method iterations impossible. Here, we created a synthetic ferrihdyrite sample coprecipitated with simple organic structures and subjected these to modifications of the most common CBD method. Method parameters (reagent concentration, time of the extraction and sample
preparation methods) were altered and $Fe_R$ recovery measured to determine which (if any) modifications resulted in the greatest release of $Fe_R$ from the sediment sample. We provide an assessment of the reducing capacity of Na dithionite in the CBD method and find that the concentration of dithionite deployed can limit OC-$Fe_R$ extractability for sediments with a high $Fe_R$ content. Additionally, we show that extending the length of any CBD extraction offers no benefit in removing $Fe_R$. Finally, we demonstrate that for synthetic OC-$Fe_R$ samples, the almost universal technique of freeze drying samples can significantly
reduce OC-$Fe_R$ extractability and we offer insight into how this may translate to environmental samples using Arctic Ocean sediments. These results provide a valuable perspective on how the efficiency of this extraction could be improved to provide a more accurate assessment of sediment OC-$Fe_R$ content. Accurate determinations of OC-$Fe_R$ in sediments and soils represents an important step in improving our understanding of, and ability to model, the global carbon cycle.

## 1. Introduction

Constraining parameters of biogeochemical cycles remains one of the largest challenges in the development of Earth system and climate models (Achterberg, 2014). Understanding in which environments organic carbon (OC) persists, the mechanisms which facilitate its preservation and the extent of such mechanisms is crucial for our understanding of the global carbon cycle. Marine sediments represent the largest sink for OC on Earth (Hedges and Keil, 1995), and as such the preservation of OC here is crucial in controlling atmospheric CO2 levels and maintaining an oxygenated Earth (Canfield, 1993). Preservation of OC
has been linked to different mechanistic and environmental factors, e.g. intrinsic recalcitrance of biomacromolecules, physical protection of OC by organic/inorganic matrices and redox conditions (Burdige, 2007 and references therein). Physical



protection of OC by association with reactive iron (Fe$_R$) minerals, via mono or multi-layer adsorption and/or coprecipitation, is thought to represent a significant mechanism by which OC is preserved in marine sediments, accounting for 10-20% of the preserved sediment OC pool (Lalonde et al., 2012; Salvadó et al., 2015; Faust et al., 2020; Zhao et al., 2018; Wang et al., 2019;

Ma et al., 2018). Additionally the OC-Fe$_R$ interaction is equally important in extending the residence time of OC in soils, important for water retention (Rawls et al., 2003), resilience to erosion, and overall soil fertility via nutrient bioavailability (Milne et al., 2015). The prevalence of OC-Fe$_R$ is generally greater in soils than in sediments, accounting for approximately 40% of soil total organic carbon (TOC) as per Wagai and Mayer (2007) and Zhao et al. (2016).

Extraction of reactive Fe bound OC (OC-Fe$_R$) has been conducted by various iterations of the citrate-bicarbonate-dithionite (CBD) method, originally in soils (Deb, 1950; Mehra and Jackson, 1958) before being applied to marine sediments by Lalonde et al. (2012). The OC-Fe$_R$ extraction operates on the principle that reductive dissolution of reactive Fe phases with sodium (Na) dithionite liberates Fe$_R$-bound OC from the sediment matrix. The dissolution is conducted at circumneutral pH buffered with sodium bicarbonate and trisodium citrate to prevent hydrolysis of OC (Mehra and Jackson, 1958; Lalonde et al., 2012).

Despite the longevity of this method, recent studies have identified inefficiencies. For example, Adhikari and Yang (2015) reported that only 5-44% of OC was released from hematite-humic acid complexes upon Fe dissolution. Fisher et al. (2020) also document incomplete (<60%) reduction of OC-ferrihydrite complexes by the same method and show that the molecular composition of associated OC has a large influence on Fe reactivity, with carboxyl rich compounds being most resistant towards extraction. As the extraction is operationally defined (i.e. extracts based upon susceptibility of an individual

compound/mineral to chemical treatment, not upon the identity of that compound/mineral), organic matter (OM) composition and Fe phase crystallinity both have the ability to alter the reactivity of an Fe-OC compound to a point where a compound is not extracted any more by the CDB method. These findings contrast the previous understanding of the CBD method performed in an experimental context which states that this extraction will "fully reduce all solid reactive iron phases and associated organic carbon" (Lalonde et al., 2012). So far, our developed knowledge has allowed for a cautious understanding and

interpretation of OC-Fe$_R$ values, but it has yet to be seen whether this extraction can be improved in order to provide a more accurate assessment of the extent to which OC-Fe$_R$ represents an important mechanism for OC preservation in marine sediments.

Here, we synthesised a ferrihydrite sample coprecipitated with simple organic compounds spiked into marine sediment,

mimicking a natural marine sediment matrix containing OC-Fe$_R$. This method allows for the creation of a sample with a known concentration of reducible Fe, therefore an accurate determination of ferrihydrite extraction efficiency can be obtained by subtracting end state Fe from initial concentrations while accounting for mass loss. Here we aim to maximise Fe extractability in order to increase the proportion of the OC-Fe$_R$ pool extracted by CBD.



Systematic improvements to the CBD method have not yet been attempted, most likely due to the multiple constraints associated with trying to quantitatively extract both OC and Fe. One such constraint is the pH at which the extraction is conducted; lowering the pH below circumneutral values will result in hydrolysis of OC, releasing OC not associated with $Fe_R$ from the sediment matrix. It should be noted, however, that lower pH extractions are known to be more efficient at extracting the targeted reactive Fe phases if the co-extracted organic compounds are not of interest (Thompson et al., 2019).


In previous studies using the CDB method, concentrations of Na dithionite, and the ratio of Na dithionite to sample mass in the reaction, were not uniform, and a summary of the most substantive differences is shown in Table 1. This demonstrates how the same extraction has been conducted with different "chemical strengths" which, for an operationally defined extraction, makes comparison of results from such experiments impossible. Despite these wide variations in Na dithionite concentration,

previous studies have not conducted systematic assessments of the reductive strength of dithionite for soils or sediments. Additionally, earlier studies make reference to repeating the extraction multiple times for Fe rich samples (Mehra and Jackson, 1958; Aguilera and Jackson, 1953), or to altering sample mass to account for variability in Fe contents (Wagai and Mayer, 2007). Such considerations have been lost in more recent iterations of the CDB method applied to sediments (Lalonde et al., 2012).


Our approach is targeted towards testing variations in the physical parameters of the CBD method without changing the chemical "recipe" (including pH conditions) of the reductive dissolution method. Stages of the CBD method were individually tested for different extraction times, Na dithionite concentrations and sample preparation methods.

Reaction time was investigated as some studies have extended the run time of the CBD extraction from the most common time of 15 minutes (Lalonde et al., 2012; Aguilera and Jackson, 1953; Mehra and Jackson, 1958). Wagai and Mayer (2007) performed a 16 hour extraction (substituting citrate with weak HCL acid rinses to avoid use of organic compounds), and Patzner et al. (2020) extended to 6 hours. The usefulness of extending a dithionite based extraction is questionable due to the known rapid decomposition of dithionite in aqueous form suggesting, a quick loss of reduction potential (Lister and Garvie,

1959; Lem and Wayman, 1970). Finally, sample preparation methods were examined since the commonly applied method of freeze drying is thought to cause particle aggregation, which may artificially shield Fe phases from reduction. The alternative approach of using a sediment slurry has been applied to soils (Chen et al., 2020) but has not been investigated in the context of marine sediments.

After conducting extractions under a wide range of parameter variations, we aimed to establish whether there was one clear set of optimum conditions for all CBD extractions or whether the extraction approach should be dynamic, i.e. adjusted to chemical and physical characteristics of any individual sample set.



## 2. Methods

### 2.1 Synthesis of ferrihydrite coprecipitates.

A coprecipitation of ferrihydrite with an organic carbon compound (hexanedioic acid) was conducted as described in Fisher et al. (2020). Briefly, 3 g of hexanedioic acid ($C_6H_{10}O_4$) was dissolved in 250 mL of deionised (DI) water with 20 g of Fe (III) nitrate nonahydrate [Fe(NO3)3.9H2O]. 1 M potassium hydroxide (KOH) was then added by titration to achieve a pH of $7.0 \pm 0.3$ to precipitate 2-line ferrihydrite according to the original method of Schwertmann and Cornell (2000). The resultant slurry was rinsed 5 times in 5 L of DI water until gravitationally settled, the pH was then raised to 7 through the dropwise addition

of 0.1 M NaOH, centrifuged (2750 g, 20 mins), and the precipitate retained. The precipitate was then either immediately frozen and freeze dried, or retained in slurry form and stored at 4 °C, as dictated by the experimental conditions detailed in section 2.4.

Two additional precipitates were prepared according to the method described above, but substituting hexanedioic acid with

pentanoic ($C_5H_{10}O_2$) or 1,2,4-Butanetricarboxylic acid ($C_7H_{10}O_6$) for use in the sample preparation experiment. These acids differ in their carboxyl group content (pentanoic- 1 COOH, hexanedioic- 2 COOH, 1,2,4-Butanetricarboxylic- 3 COOH), and factor thought to influence their binding to $Fe_R$ (Mikutta, 2011; Karlsson and Persson, 2012, 2010). This step was taken to investigate whether there was a relationship between the binding association and the sample state (i.e, to check if compounds were more easily removed in the slurry state because they were initially weakly bound or due to a true difference in physical

behaviour between slurry and dry samples).

### 2.2 Spiking of marine sediments.

The OC-$Fe_R$ content of synthetic samples was varied to explore whether mechanistic trends persisted at environmentally relevant concentrations. To achieve this, we mixed the precipitate with a marine sediment 'carrier' material as described by Fisher et al. (2020), using the same original carrier sample and similarly treated to liberate OC and inorganic carbon. The ratio

of ferrihydrite-organic coprecipitate relative to marine sediment was differed to create a concentration gradient, and these ratios are detailed in Table 2. Spiking was achieved by agitation of either the freeze dried coprecipitate with the sediment carrier or the dry weight equivalent of slurry samples with the sediment carrier. Dry weight of slurry samples was determined by drying 10 x 1 mL aliquots of coprecipitate slurry at 40 °C to calculate mg/ml of coprecipitate and taking the mean value.

### 2.3 Citrate-bicarbonate-dithionite reduction of $Fe_R$

Reductive dissolution of reactive Fe phases was conducted according to an established CBD protocol (Lalonde et al., 2012; Salvadó et al., 2015). A synthetic sediment sample (0.25 g, or dry weight equivalent for slurry samples) was added to 13 mL of 0.11 M sodium bicarbonate ($NaHCO_3$) and 0.27 M trisodium citrate ($Na_3C_6H_5O_7$) in a centrifuge tube then placed in a water bath at 80 °C to pre-heat. Subsequently, 0.25 g of sodium dithionite was dissolved in 2 ml of 0.11 M NaHCO3 and 0.27 M



Na$_3$C$_6$H$_5$O$_7$ solution and added to the pre-heated mixture before agitation and further heating at 80 °C for 15 mins. Despite

known degradation of sodium dithionite in aqueous solutions, no difference in Fe extractability was observed when compared to addition of dithionite in solid phase for repeat samples. Prior dissolution of Na dithionite also allowed for a more rapid and less labour-intensive addition when performing parallel sample analysis, compared to individual additions of 0.25 g dry Na dithionite as detailed in earlier methods (Lalonde et al., 2012; Salvadó et al., 2015). A control extraction was conducted alongside, replacing Na dithionite and trisodium citrate with Na chloride at an equivalent ionic strength; 13 mL of 1.6 M NaCl

and 0.11 M NaHCO$_3$, followed by 0.22 g NaCl dissolved in 2 mL of the 1.6 M NaCl and 0.11 M NaHCO$_3$ solution. Following the extraction, samples were centrifuged (3000 g, 10 mins) and the supernatant retained. A three times rinse cycle using artificial seawater was then conducted on the precipitate to remove any residual dissolved Fe, a 15 ml aliquot from each of these rinses was retained and combined per sample. All supernatants were acidified to pH <2 with 12 N HCl to prevent Fe precipitation.

**2.4 Alteration of method conditions.**

In order to investigate potential improvements to the method, it was necessary to alter individual parameters of the extraction protocol. Each parameter was changed in isolation in order to allow any subsequent change in Fe extractability to be associated with this variable. Sample preparation methods were compared by the extraction of synthetic freeze dried samples compared to untreated slurry samples as previously described. The amount of Na dithionite added to a reaction was also changed, with

amounts both lower (0.125 g) and higher (0.375 g, 0.500 g, 0.625 g) than the standard addition of 0.25 g Na dithionite per 0.25 g of dried sample performed. Where the dithionite addition was changed in the reduction reaction, an equivalent change was made for the control experiment to maintain the equivalence of ionic strengths. Finally, reaction time was increased at 3 time points beyond the usual 15 minutes reaction (30, 45 and 60 mins).

For the sample preparation and Na dithionite experiments, repeats were conducted over a concentration gradient dependent on the amount of OC-Fe$_R$ spiked into the sediment. While performing these extractions on pure synthetic OC-Fe$_R$ is useful for uncovering a mechanistic trend, dilution with OC free sediment to lower OC-Fe$_R$ contents in the sample ensured any trends uncovered are noticeable at environmentally relevant conditions (OC-Fe$_R$ <50%). Repeats of samples across this concentration gradient are in lieu of direct replicates for each unique sample condition. These were not possible due to yield limitations

imposed by ferrihydrite coprecipitate synthesis (net ~5g per 5 L rinse solution). It was essential that all samples within any one experiment originated from the same batch of ferrihydrite as OC adsorption and Fe content are not consistent across batches. Outliers can still be identified by comparison to the trends present in replicates at differing concentrations of OC-Fe$_R$.

**2.5 Environmental sample treatment.**

To allow comparison between sample preparation methods applied to samples containing synthetic OC-Fe$_R$, natural samples

were evaluated and subjected to the same methods of CBD extraction and Fe elemental analysis. Arctic Ocean seafloor



sediment was collected (Cruise: JR17007, Latitude (N): 80.1167, Longitude (E): 30.06827, water depth 283 m, station B16, sediment depth 22-23 cm), of which half was freeze thawed and half was freeze dried.

### 2.6 Elemental analysis for iron.

Initial concentrations of Fe in synthetic samples were obtained by digesting ~2 mg of dried sample in 1 mL 12N HCl at room
temperature followed by a 10-fold dilution with 1% HCl solution. Further dilutions were made as necessary, dependent on Fe content, using MilliQ water to produce a subsample within the detectable window (1–10 ppm Fe). Fe concentrations for both the initial samples and supernatants from the extraction were determined by atomic absorption spectroscopy (Thermo Fisher iCE3300 AAS). Calibration was performed using matrix matched standards and quality control was confirmed following every 10 samples by repeat sampling of calibration standards to check for drift. Supernatants from control experiments were also
measured for Fe content and these were diluted 20-fold to prevent salt blockages, supernatant from seawater rinses remained undiluted except for where the Fe concentration was >10 ppm, whereby these were diluted 10-fold. Extraction of Fe was calculated by subtracting the amount of Fe lost in the control experiment from Fe lost following extraction, then subtracting this from the initial Fe of each sample.

### 2.7 Elemental analysis for carbon.

Carbon contents of synthetic samples before and after extraction were measured using a LECO-SC144DR C&S analyser. Carbon content was not measured for all samples, but was used during the experiment where Na dithionite concentrations varied (see section 3.1). This was performed to ensure that at the end point samples with incomplete Fe recovery also experienced incomplete OC recovery, as expected due to the <1 OC:Fe molar ratio of our coprecipitates. This measurement can therefore be used to confirm the choice of %Fe loss as a proxy for %OC-Fe$_R$ recovery across the entire dithionite
concentration gradient. The LECO analyser was calibrated with, and quality control checked against, a known standard (LECO 502-694). All carbon samples were analysed in an oven dried state (40 °C, 12 hours). Carbon loss was calculated according to Supplementary equation 1 of Fisher et al. (2020), adapted from Peter and Sobek (2018) and Salvadó et al. (2015), to correct for mass loss during the dissolution. Instrument error for the LECO analyser was low (≤1% RSD) due to drift calibration throughout the analytical run.

## 3. Results

### 3.1 Addition of varying quantities of Na dithionite

While the concentration of Na dithionite in the CDB extraction was varied considerably over method iterations, the contemporary method of Lalonde et al. (2012) requires a 0.25 g addition relative to 0.25 g of dried sediment sample. Here, the mass of dithionite added to our reaction was adjusted (0.125 g, 0.375 g, 0.500 g, 0.625 g) while the sediment mass remained
at 0.25 g. The %Fe extracted from these adjusted extractions is shown in Fig. 1, and this figure can be interpreted as a

visualisation of the reduction capacity of Na dithionite relative to initial Fe content. All samples show incomplete reduction of Fe regardless of Na dithionite addition, with those samples containing the least Fe proving extractable for the greatest proportion of Fe. The 30 and 40% OC-Fe$_R$ containing samples tracked almost identical paths for their extractable %Fe, while 20% OC-Fe$_R$ is more readily extracted and 50% is the least extractable.


For the sediment sample containing 20% OC-Fe$_R$, maximal Fe extraction occurs at the baseline 0.25 g dithionite addition (89%), while for sediments with a greater initial %OC-Fe$_R$ content, maximal Fe extraction occurs past the baseline. For the 30 and 40% OC-Fe mix, maximal Fe extraction occurs at 0.5 g Na dithionite addition where ~88% of Fe is extracted. At 50% OC-Fe$_R$, 60% of total Fe is extracted at both 0.5 g and 0.625 g Na dithionite additions. From this, we can deduce the maximal

%Fe in sediment extractable by 0.25 g Na dithionite lies between a 20 and 30% OC-Fe$_R$ mix, equivalent to 7-10 wt% Fe content in the sediment.

OC-Fe$_R$ extracted was measured for the point at which maximal Fe extraction was achieved by addition of excess Na dithionite. OC-Fe$_R$ values are shown in blue and indicate OC-Fe$_R$ extraction is incomplete (<100%) across all concentrations. Extraction

of OC-Fe$_R$ was roughly similar to Fe, variable within 10%.

### 3.2 The effect of sample preparation methods on Fe extractability

Two forms of each synthetic sample were prepared, one freeze dried and one as a slurry (referred to as 'wet'). Recovery of Fe following extraction is shown as %Fe extracted in Fig. 2. A total of 3 different coprecipitates (with varying C content) at 5 different OC-Fe:sediment ratios are shown with solid colours representing the freeze dried form and patterns representing the

slurry samples. Overall, a greater proportion of Fe is extracted from the slurry samples than from freeze dried samples for all coprecipitates at all concentrations. Dry samples achieve a maximum Fe extractability of 71% (3 COOH, 60%), while in slurries up to 87% is recovered (3 COOH, 100%). No 100% recovery of added Fe was achieved in any of the experiments.

Typically, Fe recovery from samples increases with the number of carboxyl groups in the coprecipitate for both dried and wet samples; this trend is clearly shown at 100, 60 and 40% concentrations of initial OC-Fe$_R$ content (shown by different shapes). The 1 COOH sample at the 80% concentration is an outlier to this trend, proving extractable for a greater mean amount of Fe compared to the 2 COOH precipitate for both the dried and slurried sample. Additionally no trend in the number of carboxyl groups is present for the 20% OC-Fe concentration, however, Fe concentrations at this level are comparatively low which may

obscure trends within this data series.

Following the experiments on synthetic OC-Fe$_R$ compounds, a similar investigation was conducted on environmental samples to observe whether the trend observed for synthetic samples could be replicated. This experiment only differed from the





previous one in comparing freeze dried vs freeze thawed (not slurry) samples. Freeze dried samples were extractable for
22.34% (± 4.05 (1 S.D)) Fe compared to 22.68% (± 6.67) for freeze thawed samples. There was no notable difference in the
amount of Fe extracted for environmental samples which had been freeze dried compared to those which were freeze thawed.

### 3.3 Variability in exposure time of a sample to CBD treatment.

Following the method of Lalonde et al. (2012) the CBD extraction is performed over a period of 15 minutes. Here we examined
whether extending this time period would increase the amount of Fe extracted. All other parameters of the extraction remained
the same, a 2 COOH coprecipitate at a 60% concentration relative to labile sediment was used in freeze dried and wet forms.
Times were advanced in 15 minute increments from 15 minutes to 60 minutes, and results from the subsequent extractions are
shown in Fig. 3. The percentage of Fe extracted remains consistent across the time series for both wet and dried samples, and
there is no evidence that increasing the extraction duration systematically increases Fe liberation.

### 4.0 Discussion

Chemical extraction of OC-Fe$_R$ from sediment samples remains an important and widely used method for determining the fate
of marine organic compounds. Like all chemical extraction techniques, the method used here is operationally limited in its
ability to remove OC-Fe$_R$, with evidence for incomplete extraction of some OC-Fe$_R$ complexes (Adhikari and Yang, 2015;
Fisher et al., 2020). Despite these apparent method inefficiencies, it is difficult to propose any substantive changes to the CBD
method due to the neutral pH constraints required to prevent OC hydrolysis. Therefore, we targeted physical, as opposed to
chemical, aspects of the CBD method, with an aim to increase Fe liberation from synthetic marine sediments containing OC-
Fe$_R$ complexes. Out of the varied experimental parameters, increasing the amount of Na dithionite added to the reaction was
most successful in increasing Fe liberation for samples containing >20% OC-Fe$_R$ (~7 wt% Fe). The process of sample
preparation was found to have a significant effect on Fe loss, with non-freeze dried samples proving extractable for a much
greater proportion of Fe compared to freeze-dried samples, likely due to particle aggregation. However, we were unable to
replicate this phenomenon in natural sediment samples, potentially due to freeze-thaw induced aggregation of the non-freeze
dried samples. Finally, an increase in reaction time was found to have no effect on increasing Fe extractability. Here we
consider the implications of these findings and discuss the practicality of applying these changes to the CBD method.

### 4.1 Concentration of Na dithionite as a primary control on OC-Fe$_R$ extraction.

Sodium dithionite as a reducing agent, buffered by bicarbonate and citrate, has been used to extract reactive Fe phases from a
range of media. One of the most important variables across these methods is the ratio of Na dithionite relative to the sample
size and its iron oxide (Fe$_2$O$_3$) content. Here, we altered the method of Lalonde et al. (2012) by changing the mass of Na
dithionite added to the CBD extraction for four synthetic sediments, each differing in Fe content, to determine whether an
increased concentration of Na dithionite would liberate more Fe than the standard method.



For the four synthetic samples we subjected to dithionite reduction, these differed in composition (7-24 wt% Fe, 20-50% initial OC-$Fe_R$ content). The concentration of Fe in these samples results in an effective dithionite to (wt) Fe reduction reaction ratio of 1:0.07-0.24. This is multiple times stronger than the concentration of dithionite previously used in incomplete Fe extractions. For example, Adhikari and Yang (2015) report <50 wt% Fe was extracted with a dithionite to Fe ratio of 1:0.8 for a humic-hematite complex. While reactive Fe content in bulk natural sediments is usually below 7 wt% Fe (Raiswell and Canfield,

1998; Canfield, 1989), inherent heterogeneity and clustering of Fe can be seen in the Iron $L_3$ edge XAS spectra of sediments (Barber et al., 2017). This has the potential to drive wt% Fe higher in small samples of sediment such as those treated by the method (0.25 g). Additionally, OC-Fe has been observed at concentrations exceeding 40% in terrestrial environments (Zhao et al., 2016; Patzner et al., 2020) and 50% in sandy beach sediments of subterranean estuaries (e.g. 56.31% ± 5.56 Martinique Beach, Canada (Sirois et al., 2018)), explaining the choice to include samples with high OC-Fe compositions in the matrix.

We find that the sample containing 20% OC-Fe (~7 wt% Fe) is maximally extracted for its reactive Fe component under the 0.25 g (0.1 M) treatment as described by Lalonde et al. (2012) (Fig. 1). Maximal extraction here is defined as the point from which further addition of Na dithionite does not increase the extraction of Fe beyond the amount of Fe extracted under the previous dithionite addition mass ± error. For example, the 20% OC-$Fe_R$ sample subject to 0.25 g dithionite is removable for 88.79% ± 3.55 of $Fe_{Total}$ while 0.375 g addition extracts 90.94% ± 3.64; as these values are within error, it can be said that

maximal extraction is achieved with 0.25 g dithionite addition per the original method of Lalonde et al. (2012).

At increased concentrations of OC-$Fe_R$, 0.25 g of Na dithionite is no longer sufficient to achieve maximal extraction. The samples with 30 and 40% OC-$Fe_R$ content, which follow almost identical trajectories, reach maximal extraction at 0.5 g/0.2 M with 88.65% ± 3.54 and 88.22% ± 3.53 of $Fe_{Total}$ recovered, respectively. These values are within the error of maximal

extraction for 20% OC-$Fe_R$ and are significantly higher than the amount of Fe liberated under the standard 0.25 g/0.1 M extraction (63.03% ± 2.52 and 67.21% ±2.69, respectively). This finding demonstrates that the OC-$Fe_R$ composition would not be correctly determined following the method of Lalonde et al. (2012) for these OC-$Fe_R$ rich sediments, and the overall extent of OC-$Fe_R$ in the marine sediment pool would be underestimated. While 30-40% OC-$Fe_R$ content is above the average for marine sediments, many samples exist in the 20-30% range. Indeed, the average value for marine sediment OC-$Fe_R$

composition given by Lalonde et al. (2012) is greater than 20% with individual marine sediments recorded as exceeding 30% OC-$Fe_R$ (e.g. Equatorial Pacific 0°N, 34.79% (Barber et al., 2017)).

The indication that Na dithionite at the 0.25 g/0.1 M addition is increasingly inefficient with increasing OC-$Fe_R$ content is confounded at the 50% OC-Fe (24 wt% Fe) composition. Here, %Fe extracted is increased from 39.96% ± 1.60 with 0.1 M

(0.25g) Na dithionite to 59.58% ± 2.38 at double strength (0.2 M). Note, however, that this differs from the previous compositions in reaching a maximum at ~60% Fe, as opposed to the ~90% achieved for 20-40% OC-$Fe_R$. Given that %Fe removed does not increase with further addition of Na dithionite (0.625 g), the amount of Na dithionite is no longer the limiting



factor in extracting Fe from OC-Fe$_R$ rich samples. It is likely that another reagent, potentially trisodium citrate, may become limiting. In the extraction reaction, citrate acts as a complexing agent to keep Fe dissolved in solution (Lalonde et al., 2012;
Sirois et al., 2018). If the increased strength dithionite treatment increases dissolved Fe beyond the complexing capacity of citrate, then excess Fe likely precipitates out of solution before measurement.

Measurement of OC-Fe$_R$ extracted for the concentration of Na dithionite at which maximum Fe is extracted showed incomplete OC-Fe$_R$ loss (Fig. 1). The similarity of OC-Fe$_R$ and raw Fe extraction values indicates that OC and Fe are reductively released
from the sediment in comparable proportions, as is expected due to the low molar OC:Fe$_R$ ratio of the coprecipitate (~0.7:1) . From these results, it is apparent that Na dithionite concentration can limit the extractability of reducible Fe and associated OC in Fe rich sediments, and that current approaches could benefit from using increased strength Na dithionite compared to the 0.1 M treatment currently used. Based on the set of experiments we conducted, an increase to 0.2 M would be sufficient. However, if increasing the amount of Na dithionite beyond its current level, other considerations need to be made, such as the
decomposition of Na dithionite in AAS standards which may skew quantifications (Taylor and Crowder, 1983). Additionally, the reduction in maximal Fe extraction seen for the 50% OC-Fe$_R$ sample, thought to be due to rapid precipitation of reduced Fe, suggests that the concentration of sodium bicarbonate and trisodium citrate may need to be changed to maintain the buffering and complexation capacity of the extraction. It is important to note that by increasing the concentration of these organic reagents, the background DOC of the experiment will also increase, which has the potential to interfere in
quantification of OC released from Fe in the reduction. It would be useful to include background DOC detection for samples (as per Patzner et al., (2020)) to avoid the accidental inclusion of organic reagents in OC-Fe$_R$ determination.

### 4.2 Freeze drying of samples as a limiting factor on Fe reduction.

Typically, chemical extractions have been performed on freeze dried samples, but how this process affects the physical properties of samples and their subsequent behaviour towards chemical reagents has not been defined. We found that the Fe
extraction efficiency from freeze dried sediment samples was much less than that measured for chemically identical samples retained in slurry form (i.e. not freeze dried) (Fig. 2). The scope of  extractions has expanded over time and this wet-chemical treatment is now performed on a diverse range of solid phases, including soils (e.g. Deb, 1950; Mackenzie, 1954; Zhao et al., 2016; Wagai and Mayer, 2007; Schulten and Leinweber, 1995), clays (Aguilera and Jackson, 1953; Deb, 1950; Mehra and Jackson, 1958; Mitchell and Mackenzie, 1954), plant roots (Taylor and Crowder, 1983), cryoconite accumulated on glaciers
(Cong et al., 2018), estuarine sediments (e.g. Jokinen et al., 2020; Zhao et al., 2018) and marine sediments (e.g. Barber et al., 2017; Lalonde et al., 2012; Salvado et al., 2015), with each material possessing unique physical and chemical characteristics. To aid the retrieval of samples from often remote locations, freeze drying has become established as an almost universal preservation/preparation method for solid samples. Removal of the aqueous phase decreases sample mass and prevents the need for frozen storage. Freeze drying inhibits processes of microbial degradation from occurring in the sediment sample,
preserving the biochemical profile. Alternative treatments such as air drying are considered to be more aggressive as they can



alter the chemical composition of samples and may inflict significant changes on sediment chemistry, including losses of biomarkers (McClymont et al., 2007).

We postulate that freeze drying-induced aggregation of sediment particles could result in reduced Fe extractability compared to non-dried samples since grain size is a known key factor in limiting determination of bioavailable Fe (Raiswell et al., 1994). Aggregation could reduce surface contact with dithionite, preventing reduction of 'shielded' sediment particles, while this could be overcome (e.g. by crushing), and this in itself would introduce further variability in grain size (Raiswell et al., 1994).The influence of freeze drying on grain particle size has been previously noted, particularly for sediment with a high clay content (>39%) (Keiser et al., 2014). McKeague and Day (1966) similarly report that finer grinding of sediment resulted

in an increased extraction of Fe. These findings indicate that particle size is a critical parameter in determining the amount of Fe extracted, however, all methods fail to define what is meant by "finely ground". This lack of definition introduces an error of reproducibility as particle size is certain to vary with different sample preparation methods and therefore two identical chemical treatments may vary in strength because of physical differences in the sediment sample.

The alternate tested method of using wet samples has largely been avoided, with only a few studies (e.g. van Bodegom et al., 2003;Chen et al., 2020) reporting the use of a wet slurry sample in soils and none for sediments. Chen et al. (2020) justify their use of slurries as being to "minimise the physical-protection mechanisms of aggregation", acknowledging that drying methods are likely to result in superficial particle protection.

Despite the benefits of conducting analyses on slurried samples in being able to extract a greater amount of OC-Fe$_R$, and therefore gain a better understanding of sediment C content, there are several considerations to be made. Firstly, determining the dry weight equivalent of a slurried sample is difficult as each subsample is inherently heterogeneous. Density tests for the tested synthetic samples indicated this contributes up to ±5% error which, while significant, is less than the difference seen between Fe extraction from slurries compared to dried samples (up to 53%, 2 COOH, 100% OC-Fe, Fig. 2). The use of fresh

'wet' samples appears to be the only method by which aggregation can be avoided with drying, freezing and thawing all producing aggregates (IAEA, 2003). However, the use of wet sediments is likely to be inappropriate for some analyses or sample sites.

Fresh sediments retain their microbiological components which can result in biological degradation of pollutants, release of

ammonia and chemical degradation via hydrolysis and oxidation (Schwab, 1980). It has also been show that freeze drying can result in elevation of DOC in sediment samples (Geffard et al., 2004), further Barbanti and Bothner (1993) report an increased amount of OC (20-44 % greater) and some metals (Zn, Cu) in the coarse fraction of freeze dried sediment compared to slurry samples, suggesting freeze drying can similarly alter sediment chemistry. Therefore, it should be acknowledged that any storage method of water-containing samples is likely to cause some level of chemical change, and samples cannot truly be





regarded as pristine. While this cannot easily be avoided, we suggest that rigorous documentation is key to making resulting data sets comparable.

Although extraction efficiency of Fe from ferrihydrite is improved through the use of wet samples, this only achieves a maximal extraction efficiency of 87% (3 COOH, 100% OC-Fe$_R$) (Fig. 2). For Fe$_R$ phases associated with less complex OC,

the extraction efficiency is even lower, e.g. for the 1 COOH sample at the same concentration (100% OC-Fe$_R$) only 30% of Fe is liberated. Trends between Fe extraction and carboxyl content have been discussed in (Fisher et al., 2020) and are mirrored here in Fig. 2, besides an inflation in 1 and 2 COOH values at the 20% concentration, likely due to errors in small numbers as a result of the dilution. The 1 COOH complex at 80% OC-Fe also appears inflated and out of step with the trends set by other concentrations with no obvious explanation. While application of the CBD method to slurried samples has the ability to

increase the proportion of Fe associated with OC extracted, this protocol may not always be practicable. Additionally, the inability to fully extract Fe even when sediments are in a slurry form indicates that other limiting factors persist which prevent complete extraction of Fe$_R$ phases by the circumneutral CBD method.

### 4.3 Rapid reduction of Fe$_R$ by Na dithionite.

One parameter of the extraction method which has remained largely consistent across all iterations of CBD treatment is an

extraction length of 15 minutes (Mehra and Jackson, 1958; Wagai and Mayer, 2007; Lalonde et al., 2012). As we observed incomplete Fe extraction (Fig. 1) for all our samples, a range of CBD extraction times were trialled to understand whether increasing the length of a reaction would increase Fe liberated, as seen for other chemical Fe extractions; oxalate, for example, is known to continue to extract Fe beyond a standard 1 hour treatment (McKeague and Day, 1966). Additionally, as previously mentioned, some iterations of the CBD method have been repeated multiple times in succession to extract the full Fe$_{CBD}$ pool,

but it is unclear whether time or reagent concentration limit full extraction of this pool on the first treatment.

Exposure time of wet and dried synthetic samples (2-COOH, 40%) to CBD was increased from the standard 15-minute treatment in 15-minute intervals to 60 minutes (Fig. 3). No difference was observed for the amount of extractable Fe across the time series, concluding that an increase in chemical exposure time has no difference on Fe extractability. This shows that

time is not a limiting factor in the CBD extraction, and that reductive dissolution of the susceptible Fe phases occurs rapidly. We would perhaps not expect any benefit from increasing the length of CBD treatment as dithionite, the reductive component, is known to undergo degradation to form sodium thiosulfate and bisulfite in aqueous solutions with a rapid second order rate constant ($K_2$) of 3.0 (g-molecule/L)$^{-1}$ min$^{-1}$ at 79.4 °C, indicating reducing conditions are unlikely to be sustained for long (Lister and Garvie, 1959).


While increasing extraction time has no benefit for extracting Fe with the purpose of determining the OC-Fe$_R$ pool, a recent adaptation of the CBD method has extended the time of the extraction in order to compensate for a reduction in the temperature

of the reaction. Patzner et al. (2020) performed a 16 hour CBD extraction at room temperature on permafrost samples to, in the first instance, determine %OC-Fe$_R$, then subsequently apply scanning electron microscopy (SEM) and nanoscale secondary

ion mass spectrometry (nanoSIMS) to analyse the extracted organominerals. Here the authors had to alter the CBD method as they were concerned that exposure of organic compounds to high temperature may alter OC structure and fate, which they wished to analyse. This raises an interesting question as to whether temperature and length of the extraction can compensate for each other to achieve the overall same %Fe extraction.

In our series of experiments, temperature was not altered as we saw no benefit to decreasing temperature, and therefore energy, of the reaction as we were focused on maximising %Fe extraction. While we saw no benefit from increasing extraction time, likely due to rapid decomposition of Na dithionite, the decomposition process may occur much slower at room temperature due to the decreased reaction energy. The authors of this pre-print (Patzner et al., 2020) have yet to calibrate their method against the standard 80 °C treatment, however, the values they obtain for %OC-Fe in permafrost soils appear agreeable, if not

a little higher than much of the literature for terrestrial samples (e.g. Zhao et al., 2016). This could potentially be a benefit to the CBD method in preserving the structural component of OC, which would subsequently allow for much wider analysis on the extracted OC, such as biomarkers, which has previously been limited by both transformation of C in extraction and by sample size. This may allow us to better understand the origins and molecular makeup of OC involved in mineral preservation processes and offers promising scope for future experimentation with the CBD method.

**5. Conclusion**

Reductive dissolution of OC-Fe$_R$ by CBD is an important and widely used method for quantifying mineral associated OC in sediments and soils, but has been shown to be misunderstood in its efficiency. Varying, often uncalibrated iterations of the method have made comparison of these extraction values impossible, compromising our ability to gain an understanding of the true extent of OC-Fe$_R$ in the global carbon cycle. In this study we aimed to address these uncertainties in the CBD method

to understand which, if any, parameters of this method could be changed to improve extraction of the targeted Fe$_R$ phases and associated OC.

We found that the mass of dithionite added to a sample appears to be limiting in extracting the total easily reducible Fe pool for Fe rich sediments and a doubling of Na dithionite addition for these sediments can increase Fe extracted from ~60% to

~90%, representing a much more complete removal of the OC-Fe$_R$ pool. We suggest that if future studies were to increase Na dithionite addition in the CBD method this should be followed by a similar increase in trisodium citrate to ensure the entire reduced Fe pool is complexed, preventing precipitation of Fe before quantification. This additional input of an organic reagent may be offset by subtraction of background DOC values from the final C content of the sediment post extraction as per the method described by Patzner et al. (2020).


Freeze drying induced aggregation appears to reduce Fe liberation in synthetic coprecipitates that were freeze dried relative to slurried, however, we were unable to replicate this increased extraction for natural samples. While we speculate this may be due to the use of freeze thawed samples, which can introduce aggregation in itself, it is hard to see a practical implementation of this adjustment for marine sediments due to the difficulty in transport of pristine samples. Consideration should also be

given to the error introduced in determining dry weight if slurried samples are to be used.

Finally, increase of reaction time (up to 1 hr, 4x standard) showed no benefit for Fe extraction Given that alteration of sample preparation methods would be impractical and extraction time offered no benefit, we suggest future work should focus on increasing the concentration of Na dithionite used in extracting a range of environmental samples known to differ in their OC-

$Fe_R$ content. However, in highlighting these methodological inefficiencies, even if they cannot be entirely offset we have established an empirical basis from which future studies can understand the likely errors associated with CBD extractions and can interpret their results accordingly.

## Author Contributions

BJF, JCF, OWM, CLP and CM designed the study, BF conducted the experimental work with support from JCF and OWM.
JCF supplied the sample material. BJF, JCF, OWM, CLP and CM all contributed towards analysis of the data sets. BJF prepared the first draft of the manuscript and all authors contributed towards review and editing of the final version.

## Competing Interests

The authors declare no competing interests.

## Data Availability

The equation used to compute OC-$Fe_R$ is available in Fisher et al., (2020), examples of implementation can be found in the associated electronic annex of that paper (https://data.mendeley.com/datasets/gpt8f8kpcs/1). Raw figure data for this manuscript is available upon request.



**Acknowledgments**

This work was supported by funds from the ChAOS project (NE/P006493/1), part of the Changing Arctic Ocean programme, jointly funded by the UKRI Natural Environment Research Council (NERC) and the German Federal Ministry of Education and Research (BMBF). Additionally, this research project has received funding from the European Research Council (ERC) under the European Union's Horizon 2020 research and innovation programme (Grant agreement No. 725613 MinOrg). We

thank Andrew Hobson and Fiona Keay of Cohen Geochemistry, University of Leeds for their assistance with sample analysis.





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





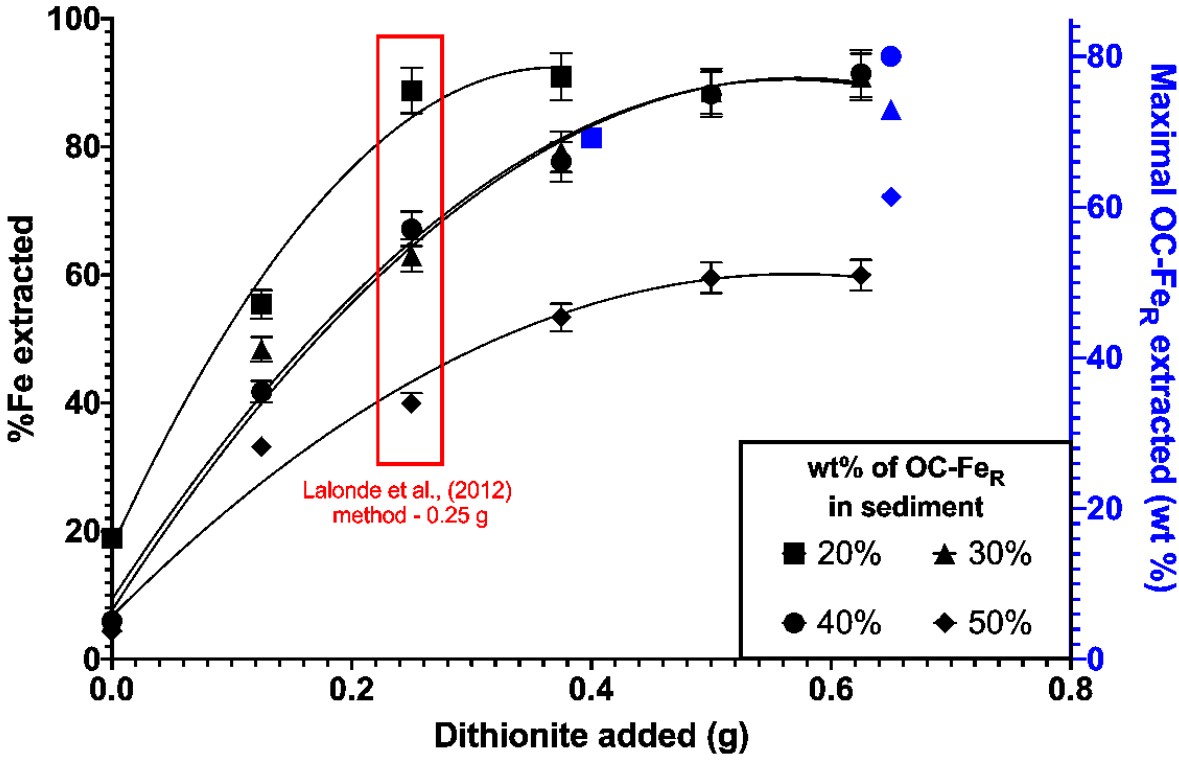

**Figure 1: Reduction capacity of Na dithionite in the extraction estimated from %Fe extracted with varying Na dithionite additions across an OC-Fe$_R$ concentration gradient. Error bars show maximal compound instrument error. Blue shapes indicate the amount**
**of OC-Fe$_R$ extracted for the concentration of Na dithionite at which Fe is maximally extracted for that sample (black).**





**Figure 2-** **Fe recovery from freeze dried vs slurry coprecipitates. Solid bars show dried samples while patterns show the wet (slurry) samples. 1/2/3 COOH refers to the number of carboxyl groups present in the coprecipitated organic acids. Each error bar shows maximum compound error.**






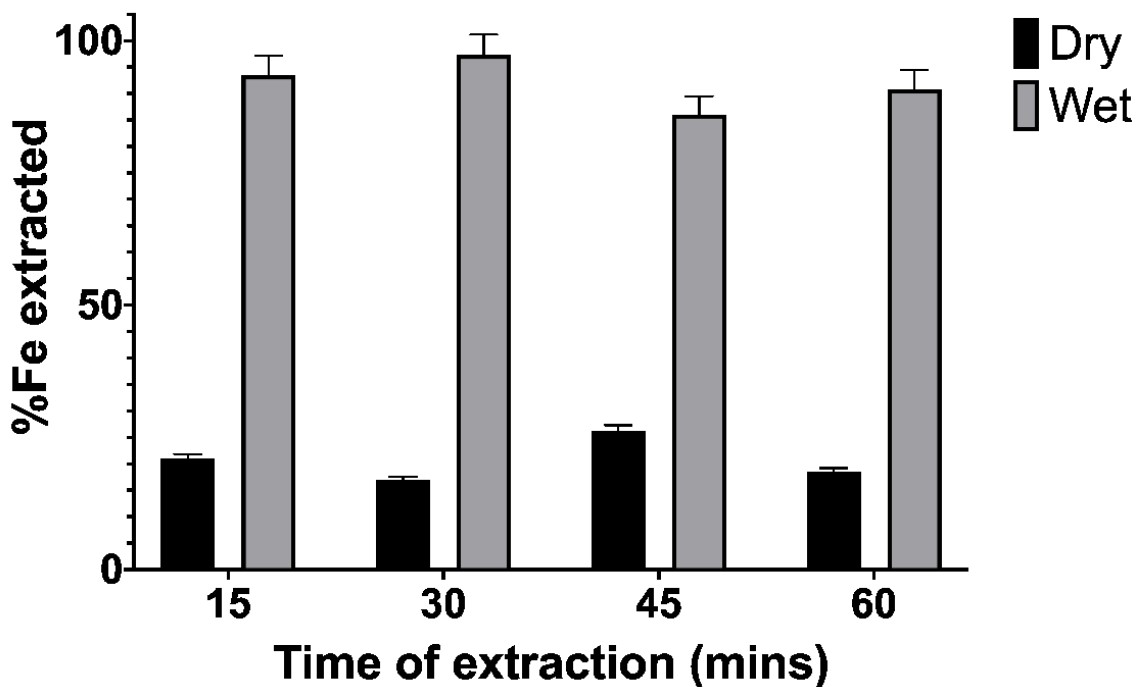

**Figure 3- %Fe extracted across a time series for CBD extraction. Error bars show compound maximal instrument error. The sample**
**used in this experiment was a synthetic sediment spiked with 2 COOH coprecipitate at 40% OC-Fe$_R$.**





| Reference | Dithionite concentration | Sample to solution addition (mg mL $^{-1}$) | Dithionite to sample mass ratio |
|---|---|---|---|
| Aguilera and Jackson (1953) | 0.144 M | 12.5[a,b] | 1:0.5[a] |
| (Mehra and Jackson, 1958) | 0.128 M | Soils: 88.89[b] | 1:4 |
| | | Clays: 22.22 | 1:1 |
| Wagai and Mayer (2007) | 0.049 M | Fe rich: 4.3 | 1:0.5 |
| | | Fe poor: 7.1 | 1:1.2 |
| Lalonde et al. (2012) | 0.1 M | Sediments (Lalonde) and | 1:1 |
| Zhao et al. (2016) | | Soils (Zhao): 16.67 | |

Table 1: Comparison of dithionite strength to sample mass in iterations of the CBD method applied to soils and sediments.

[a] Sample size in this method is variable due to variable $Fe_2O_3$ contents; samples should not exceed 0.5 g $Fe_2O_3$ so may be a 10 g sample with 5% $Fe_2O_3$ content or a 1 g sample with 50% $Fe_2O_3$ content. The ratio given is calculated on the basis of a 0.5 g $Fe_2O_3$ sample, so represents a minimal rather than absolute ratio. Dithionite concentration is based on a 40 ml reaction, while Aguilera and Jackson (1953) refer to the addition of a dithionite solution without reporting the exact volume.

[b] If any sample exceeds 5% $Fe_2O_3$, the extraction should be repeated an additional 1-2 times.

| %OC-Fe:sediment | 20 | 30 | 40 | 50 | 60 | 80 | 100 |
|---|---|---|---|---|---|---|---|
| OC-$Fe_R$ Coprecipitate (mg) | 50 | 75 | 100 | 125 | 150 | 200 | 250 |
| Sediment (mg) | 200 | 175 | 150 | 125 | 100 | 50 | 0 |

Table 2: Concentration matrix of spiked samples.