# Peer review of "Technical Note: Uncovering the influence of methodological variations on the extractability of iron bound organic carbon"

_Biogeosciences, 2020_

## Referee Comment (RC1) · Susann Henkel (Referee) · 20 Nov 2020

Review to "Technical Note: Uncovering the influence of methodological variations on the extractability of iron bound organic carbon" by Fisher et al.

Fisher et al. investigated how modifications of the frequently used citrate-dithionite-buffer (CDB) extraction for iron-bound organic carbon influence the respective results. The CDB method is widely applied in soil and marine sciences to extract iron (Fe) and co-precipitated or adsorbed organic matter. Despite its common application, the method has some drawbacks that are, according to the authors, often neglected or at least not properly discussed. In this sense, this study reminds me of the recent publication by Hepburn et al. in Chem. Geol.: "The use of operationally-defined sequential Fe extraction methods for mineralogical applications: A cautionary tale from Mössbauer spectroscopy" and the study by Oonk et al. (2017, Chem. Geol.): "Fraction-specific controls on the trace element distribution in iron formations: Implications for trace metal stable isotope proxies". As in these previous publications, Fisher et al. try to tackle the problem that wet chemical extractions lead to operationally defined fractions that are not entirely specific to distinct minerals. The authors set up experiments where they varied the strength of the chemical extract as well as the composition of the sample that's to be leached. They also tested whether a longer duration of the CDB treatment leads to higher Fe and Fe-OC yields. Studies like this are urgently needed to achieve comparability of datasets even though they unfortunately never result in a crystal clear recipe that is to be preferred for all kind of samples. However, this article will make researchers more aware of the shortcomings of the CDB method so that they are put into a better position to judge in which way they should apply it and discuss their data. The manuscript is for the most part well written and easy to understand. The figures and tables are adequate and the discussion is supported by the presented data. What is missing a bit is a wider implication of the finding that CDB treatments do not lead to a full recovery of present reactive Fe in any of the tested samples. CDB is not only used for Fe-bound OC, but also for Fe-bound phosphate (see papers by Ruttenberg, Slomp, Kraal) and of course Fe-oxide extractions after Poulton and Canfield (2005), whereby dithionite was recently shown to also extract substantial amounts of magnetite (and clay). This might not be the exact topic of this article, but should at least be mentioned as I feel that it would increase the relevance of this article. The authors say that FeR extraction was incomplete for their synthesized sediment samples. I assume that it could potentially also be overestimated in some cases. (At least this is what I often observed and has been described in previous papers.) So my main recommendation would be to revise and complement the discussion accordingly and to expand the "framework" of the discussion a bit further in order to address more readers and demonstrate the real relevance of this nice experimental work. I will give some more recommendations

in the following and recommend publication of this study after major revision.

General comments: There should not be a period after a title. (You wrote e.g. "Abstract.") Titles aren't sentences. The manuscript should be checked for a consistent use of the expressions "concentration" and "content". I recommend reading Tolhurst et al. (2005, Estuarine, Coastal and Shelf Science): "Content versus concentration: Effects of units on measuring the biogeochemical properties of soft sediments". Furthermore, please check the order of references in the text. The cited publications should be ordered according to their year of publication.

Scientific remarks:

Line 86-88: "Wagai and Mayer (2007) performed a 16 hour extraction (substituting citrate with weak HCL acid rinses to avoid use of organic compounds), and Patzner et al. (2020) extended to 6 hours." "HCl" instead "HCL". And just a comment: I'm a bit puzzled by this statement regarding citrate. Citrate is added so that Fe-complexes are formed and Fe is kept in solution. I should probably read the paper by Wagai and Mayer, but acid rinses seem critical to me when it comes to comparability of datasets (which is obviously why you investigated it).

Lines 278-281: While 30-40% OC-FeR content is above the average for marine sediments, many samples exist in the 20-30% range. Indeed, the average value for marine sediment OC-FeR composition given by Lalonde et al. (2012) is greater than 20% with individual marine sediments recorded as exceeding 30% OC-FeR (e.g. Equatorial Pacific 0°N, 34.79% (Barber et al., 2017))." I am, to be honest, a bit confused by these numbers. 20-30% of Fe-OC really seems high to me. I never had such high amounts of reactive Fe. Fe plus bound OC is surely higher than reactive Fe alone, but with dithionite you typically reduce all kinds of Fe oxides including ferrihydrite, lepidocrocite, hematite and goethite as well as (unfortunately) some magnetite. (At least with the Poulton and Canfield method published in 2005.) Nevertheless, by applying this method I never ended up with more than 3 wt% extracted Fe in the sediment

out of usually around 6 wt% total Fe including all silicate Fe and sulfides. Please double-check your numbers! As mentioned above, the usual CDB extraction includes crystalline phases like goethite and hematite that might not be so relevant for OC. I am missing a statement concerning how the (maybe in your case unintended?) leaching of more crystalline phases potentially skews the FeR : OC relationship. The typical amount of highly reactive Fe (amorphous phases) in shelf sediments is, I would say, less than 1 wt% (so by far lower than what you were testing for). So I would therefore be a bit hesitant to transfer the results of your experimental data to real marine sediments and it's good that you included tests with Antarctic sediments in this study.

Lines 324-328: "We postulate that freeze drying-induced aggregation of sediment particles could result in reduced Fe extractability compared to non-dried samples since grain size is a known key factor in limiting determination of bioavailable Fe (Raiswell et al., 1994)."

I'd actually argue the other way around. I am wondering about the potential differences between grinded and non-grinded natural samples. You typically freeze-dry samples to be able to grind them and make them more homogenous. I would assume that the freeze-drying itself might result in a transfer of Fe from a more reactive into a less reactive pool. But at the same time I would guess you reduce effects of grain size differences or clogging/shielding of grains (coatings) by grinding the samples. Ok. I see that you mention this in the following sentence. (Add a space before "The influence. . .") As grinding is what's typically done, I'm not convinced that the aggregation plays the dominant role. I'd rather think that the amorphous Fe compounds aren't stable during the processing (freeze drying). Would be worth checking whether there is a transformation of ferrihydrite during and after drying...

Line 335-336: "The alternate tested method of using wet samples has largely been avoided, with only a few studies (e.g. van Bodegom et al., 2003;Chen et al., 2020) reporting the use of a wet slurry sample in soils and none for sediments." Suggestion: The alternative method of using wet samples has largely been avoided, with only a few

exceptions in soil studies (e.g. van Bodegom et al., 2003; Chen et al., 2020).

Your statement is not true when you don't limit your view to the Fe-OC extraction by dithionite but also consider the many studies focussing on the Fe or P. The Poulton and Canfield (2005) method that includes a similar dithionite step is often applied to wet sediments. Check papers by Natascha Riedinger, Laura Wehrmann and Katja Laufer (2019, Reactivity of Iron Minerals in the Seabed Toward Microbial Reduction – A Comparison of Different Extraction Techniques). The same is true for Fe-P extractions with CDB (Kraal, e.g. 2017 GCA paper). One reason for people sticking to the freeze-drying and grinding is that with lots of samples, that's the only option. I'm thinking of IODP material (usually pretty hard mud rock) or black shales.

Lines 370-373: "As we observed incomplete Fe extraction (Fig. 1) for all our samples, a range of CBD extraction times were trialled to understand whether increasing the length of a reaction would increase Fe liberated, as seen for other chemical Fe extractions; oxalate, for example, is known to continue to extract Fe beyond a standard 1 hour treatment (McKeague and Day, 1966)." Okay, but it does not make too much sense to compare the CDB method to the oxalate method, because the oxalate extraction works differently. The extraction is actually catalyzed by dissolved $Fe^{2+}$. So the longer the extraction continues, the more $Fe^{2+}$ is in solution and the stronger gets the extraction (well described in Oonk et al., 2017, Chem Geol. and references therein).

Line 415-417: "We suggest that if future studies were to increase Na dithionite addition in the CBD method this should be followed by a similar increase in trisodium citrate to ensure the entire reduced Fe pool is complexed, preventing precipitation of Fe before quantification." I ran some tests with citrate myself with Fe contents that are comparable to natural occurrences and found that it's usually not limiting. The citrate concentration can in fact be reduced compared to original protocols (I tested the Poulton and Canfield method) as long as you work under strictly anoxic conditions (Henkel et al. 2016, Chem. Geol.). Might be a good alternative.

More specific technical remarks (sentence structure, typos etc.):

Line 16: I suggest to use "synthesized sample" instead of "sediment".

Line 36: Delete "important" before "for water retention" as it is an unnecessary repetition.

Line 99: "...rapid decomposition of dithionite in aqueous form suggesting, a quick loss of reduction potential..." Incorrect comma placement.

Line 118-119: "To achieve this, we mixed the precipitate with a marine sediment 'carrier' material as described by Fisher et al. (2020), using the same original carrier sample and similarly treated to liberate OC and inorganic carbon." Weird sentence structure. In order to make it easier for the reader I suggest to include one or two sentences to what the carrier material is. I guess the original FeR contents are known? It's fine to refer to the previous publication, but the reader shouldn't be "forced" to look it up.

Line 126: Replace "A" by "The".

Lines 164-166: "Initial concentrations of Fe in synthetic samples were obtained by digesting ~2 mg of dried sample in 1 mL 12N HCl at room temperature followed by a 10-fold dilution with 1% HCl solution. Further dilutions were made 165 as necessary, dependent on Fe content, using MilliQ water to produce a subsample within the detectable window (1–10 ppm Fe)." I'm not quite sure about the fraction that is intended to be dissolved here. Bulk Fe? I guess it's okay in case that the synthetic sample does not contain Fe-bearing silicates. As mentioned above it would be good to add what was used as "carrier" for the Fe oxide-OC spikes.

Add "of the extract" after "10-fold dilution".

Section 2.6 about ICP-OES analyses: Generally (for future), I would recommend using an internal standard for correction of different ionic strengths.

Line 171-173: "Extraction of Fe was calculated by subtracting the amount of Fe lost in the control experiment from Fe lost following extraction, then subtracting this from the initial Fe of each sample." The formulation of this sentence seems more complicated than necessary and I don't fully get it. How about: "The recoveries of the extractions were determined as extracted Fe compared to the initial Fe content." (I suppose you made sure that the carrier sediment that you spiked did not contain any Fe?)

Line 188: "...requires a 0.25 g addition relative to 0.25 g of dried sediment sample" Recommend to use "per" instead of "relative to".

Lines 191-194: "All samples show incomplete reduction of Fe regardless of Na dithionite addition, with those samples containing the least Fe proving extractable for the greatest proportion of Fe." Unnecessarily complicated formulation. How about: "highest recovery of Fe in samples with low OC-Fe contents".

Line 199-201: "From this, we can deduce the maximal %Fe in sediment extractable by 0.25 g Na dithionite lies between a 20 and 30% OC-FeR mix, equivalent to 7-10 wt% Fe content in the sediment." I would slightly reformulate the last part of the sentence to not imply that this is total Fe you're talking about. And wouldn't it make sense (for practical reasons) to translate your "20 and 30% OC-FeR mix" into an absolute amount of Fe (e.g. in mmol or mg) that can be liberated?

Lines 203-205: This is about LECO data, right? I wonder whether you could avoid confusion by just calling it "extracted OC" or OCFeR. Calling this fraction OC-FeR is a bit confusing as I would intuitively translate it as "OC-bound reactive Fe". But you mean "reactive Fe-bound OC". As for the Fe I assume that your carrier did not contain any further OC?

Line 255-256: "For the four synthetic samples we subjected to dithionite reduction, these differed in composition (7-24 wt% Fe, 20-50% initial OC-FeR content)." Didn't you also have a batch with 100% OC-FeR??? (See Table 2.)

Lines 256-257: "The concentration of Fe in these samples results in an effective dithionite to (wt) Fe reduction reaction ratio of 1:0.07-0.24." Replace "concentration of Fe" by "Fe contents", "results" by "resulted" and replace "effective dithionite to (wt) Fe reduction reaction ratio" by "effective dithionite to Fe mass ratio".

Line 261-262: "This has the potential to drive wt% Fe higher in small samples of sediment such as those treated by the method (0.25 g)." Recommend to replace "treated by the method (0.25 g)" by "typically used for the CDB extraction".

Lines 266-269: "Maximal extraction here is defined as the point from which further addition of Na dithionite does not increase the extraction of Fe beyond the amount of Fe extracted under the previous dithionite addition mass $\pm$ error. For example, the 20% OC-FeR sample subject to 0.25 g dithionite is removable for 88.79% $\pm$ 3.55 of FeTotal while 0.375 g addition extracts 90.94% $\pm$ 3.64; ..." What is meant by "is removable for"??? Unnecessarily complicated formulation. Use "yields" or "liberates". Delete "beyond the amount of Fe extracted under the previous dithionite addition mass $\pm$ error" and add a "further" before "increase". How this is meant is getting clear through your example.

Line 276: Missing space before 2.69.

Line 276-278: "This finding demonstrates that the OC-FeR composition would not be correctly determined following the method of Lalonde et al. (2012) for these OC-FeR rich sediments, and the overall extent of OC-FeR in the marine sediment pool would be underestimated. You can delete the "pool". It kind of implies that you're talking of a specific fraction of the marine sediment, but here you mean the sediment itself (bulk). Why would you limit this to marine sediments? Couldn't you say this is a general outcome of your study no matter which sediment (fluvial or marine or soil) is used? (Now, again, it would be nice to know the composition of your carrier material.) I would write "amount" instead of "extent".

Line 290-291: "If the increased strength dithionite treatment increases dissolved Fe

beyond the complexing capacity of citrate, then excess Fe likely precipitates out of solution before measurement." This can be avoided when performing the extraction under anoxic conditions (e.g. Henkel et al. 2016).

Line 293-295: "Measurement of OC-FeR extracted for the concentration of Na dithionite at which maximum Fe is extracted showed incomplete OC-FeR loss (Fig. 1)." I would replace "loss" by "liberation" or "recovery".

Line 294-295: "The similarity of OC-FeR and raw Fe extraction values indicates that OC and Fe are reductively released from the sediment in comparable proportions, as is expected due to the low molar OC:FeR ratio of the coprecipitate ($\sim$0.7:1) ." What is meant by "raw" values? Raw data is typically used in another sense.

Lines 297-298: "...could benefit from using increased strength Na dithionite compared to the 0.1 M treatment currently used." Or shorter: "compared to the conventional 0.1 M treatment".

Line 308: Replace "have been" by "are".

Line 309: Replace "defined" by "assessed so far".

Line 346-347: However, the use of wet sediments is likely to be inappropriate for some analyses or sample sites. Yes! You should add one or two sentences to that. I believe it's for most cases not as if people using these methods are not aware of its shortcomings.

Line 354: You can delete the "method" after "storage". "Any storage" is enough.

Line 366: "...slurry form..." Delete "form".

Lines 373-375: "Additionally, as previously mentioned, some iterations of the CBD method have been repeated multiple times in succession to extract the full FeCBD pool, but it is unclear whether time or reagent concentration limit full extraction of this pool on the first treatment." By you or others? It's not getting clear here. "Iteration repeated

multiple times in succession..." Here you say the same thing twice (or actually three times).

Line 379: "...concluding that an increase in chemical exposure time has no difference on Fe extractability." Replace "has no difference on" by "has not enhanced" or "has no effect on".

Line 381: "We would perhaps not expect any benefit from increasing the length of CBD treatment as dithionite, ..." You don't seem to be very convinced by your data. Replace "would perhaps not" by "do not".

Line 382-383: "... with a rapid second order rate constant (K2) of 3.0 (g-molecule/L)-1 min-1 at 79.4 °C, ..." The unit is written in an unnecessarily complicated way. I guess it should be L/(mol*min)? Please check!

Lines 421-422: "Freeze drying induced aggregation appears to reduce Fe liberation in synthetic coprecipitates that were freeze dried relative to slurried, however, we were unable to replicate this increased extraction for natural samples." Suggestion: Freeze drying induced aggregation appears to reduce Fe liberation from synthetic coprecipitates. However, we were unable to confirm this reduced Fe extraction for a set of natural samples.

Lines 422-424: "While we speculate this may be due to the use of freeze thawed samples, which can introduce aggregation in itself, it is hard to see a practical implementation of this adjustment for marine sediments due to the difficulty in transport of pristine samples." Replace "which" by "where freeze thawing", otherwise your reference isn't fully correct. (You'd refer to the samples and not the process of thawing.) And I believe you can delete the "in" before "itself".

Line 425: Add "the" before "dry weight".

Line 427: Period missing after "extraction".

Figure 1: I have difficulties understanding your black and blue symbols. Shouldn't

the percentage of extracted Fe be equivalent to the extracted OC-Fe? Or is the data behind the blue symbols the LECO-data? Do you really need the separate axis with the different scale??? It's just (at first glance) confusing that e.g. the blue diamond is so much further up the fitted curve. And the offset in "dithionite added" between blue and black symbols (equivalent to maybe 0.1 g) is odd, too.

Figure 2: You don't need 3 different patterns if you distinguish between the different OC contents by different colors (gray scales). So reduce the complexity of this graph by just using 3 colors for the three differing OC batches and filled vs. hatched bars for dry and wet. I would also (for clarity) change the figure a bit so that it doesn't appear as if the OC-Fe to total sediment ratio was 5% for the lowermost wet batch 3 COOH mix and close to 30% for the lowermost dry batch 1 COOH mix. You know what I mean? Those extractions all belong to the 20% test, right? Figure caption: Colon after "Figure 2".

Figure 3: Colon after "Figure 3".

Table 1: Use format "left-aligned" in the first column.

Table 2: I find the expression "%OC-Fe:sediment" a bit confusing. I guess you mean % of OC-Fe coprecipitate to total sample". It's inconsistent because when you write "Sediment (mg)" you mean the carrier only.

I trust this review is fair and constructive.

Susann Henkel

---

## Author Comment (AC1) · 26 Nov 2020

Response to review by Susann Henkel.

Original colours in black, author responses are in blue

Review to "Technical Note: Uncovering the influence of methodological variations on the extractability of iron bound organic carbon" by Fisher et al.

Fisher et al. investigated how modifications of the frequently used citrate-dithionite buffer (CDB) extraction for iron-bound organic carbon influence the respective results. The CDB method is widely applied in soil and marine sciences to extract iron (Fe) and co-precipitated or adsorbed organic matter. Despite its common application, the method has some drawbacks that are, according to the authors, often neglected or at least not properly discussed. In this sense, this study reminds me of the recent publication by Hepburn et al. in Chem. Geol.: "The use of operationally-defined sequential Fe extraction methods for mineralogical applications: A cautionary tale from Mössbauer spectroscopy" and the study by Oonk et al. (2017, Chem. Geol.): "Fraction-specific controls on the trace element distribution in iron formations: Implications for trace metal stable isotope proxies". As in these previous publications, Fisher et al. try to tackle the problem that wet chemical extractions lead to operationally defined fractions that are not entirely specific to distinct minerals. The authors set up experiments where they varied the strength of the chemical extract as well as the composition of the sample that's to be leached. They also tested whether a longer duration of the CDB treatment leads to higher Fe and Fe-OC yields. Studies like this are urgently needed to achieve comparability of datasets even though they unfortunately never result in a crystal clear recipe that is to be preferred for all kind of samples. However, this article will make researchers more aware of the shortcomings of the CDB method so that they are put into a better position to judge in which way they should apply it and discuss their data. The manuscript is for the most part well written and easy to understand. The figures and tables are adequate and the discussion is supported by the presented data.

We thank Dr Henkel for their review and agree that the overall importance of this study, and that of the similar studies outlined, is to increase the awareness of the factors which can influence the efficiency of these type of operationally defined chemical extractions.

What is missing a bit is a wider implication of the finding that CDB treatments do not lead to a full recovery of present reactive Fe in any of the tested samples. CDB is not only used for Fe-bound OC, but also for Fe-bound phosphate (see papers by Ruttenberg, Slomp, Kraal) and of course Fe-oxide extractions after Poulton and Canfield (2005), whereby dithionite was recently shown to also extract substantial amounts of magnetite (and clay). This might not be the exact topic of this article, but should at least be mentioned as I feel that it would increase the relevance of this article.

The CBD approach we apply here (per Lalonde et al. (2012)) differs from the method used for Fe-oxides by Poulton and Canfield (2005) both due to being operated at a higher pH and using a bicarbonate buffer rather than acetic acid. We currently cite Thompson et al. (2019) who compares the two different extractions on line 68 as a reference to this *"It should be noted, however, that lower pH extractions are known to be more efficient at extracting the targeted reactive Fe phases if the co-extracted organic compounds are not of interest (Thompson et al., 2019)."* We therefore did not seek to apply our results, showing incomplete recovery of reactive Fe in a pH 7 system, to suggest that this would similarly apply to the pH 4.8 extraction.

We thank Dr Henkel for the suggestion of including the Fe bound phosphate literature for the traditional SEDEX protocol, as this is a more comparable method (circumneutral pH). We accept the point that this could increase the relevance of the paper, so have, in the introduction made reference to these methods.

Line 45 onwards now includes: *"The circumneutral pH CBD extraction has also been used as part of the original SEDEX protocol for the extraction of Fe bound phosphate ($Fe_P$) (Ruttenberg, 1992;Kraal et al., 2012). Although thermodynamically different from the CBD extraction for OC-$Fe_R$ (8 hours at 25 °C vs 15 minutes at 80 °C), Slomp et al. (1996) showed no difference between the efficiency of this extraction and the shortened high temperature extraction of Mehra and Jackson (1958). While Ruttenberg (1992) and Thompson et al. (2019) report 90-100% of synthetic ferrihdyrite is extracted by the CBD method for $Fe_P$ it should be noted that the effective dithionite concentration used here is potentially more than twice as strong than that used for the OC-$Fe_R$ extraction by Lalonde et al., 2012, (1.125g dithionite for 0.5g sediment vs 0.25g dithionite for 0.25g sediment). However, it has recently been shown that CBD is less efficient at extracting crystalline hematite than previously thought, with only 18.4 ± 0.7% of Fe in a hematite sample recovered by Thompson et al. (2019). Further, this inefficiency has been similarly shown in the context of OC-$Fe_R$ extractions conducted at the lower dithionite strength with Adhikari and Yang (2015) reporting only 5-44% of OC was released from hematite-humic acid complexes upon Fe dissolution."*

The authors say that FeR extraction was incomplete for their synthesized sediment samples. I assume that it could potentially also be overestimated in some cases. (At least this is what I often observed and has been described in previous papers.) So my main recommendation would be to revise and complement the discussion accordingly and to expand the "framework" of the discussion a bit further in order to address more readers and demonstrate the real relevance of this nice experimental work. I will give some more recommendations in the following and recommend publication of this study after major revision.

General comments:

There should not be a period after a title. (You wrote e.g. "Abstract.") Titles aren't sentences.

Removed for all titles and subheadings.

The manuscript should be checked for a consistent use of the expressions "concentration" and "content". I recommend reading Tolhurst et al. (2005, Estuarine, Coastal and Shelf Science): "Content versus concentration: Effects of units on measuring the biogeochemical properties of soft sediments". Furthermore, please check the order of references in the text. The cited publications should be ordered according to their year of publication.

Use of "content" and "concentration" have been checked, a number of changes have been made to the use of "concentration", primarily to change this to content when referring to solids (e.g. the content of OC-Fe in a sediment sample). In text citations are formatted according to the journal policy.

Scientific remarks: Line 86-88: "Wagai and Mayer (2007) performed a 16 hour extraction (substituting citrate with weak HCL acid rinses to avoid use of organic compounds), and Patzner et al. (2020) extended to 6 hours." "HCl" instead "HCL". And just a comment: I'm a bit puzzled by this statement regarding citrate. Citrate is added so that Fe-complexes are formed and Fe is kept in solution. I should probably read the paper by Wagai and Mayer, but

acid rinses seem critical to me when it comes to comparability of datasets (which is obviously why you investigated it).

Corrected HCL to HCl. Wagai and Mayer use weak acid rinses instead of citrate to try and create an organic free extraction (since the method seeks to measure organic carbon). They say "In our fully inorganic dithionite extraction, we eliminated the citrate to allow measurement of liberated OC, which we compensated by a weak acid rinse of residues following dithionite extraction to redissolve Fe precipitated as acid-volatile sulfides and associated OC."

We have added the section about redissolving Fe to clarify this in our text. This now reads *"Patzner et al. (2020) extended the CBD extraction to 6 hours and Wagai and Mayer (2007) performed a 16 hour inorganic extraction. In this method citrate, used to complex Fe, was substituted with a weak HCl rinse to redissolve precipitated Fe and avoid the interference of citrate (an organic compound) in OC quantification."*

Lines 278-281: While 30-40% OC-FeR content is above the average for marine sediments, many samples exist in the 20-30% range. Indeed, the average value for marine sediment OC-FeR composition given by Lalonde et al. (2012) is greater than 20% with individual marine sediments recorded as exceeding 30% OC-FeR (e.g. Equatorial Pacific 0□N, 34.79% (Barber et al., 2017))." I am, to be honest, a bit confused by these numbers. 20-30% of Fe-OC really seems high to me. I never had such high amounts of reactive Fe. Fe plus bound OC is surely higher than reactive Fe alone, but with dithionite you typically reduce all kinds of Fe oxides including ferrihydrite, lepidocrocite, hematite and goethite as well as (unfortunately) some magnetite. (At least with the Poulton and Canfield method published in 2005.) Nevertheless, by applying this method I never ended up with more than 3 wt% extracted Fe in the sediment out of usually around 6 wt% total Fe including all silicate Fe and sulfides. Please double-check your numbers!

These are two different things, in the manuscript we are discussing OC-Fe$_R$ here (i.e. the proportion of organic carbon bound to reactive iron as a fraction of total organic carbon). This is different to the Fe-OC relationship that the reviewer mentions (the proportion of the iron pool bound to organic carbon), we agree that 20-30% Fe-OC would be incredibly high and this is not something we are claiming to represent. Reference to absolute amounts of Fe in sediment are referred to by wt% throughout the manuscript.

As mentioned above, the usual CDB extraction includes crystalline phases like goethite and hematite that might not be so relevant for OC. I am missing a statement concerning how the (maybe in your case unintended?) leaching of more crystalline phases potentially skews the FeR : OC relationship. The typical amount of highly reactive Fe (amorphous phases) in shelf sediments is, I would say, less than 1 wt% (so by far lower than what you were testing for). So I would therefore be a bit hesitant to transfer the results of your experimental data to real marine sediments and it's good that you included tests with Antarctic sediments in this study.

Yes, CBD can leach goethite and hematite in addition to, e.g., ferrihydrite, and we refer to this in the now modified section in the introduction by including values for hematite leaching from the Thompson et al., 2019 study. Hematite can play a role in OC binding, so in the same section we refer to the study by Adhikari and Yang on OC release from hematite-humic acid complexes. The reviewer is correct that leaching of Fe phases not associated with OC can skew the OC-Fe$_R$ relationship and this is shown in the Barber et al., (2017) study. We have not previously discussed this in this manuscript as we only include

ferrihydrite in our experimental system (in part to get around problems such as these), so leaching of other phases is not directly relevant for this study.

Lines 324-328: "We postulate that freeze drying-induced aggregation of sediment particles could result in reduced Fe extractability compared to non-dried samples since grain size is a known key factor in limiting determination of bioavailable Fe (Raiswell et al., 1994)." I'd actually argue the other way around. I am wondering about the potential differences between grinded and non-grinded natural samples. You typically freeze-dry samples to be able to grind them and make them more homogenous. I would assume that the freeze-drying itself might result in a transfer of Fe from a more reactive into a less reactive pool. But at the same time I would guess you reduce effects of grain size differences or clogging/shielding of grains (coatings) by grinding the samples. Ok. I see that you mention this in the following sentence. (Add a space before "The influence. . .") As grinding is what's typically done, I'm not convinced that the aggregation plays the dominant role. I'd rather think that the amorphous Fe compounds aren't stable during the processing (freeze drying). Would be worth checking whether there is a transformation of ferrihydrite during and after drying...

We have previously shown that the complexes we produce are definitely still 2-line ferrihydrite by XRD of the freeze-dried precipitate in Fisher et al., (2020) so transformation was ruled out as a reason for the difference between Fe recovery from dried and non-dried precipitates. We acknowledge later in the manuscript that grinding is typically performed but to an undefined grain size ("finely ground" or similar), which can introduce error.

Line 329 "*McKeague and Day (1966) similarly report that finer grinding of sediment resulted in an increased extraction of Fe. These findings indicate that particle size is a critical parameter in determining the amount of Fe extracted, however, all methods fail to define what is meant by "finely ground". This lack of definition introduces an error of reproducibility as particle size is certain to vary with different sample preparation methods and therefore two identical chemical treatments may vary in extraction effectiveness because of physical differences in the sediment sample.*"

We have added a sentence to confirm transformation of the Fe phase did not occur during drying.

"*An alternate hypothesis to describe the reduced Fe recovery for dried sediment, i.e., that transformation of ferrihdyrite occurred during freeze drying, was ruled out by x-ray diffraction (XRD) characterisation of the freeze dried phase as 2-line ferrihyrite (Fisher et al., 2020).*"

Line 335-336: "The alternate tested method of using wet samples has largely been avoided, with only a few studies (e.g. van Bodegom et al., 2003;Chen et al., 2020) reporting the use of a wet slurry sample in soils and none for sediments." Suggestion: The alternative method of using wet samples has largely been avoided, with only a few exceptions in soil studies (e.g. van Bodegom et al., 2003; Chen et al., 2020). Your statement is not true when you don't limit your view to the Fe-OC extraction by dithionite but also consider the many studies focussing on the Fe or P. The Poulton and Canfield (2005) method that includes a similar dithionite step is often applied to wet sediments. Check papers by Natascha Riedinger, Laura Wehrmann and Katja Laufer (2019, Reactivity of Iron Minerals in the Seabed Toward Microbial Reduction – A Comparison of Different Extraction Techniques). The same is true for Fe-P extractions with CDB (Kraal, e.g. 2017 GCA paper). One reason for people sticking to the freezedrying and grinding is that with lots of samples, that's the only option. I'm thinking of IODP material (usually pretty hard mud rock) or black shales.

We thank Dr Henkel for highlighting these papers, we agree it would be suitable to include these as examples of wet sediment treatments. However, not all the suggested papers actually use wet sediment (e.g., the Kraal 2017 GCA paper explicitly uses freeze-dried "ground sediment of 50 and 100 mg for sequential extraction of Fe"). While Laufer et al., (2019), Wehrmann et al., (2014) and Riedinger et al., (2017) do use wet sediment, these studies all freeze sediment at -20°C which we later go on to discuss has the potential to introduce freeze-thaw aggregation.

We have now cited these studies alongside our use of Arctic freeze-thawed samples.

*"Wet thawed samples have been used more widely in the sequential extraction of Fe (e.g. Laufer et al., 2020, Riedinger et al., 2017, Wehrmann et al., 2014), additionally the Arctic sediment sample used in our analysis was similarly subject to freeze-thawing. However, our freeze-thawed sample showed no difference in its recovery for Fe compared to the dried aliquot, indicating no significant effect of the thawing stage."*

Lines 370-373: "As we observed incomplete Fe extraction (Fig. 1) for all our samples, a range of CBD extraction times were trialled to understand whether increasing the length of a reaction would increase Fe liberated, as seen for other chemical Fe extractions; oxalate, for example, is known to continue to extract Fe beyond a standard 1 hour treatment (McKeague and Day, 1966)." Okay, but it does not make too much sense to compare the CDB method to the oxalate method, because the oxalate extraction works differently. The extraction is actually catalyzed by dissolved $Fe^{2+}$. So the longer the extraction continues, the more $Fe^{2+}$ is in solution and the stronger gets the extraction (well described in Oonk et al., 2017, Chem Geol. and references therein).

This is a fair point, we have removed the reference to oxalate and changed this to reflect the OC-Fe dithionite based studies with variable time and linked this to the Fe-P methods previously added.

*"As we observed incomplete Fe extraction (Fig. 1) for all our samples, a range of CBD extraction times were trialled to understand whether increasing the length of a reaction would increase Fe liberated, as seen in some iterations of the CBD method for OC-Fe$_R$ (e.g. Patzner et al., 2020, Wagai and Mayer, 2007) as well as those for Fe$_P$ (Ruttenberg, 1992)."*

Line 415-417: "We suggest that if future studies were to increase Na dithionite addition in the CBD method this should be followed by a similar increase in trisodium citrate to ensure the entire reduced Fe pool is complexed, preventing precipitation of Fe before quantification." I ran some tests with citrate myself with Fe contents that are comparable to natural occurrences and found that it's usually not limiting. The citrate concentration can in fact be reduced compared to original protocols (I tested the Poulton and Canfield method) as long as you work under strictly anoxic conditions (Henkel et al. 2016, Chem. Geol.). Might be a good alternative.

We have added this as an alternative earlier in the discussion where we first mention increasing citrate. Line 303 now includes: *"Alternatively, Henkel et al. (2016) found that a reduced concentration of dithionite is sufficient to fully complex the reduced Fe pool when the extraction is performed under anoxic conditions, which may remove the need to further increase the addition of citrate as an organic reagent."*

 More specific technical remarks (sentence structure, typos etc.):

Line 16: I suggest to use "synthesized sample" instead of "sediment".

Changed to synthetic sample

Line 36: Delete "important" before "for water retention" as it is an unnecessary repetition.

Deleted

Line 99: ". . .rapid decomposition of dithionite in aqueous form suggesting, a quick loss of reduction potential. . ." Incorrect comma placement.

Comma removed

Line 118-119: "To achieve this, we mixed the precipitate with a marine sediment 'carrier' material as described by Fisher et al. (2020), using the same original carrier sample and similarly treated to liberate OC and inorganic carbon." Weird sentence structure. In order to make it easier for the reader I suggest to include one or two sentences to what the carrier material is. I guess the original FeR contents are known? It's fine to refer to the previous publication, but the reader shouldn't be "forced" to look it up.

Replaced with: "To achieve this we spiked the precipitate into a marine sediment sample from the Barents Sea (water depth 141 m; sediment core depth, 33.5 cm; station B6, E40; cruise JR16006). This sample was ashed (650 °C, 12 hrs) to remove OC and fumigated with HCl vapour to remove inorganic carbon. The resulting carrier material was siliciclastic in nature with a Fe content of 16.33 mg/g."

Line 126: Replace "A" by "The".

Corrected

Lines 164-166: "Initial concentrations of Fe in synthetic samples were obtained by digesting ~2 mg of dried sample in 1 mL 12N HCl at room temperature followed by a 10-fold dilution with 1% HCl solution. Further dilutions were made as necessary, dependent on Fe content, using MilliQ water to produce a subsample within the detectable window (1–10 ppm Fe)." I'm not quite sure about the fraction that is intended to be dissolved here. Bulk Fe? I guess it's okay in case that the synthetic sample does not contain Fe-bearing silicates. As mentioned above it would be good to add what was used as "carrier" for the Fe oxide-OC spikes.

Details on spike content has been added with the previous comment. The digest here is to determine the initial Fe content (i.e., the ferrihydrite we add), 12N HCl is more than sufficient to dissolve the spiked ferrihydrite and any reactive Fe in the synthetic sample.

Add "of the extract "after "10-fold dilution".

Corrected

Section 2.6 about ICP-OES analyses: Generally (for future), I would recommend using an internal standard for correction of different ionic strengths.

Atomic absorption spectroscopy (AAS) was used for the analysis of Fe rather than ICP-OES in this study.

Line 171-173: "Extraction of Fe was calculated by subtracting the amount of Fe lost in the control experiment from Fe lost following extraction, then subtracting this from the initial Fe of each sample." The formulation of this sentence seems more complicated than necessary and I don't fully get it. How about: "The recoveries of the extractions were determined as extracted Fe compared to the initial Fe content." (I suppose you made sure that the carrier sediment that you spiked did not contain any Fe?)

Changed to: *"The recovery of Fe following sample extractions was calculated by subtracting the control corrected loss of Fe from the initial Fe content of the sample."*

There was a small amount of Fe in the carrier sediment, as now detailed in the spiking section of the methods. However, this only makes a minor contribution to the overall Fe pool once spiked and not all of this will be reactive. We have no indication that the baseline Fe content affected any of the results, even if this Fe content was to have an affect it would be the same effect on all aliquots and iterations involving the sediment carrier.

Line 188: ". . .requires a 0.25 g addition relative to 0.25 g of dried sediment sample" Recommend to use "per" instead of "relative to".

Corrected

Lines 191-194: "All samples show incomplete reduction of Fe regardless of Na dithionite addition, with those samples containing the least Fe proving extractable for the greatest proportion of Fe." Unnecessarily complicated formulation. How about: "highest recovery of Fe in samples with low OC-Fe contents".

Changed

Line 199-201: "From this, we can deduce the maximal %Fe in sediment extractable by 0.25 g Na dithionite lies between a 20 and 30% OC-FeR mix, equivalent to 7-10 wt% Fe content in the sediment." I would slightly reformulate the last part of the sentence to not imply that this is total Fe you're talking about. And wouldn't it make sense (for practical reasons) to translate your "20 and 30% OC-FeR mix" into an absolute amount of Fe (e.g. in mmol or mg) that can be liberated?

Added: *"Therefore, assuming a 0.25g sample size, the absolute amount of Fe extracted would be between 17.5 and 25 mg. "*

Also changed wt% Fe to wt% $Fe_R$ to clarify this is reactive not total Fe.

Lines 203-205: This is about LECO data, right? I wonder whether you could avoid confusion by just calling it "extracted OC" or OCFeR. Calling this fraction OC-FeR is a bit confusing as I would intuitively translate it as "OC-bound reactive Fe". But you mean "reactive Fe-bound OC". As for the Fe I assume that your carrier did not contain any further OC?

This is about measuring OC, the LECO data, yes. OC-$Fe_R$ is the fraction of organic carbon bound to reactive iron, not reactive iron bound to OC so the reviewer translation is correct. And yes, the carrier is OC free, this has been clarified in response to a previous comment in the methods.

Line 255-256: "For the four synthetic samples we subjected to dithionite reduction, these differed in composition (7-24 wt% Fe, 20-50% initial OC-FeR content)." Didn't you also have a batch with 100% OC-FeR??? (See Table 2.)

The greater than 50% OC-$Fe_R$ values are only used for the wet vs dry comparison as that is a more mechanistic relationship. In this section (4.1), we are discussing the concentration of Na dithionite used to extract OC-$Fe_R$ from samples so the concentrations are closer to what may be found in a natural sediment.

Lines 256-257: "The concentration of Fe in these samples results in an effective dithionite to (wt) Fe reduction reaction ratio of 1:0.07-0.24." Replace "concentration of Fe" by "Fe

contents", "results" by "resulted" and replace "effective dithionite to (wt) Fe reduction reaction ratio" by "effective dithionite to Fe mass ratio".

Changed

Line 261-262: "This has the potential to drive wt% Fe higher in small samples of sediment such as those treated by the method (0.25 g)." Recommend to replace "treated by the method (0.25 g)" by "typically used for the CDB extraction".

Changed

Lines 266-269: "Maximal extraction here is defined as the point from which further addition of Na dithionite does not increase the extraction of Fe beyond the amount of Fe extracted under the previous dithionite addition mass ± error. For example, the 20% OC-FeR sample subject to 0.25 g dithionite is removable for 88.79% ± 3.55 of FeTotal while 0.375 g addition extracts 90.94% ± 3.64; . . ." What is meant by "is removable for"??? Unnecessarily complicated formulation. Use "yields" or "liberates". Delete "beyond the amount of Fe extracted under the previous dithionite addition mass ± error" and add a "further" before "increase". How this is meant is getting clear through your example.

Changed

Line 276: Missing space before 2.69.

Corrected

Line 276-278: "This finding demonstrates that the OC-FeR composition would not be correctly determined following the method of Lalonde et al. (2012) for these OC-FeR rich sediments, and the overall extent of OC-FeR in the marine sediment pool would be underestimated. You can delete the "pool". It kind of implies that you're talking of a specific fraction of the marine sediment, but here you mean the sediment itself (bulk). Why would you limit this to marine sediments? Couldn't you say this is a general outcome of your study no matter which sediment (fluvial or marine or soil) is used? (Now, again, it would be nice to know the composition of your carrier material.) I would write "amount" instead of "extent".

Changed from marine sediment to sample and extent to amount.

Line 290-291: "If the increased strength dithionite treatment increases dissolved Fe beyond the complexing capacity of citrate, then excess Fe likely precipitates out of solution before measurement." This can be avoided when performing the extraction under anoxic conditions (e.g. Henkel et al. 2016).

Added: "Alternatively, Henkel et al. (2016) found that a reduced concentration of dithionite is sufficient to fully complex the reduced Fe pool when the extraction is performed under anoxic conditions which may remove the need to further increase the addition of citrate as an organic reagent."

Line 293-295: "Measurement of OC-FeR extracted for the concentration of Na dithionite at which maximum Fe is extracted showed incomplete OC-FeR loss (Fig. 1)." I would replace "loss" by "liberation" or "recovery".

Changed to recovery

Line 294-295: "The similarity of OC-FeR and raw Fe extraction values indicates that OC and Fe are reductively released from the sediment in comparable proportions, as is expected

due to the low molar OC:FeR ratio of the coprecipitate ($\sim$0.7:1) ." What is meant by "raw" values? Raw data is typically used in another sense.

Deleted raw

 Lines 297-298: ". . .could benefit from using increased strength Na dithionite compared to the 0.1 M treatment currently used." Or shorter: "compared to the conventional 0.1 M treatment".

Changed

Line 308: Replace "have been" by "are".

Changed

Line 309: Replace "defined" by "assessed so far".

Changed

Line 346-347: However, the use of wet sediments is likely to be inappropriate for some analyses or sample sites. Yes! You should add one or two sentences to that. I believe it's for most cases not as if people using these methods are not aware of its shortcomings.

Added: "However, the use of wet sediments is likely to be inappropriate for some analyses or sample sites either due to practical considerations, such as the difficulty in transporting heavy wet sediments, or when there is a need to preserve the sediment profile, for example, protecting anoxic sediments from oxic biological transformations."

Line 354: You can delete the "method" after "storage". "Any storage" is enough.

Deleted

 Line 366: ". . .slurry form. . ." Delete "form".

Deleted

 Lines 373-375: "Additionally, as previously mentioned, some iterations of the CBD method have been repeated multiple times in succession to extract the full FeCBD pool, but it is unclear whether time or reagent concentration limit full extraction of this pool on the first treatment." By you or others? It's not getting clear here. "Iteration repeated multiple times in succession..." Here you say the same thing twice (or actually three times).

Changed: *"Additionally, in some examples of the CBD method the extraction stage is repeated multiple times for the same sample in order to extract the full $Fe_{CBD}$ pool (e.g. Mehra and Jackson, 1958, Aguilera and Jackson, 1953), but it is unclear for these multiple stage treatments which parameter prevents full extraction of $Fe_{CBD}$ on the first treatment."*

Line 379: ". . . concluding that an increase in chemical exposure time has no difference on Fe extractability." Replace "has no difference on" by "has not enhanced" or "has no effect on".

Changed

Line 381: "We would perhaps not expect any benefit from increasing the length of CBD treatment as dithionite, . . ." You don't seem to be very convinced by your data. Replace "would perhaps not" by "do not".

Changed

Line 382-383: ". . . with a rapid second order rate constant (K2) of 3.0 (g-molecule/L)-1 min-1 at 79.4 □C, . . ." The unit is written in an unnecessarily complicated way. I guess it should be L/(mol*min)? Please check!

This was originally taken from the, rather old, citation so can be updated. Gram molecules (g-molecule) are equivalent to moles so this has been simplified to 3.0 mol L$^{-1}$ min $^{-1}$.

Lines 421-422: "Freeze drying induced aggregation appears to reduce Fe liberation in synthetic coprecipitates that were freeze dried relative to slurried, however, we were unable to replicate this increased extraction for natural samples." Suggestion: Freeze-drying induced aggregation appears to reduce Fe liberation from synthetic coprecipitates. However, we were unable to confirm this reduced Fe extraction for a set of natural samples.

Changed

Lines 422-424: "While we speculate this may be due to the use of freeze thawed samples, which can introduce aggregation in itself, it is hard to see a practical implementation of this adjustment for marine sediments due to the difficulty in transport of pristine samples." Replace "which" by "where freeze thawing", otherwise your reference isn't fully correct. (You'd refer to the samples and not the process of thawing.) And I believe you can delete the "in" before "itself".

Changed

Line 425: Add "the" before "dry weight".

Changed

Line 427: Period missing after "extraction".

Changed

Figure 1: I have difficulties understanding your black and blue symbols. Shouldn't the percentage of extracted Fe be equivalent to the extracted OC-Fe? Or is the data behind the blue symbols the LECO-data? Do you really need the separate axis with the different scale??? It's just (at first glance) confusing that e.g. the blue diamond is so much further up the fitted curve. And the offset in "dithionite added" between blue and black symbols (equivalent to maybe 0.1 g) is odd, too.

The percentage of Fe would only be equivalent to OC-Fe if there was a 1:1 OC/Fe ratio. The blue symbols are the LECO data. Blue symbols were originally offset on dithionite concentration to make the symbols easier to distinguish but these can be put back to maintain accuracy. They do need separate axes as they are different measurements (one Fe, one C) regardless of the different scales.

[Figure]

Figure 2: You don't need 3 different patterns if you distinguish between the different OC contents by different colors (gray scales). So reduce the complexity of this graph by just using 3 colors for the three differing OC batches and filled vs. hatched bars for dry and wet. I would also (for clarity) change the figure a bit so that it doesn't appear as if the OC-Fe to total sediment ratio was 5% for the lowermost wet batch 3 COOH mix and close to 30% for the lowermost dry batch 1 COOH mix. You know what I mean? Those extractions all belong to the 20% test, right? Figure caption: Colon after "Figure 2".

Colours have been changed. The y axis represents the original mix of the sample (precipitate to sediment), they are not all the 20% extraction but the 20,40,60,80 and 100% extractions. So the y axis is what was added and the x is what was extracted, added groupings to the y to make it clearer that this is categorical and not continuous data.

[Figure]

Figure 3: Colon after "Figure 3".

Added

Table 1: Use format "left-aligned" in the first column.

Changed

Table 2: I find the expression "%OC-Fe:sediment" a bit confusing. I guess you mean % of OC-Fe coprecipitate to total sample". It's inconsistent because when you write "Sediment (mg)" you mean the carrier only.

Changed to "%OC-Fe in sample".

---

## Author Comment (AC2) · 26 Nov 2020

We received helpful comments via personal communication from Monique Patzner and colleagues, whose pre-print we cite in this manuscript (Patzner et al., 2020). We are grateful to this team for their discussion of our work and have corrected two sections in our manuscript where we cite their recent study.

1) On line 86-88 We correct an error in the original manuscript which suggested the extraction performed by Patzner et al., was 6 hours as oppose to 16, this now reads:

"Patzner et al. (2020) performed the CBD extraction of Lalonde et al., (2012), adjusted

to room temperature, over 16 hours and Wagai and Mayer (2007) performed a 16 hour inorganic extraction"

2) On line 388-390, we suggested that SEM and nanoSIMS analysis was conducted on the CBD extracts when in fact this analysis was conducted on a fine soil fraction separated by a different method. This section has been restructured to remove this previous statement and the final two paragraphs of section 4.3 have been joined, now reading as follows:

"While under the Lalonde et al., (2012) protocol increasing extraction time had no benefit for extracting Fe with the purpose of determining the OC-FeR pool, Patzner et al. (2020) performed an adaptation of the CBD method where time was extended to compensate for a reduction in the temperature of the reaction. A low temperature approach was not tested in our study as we focused on increasing the efficiency, and therefore energy, of the reaction, this adaptation may prove useful should non-destructive analysis be required. For example, subsequent analysis of biomarkers in the extracted organics, something currently not possible due to temperature induced transformation and degradation of OC when heated to 80 °C. This raises an interesting question as to whether temperature and length of the extraction can compensate for each other to achieve the overall same %Fe extraction. While this seems unlikely due to the rapid decrease in the reducing power of dithionite when in solution, the potential merits of this system mean it is worthy of further study. Additionally, it is possible that the decomposition process may occur much slower at room temperature due to the decreased reaction energy. These type of additional analyses may allow us to better understand the origins and molecular composition of OM involved in mineral based preservation processes and offers promising scope for future experimentation with the CBD method."

---

## Referee Comment (RC2) · Tom Jilbert (Referee) · 30 Nov 2020

Fisher et al. present results of experiments into the extractability of OC-FeR (that is to say, sedimentary organic carbon bound to reactive Fe) during treatment with citrate-bicarbonate-dithionite solution. CBD extractions are a commonly applied method, either stand-alone or as part of sequential extractions, for investigating elements associated with reducible phases in marine sediments. As the authors state, there is much heterogeneity in the details of applied CBD extraction protocols, even within the narrower context of studies into OC-FeR. This has led to difficulty in comparing results and the possibility that the currently used protocols may be sub-optimal for their

stated goals. Therefore there is clearly a need for studies like this one, to eventually improve/harmonize the approaches used in the community.

Overall I had the feeling that the study delivers some interesting results although some of the interpretations are left only lightly justified. This leads to the idea that a more developed set of experiments could have yielded a more useful step forward. For example, the conclusion that complexation by citrate may be limiting the recovery of Fe in the highest-OC-FeR experiment deserves to be tested through a concentration series similar to the dithionite series the authors report. This is especially the case considering the comments of the first reviewer (Henkel) questioning whether citrate limitation is a feasible explanation for the observations in Fig. 1. I do not demand that the authors produce such additional data before publication but it is clear that their conclusion would be more robust if it was available, and therefore the overall impact of the study would be greater. A similar criticism could be leveled at the interpretations of the experiment comparing freeze-dried and wet samples, although I would say this is a complex topic that warrants a separate study.

Another important point is that it seems that some of the content here may be an overflow from the authors' recent Chem. Geol. paper (cited Fisher et al. 2020), which is not a criticism as such but in some cases I had the feeling the reader is being referred there to explain what is going on in the experiments presented here, which should be avoided as this ms. must also be a stand-alone study, even if it is a Technical Note. I am specifically referring to the interpretation of the results of the experiment in which the degree of carboxylation of OM in synthetic OC-FeR is varied (Fig. 2). The discussion of the mechanisms here (Paragraph from Line 358) is too thin and the reader cannot understand why the degree of carboxylation makes such a difference to Fe extractability without accessing the other paper.

Unfortunately I read Susann's review only after making my own comments on the original ms., then later noticed that she has done a very comprehensive job and found several of the same issues that I wished to highlight. It is good to see that the authors

have responded thoroughly to Susann's comments and this will undoubtedly improve the next version. Therefore my list of additional comments is comparatively short.

General (in addition to the above; all Line numbers refer to the original ms):

- The Introduction can be better worded and arranged: First I suggest to move the para. starting Line 59 to directly above the short para. starting Line 81. This way you first describe the problems with the existing methodologies, then set out how you intend to solve them. Next, check a few key sentences: e.g. Line 26 "Understanding in which environments organic carbon (OC) persists": please clarify that you are referring to preservation of OC in sediments; Line 54 "fully reduce all solid reactive Fe phases and associated carbon"... I could not find this phrase in Lalonde et al. 2012, although it is presented here as a quotation. Please check.

- Throughout: the terminology in this field is easily misunderstood. E.g. the first reviewer thought for the whole time that % OC-FeR refers to % of total sediment, when in fact it refers to % of total OC. I also had major difficulties to get this upon first reading. So I suggest to clarify terminology early on, and modify figures and captions to make this easier to follow. E.g. I note that Barber et al. 2017 use more descriptive terminology in tables and figures e.g. "OC bound to Fe (% of total OC)" in their Table 1 and "Fraction of total sediment Fe" in their Fig. 4. Also check that CDB/CBD is used consistently. Both current appear.

- Description of Fig. 1 results. The phrase "maximal extraction" is used repeatedly when describing the results, but it is only explained in the Discussion (Line 266). The best place for this description is actually Methods, because you can already state how you intend to use the data to estimate this value. That will make reading the paper a whole lot easier overall.

Specific

- Line 67: maybe qualify with 'partial hydrolysis' or similar. For significant digestion of

OC from sediments, either very low pH (and use of specific oxidizing acids) or very high pH are required.

-Line 89: misplaced comma after "suggesting"

-Line 111: "and" should be "a"

-Line 120: "varied" in preference to "differed"

-Line 192-193: rephrase to "with those samples containing the least Fe showing the greatest proportional/relative extraction of Fe"

-Line 203-205: this looks more like part of a caption for Fig 1

-Line 244-246: Does this mean that the natural sediment samples in these experiments were freeze-thawed before the experiments? If so it will be important to state this in Section 2.2.

-Line 251: why give the formula? there are many Fe oxides that can be dissolved in dithionite so I suggest just to leave it out

Line 259-263: Not clear how XAS can indicate clustering. If you are referring to locally enhanced concentrations ("hotspots"), yes this is a real phenomenon observed by high-resolution mapping techniques. Still, I would be surprised if a homogenized sample of 0.25g would have a distinctly different OC-FeR content from the bulk sediment, so the logic of the statement is not clear and the paragraph does not really benefit from it.

Line 293-294: This is a confusing opening sentence to the paragraph. Rephrase to make more concise.

Line 304-305: It is not clear to me how an increase of DOC (citrate) during the extraction would impact on the quantification of OC after the extraction., if this is done on the solid phase. Can the sample not simply be rinsed before the drying and analysis?

Line 326-328: Clauses of the sentence are not well constructed

---

## Referee Comment (RC3) · Anonymous Referee #3 · 30 Nov 2020

The manuscript is an extended footnote to an earlier paper Fisher et al. paper published this year on the effect of organic acids on reactivity and solubility of iron oxides. The manuscript addressed three aspects of a modified citrate-dithionite-bicarbonate (CDB) extraction method used for determining Fe content associated with various iron oxide mineral phases. Variables tested were the dithionite concentration, freeze-drying versus wet extraction, and time of extraction.

Starting with the title, the goal of the study and the actual study is mismatched. The paper is about extraction of iron oxides, and the entire discussion revolves around the efficiency of the dithionite method toward the extraction iron oxides at circumneutral

pH, not the organic carbon that is extracted. My reading of the Lalonde et al. 2012 article is that they were employing a more gentle (i.e. circumneutral pH) treatment in order to not overestimate the loss of organic carbon due to hydrolysis. That is perfectly reasonable, as they did not want to overestimate OC losses from the main pool due to hydrolysis. Their goal was not to accurately quantify the Fe content, but to dissolve most of the iron oxide fraction and thereby release iron oxide bound organic matter. Here, the authors imply that this approach is not quantitative.

It is not clear at whom or at which samples this study is aimed. The authors only considered ferrihydrite. So is this applicable only for modern sediments? What about sediments or rocks containing greater concentrations of goethite or hematite?

The authors claim that no study has thus far has performed a determination of the reductive capacity of the dithionite method (Lines 74-75). The authors, however, also do not clearly provide the criteria for "reductive capacity". It is only implied in their approach of using varying "weight percentages of OC-Fe" that simulates a titration of sorts. And as described below, there are methodological problems with this approach.

It is not clear that this a substantial step beyond the Fisher et al., 2020 paper. The authors pose the question of whether there is a one-size-fits-all solution (line 95), or should the extraction be adjusted to fit the set of samples and exact research question. But they do not really answer this question. For instance, the effects of freeze drying on wet chemical extractions of sediments as extensively discussed in Section 4.2, has long been known (e.g., Rapin et al., 1986, ES&T; and more recently Raiswell et al., 2004, Chem Geol.). This discussion is superfluous. The authors point out the problems faced by every sediment biogeochemist, but offer no new insights of their own, or at least none that have not been already considered by other studies. They propose no solutions to any of these aspects, except to say that methods employed should be rigorously documented. As the authors point out, analyzing freshly collected wet sediments is not practical for most studies. One has to ask if the efforts to improve the dithionite method are even worth the effort, if freeze-drying is out of the question.
The topic of study is an important one in sediment biogeochemistry, especially in how do we deal with examining organic carbon concentrations and speciation in complex matrices. But does this paper bring about a consensus on how to proceed? Unfortunately, I have to answer, no it does not, outside of stating that when using wet chemical sediment extraction methods, that geochemists should carefully consider the type of sediment being analyzed, the amounts of reactants in the methods, sample storage and the exact question being investigated.

**Methodology**

The high iron oxide contents used in these experiments are problematic. First of all, it is not entirely clear what exactly is being measured (see comments on the term "OC-Fe"). Let's assume that it is %Fe. For example, the 20% OC-Fe sample contains 0.2 x 0.25 g artificial sediment, or 0.05 g Fe. In the 15 mL of reaction solution, this gives ca. 0.06 mole/L Fe. The dithionite solution of 0.25 g Na2S2O4 in 15 mL yields 0.093 moles S2O42- anion per L. Assuming that upon dispropotionation of the dithionite in water yields two reducing equivalents, which is probably overestimated due to side reactions with oxygen and other S decomposition products, we would have

experiment somewhat difficult to interpret.

There appear to be no replicates for each dithionite addition. This makes interpretation of the results, especially in Figure 2, difficult.

Style and Readability

The manuscript would be better served by a radical reduction in length. This is a technical note describing three relatively short comparison experiments that are an extension of the Fisher et al. 2020 paper. For instance, the first two sentences of the manuscript (lines 25 to 27) are obvious to readers of Biogeosciences. There are details (Lines 166- 174) about diluting samples for AAS analysis that do not need to be repeated in such detail. The reader assumes that the authors have a basic understanding concerning the basics of the instrumental analysis. Section 2.7 appears superfluous because there is no where in the Results where organic C is discussed.

On the other hand, the any clear description of the carrier material was lacking, and I had to read the Fisher et al. 2020 Chem Geology article to understand how this key component had been treated.

I am confused by the use of the term "OC-FeR". What exactly is this? Organic C associated with iron oxides, as per Lalonde et al., 2012? Or is this Fe that is somehow made unreactive by Organic Carbon? Or is this simply the total iron oxide content? Or perhaps, the reactive iron content, whatever that may be? Are they referring to %dry weight Fe? Or are they referring to %dry weight FeOOH, or perhaps Fe2O3?, or perhaps %weight of whatever happens to precipitate including the organic fraction added?

**Further Comments**

Line 183 The clause in the first line of the Results has no meaning. The manuscript is plagued by ill-defined discussion of reactivity. There are sentences such as "associated OC has a large influence on Fe reactivity." Towards what?
Line 65: This sentence is misleading. Many permutations, improvements and evaluations to and of the dithionite method have been made, particularly with respect to marine sediments. See for instance Lord 1982 (J Sed Petr.), Kostka and Luther 1994 (GCA) and Raiswell et al. 1994 (Chem Geol). The authors must be referring to the extraction of organic matter. Line 46: This is not surprising as hematite has been shown to be only partially dissolved by CDB method (see Kostka and Luther, GCA, 1994).

Line 48 What do the authors mean be Fe reactivity here? Is this the goal of the study? Or the extractability of organic compounds.

Line 54 "developed" knowledge?

Line 120 This is not a concentration gradient. First of all, the authors are referring to contents, not concentrations (there seems to be confusion about the terms concentration and content throughout this manuscript). Secondly, a gradient implies a change in concentration over some property (e.g. depth, distances, density, etc..)

Line 122 Confusing. Was the carrier material freeze-dried before or after mixing (or not all)?

Line 140 This is repetition of the Lines 80 and following.

Line 170 Samples that were highly concentrated were diluted only 10 times while the more dilute samples were diluted 20 times?

Line 180 It's not clear that the authors differentiate here between a standard that is used for calibration and a secondary standard used as control.

Line 190: This does not show the reductive capacity of the dithionite. If, for instance, dithionite is in excess, then 100% Fe extraction cannot show the reductive capacity of the dithionite.

Line 224 Freeze-thawed samples? This experiment is not mentioned in the methods. Furthermore, this sentence (which is also discussion/interpretation) does not make
sense. What "previous one". The sentence refers to Figure 1. There are no freeze-thaw or freeze-dry samples in Figure 1.

Figure 1: What do the fits represent and how is the fitting done? It looks to me like if you added more dithionite, eventually the %Fe recovery would start to decrease at some point. Also, the blue symbols representing maximal Fe extraction do not match the corresponding curves for the black symbols.

Figure 2 is difficult to interpret. Firstly, the dependent variable is plotted on the x-axis, which is confusing for the bar chart depiction. Secondly, outside of the observation that freeze-dried sediments tend exhibit lower extractability than the fresh samples at high Fe contents, it is difficult to ascertain any kind of trend. Given the lack of replicates for each sample, and the large degree of variability in extractabilities, I find it difficult to be able to say anything concrete about these results.

Line 215 : "Typically"?

Line 251: Repetition.

Line 261. This is a red herring type of argument. One of the reasons that sediments are dried or freeze-dried and ground, is to avoid the problem that very small sample sizes and heterogeneity incur in solid phase analytical chemistry, when comparing average samples within a study. If one is interested in very small scale Fe-C heterogeneity, then a wet chemical extraction method is not the right approach.

Line 284: Confounded?

Line 284: Just state that the reagents were no longer in excess (see comments above).

Line 294: I don't believe that the authors mean to say that organic carbon is reduced and released into the solution phase. Interestingly, sulfite incorporation into carbonyl groups may promote organic carbon solubility.

Line 303: If increasing reagent additions make more problems, then what is the point?
Line 399: There is no "standard" method against which to calibrate.
**BGD**

---

## Referee Comment (RC4) · Peter Kraal (Referee) · 2 Dec 2020

Review of bg-2020-399

With interest I have read this manuscript, in which the authors explore the impact of adaptations to an established chemical method (reductive dissolution of Fe(III) by dithionite) to extract OC associated with Fe oxide minerals. This topic is of interest, because the impact of Fe-OC interactions have a bearing on the environmental fate (and possible global budgets) of both Fe and C. However, the manuscript seems to imply that no adaptation isnecessary for the majority of marine sediment samples, and the full analytical impact of some suggested changes was in fact not explored. As such, in my

honest opinion, I do not really see the added value of this "technical note". I do not think that the rather loose suggestion, that increasing dithionite concentration during OCFe extraction might be useful with the caveat that it may have negative consequences that were not investigated, is particularly useful to the geochemical community. Furthermore, I think that the authors overlook some key points that can be taken from the data and do not properly consider the relationships between the findings for the poorly ordered synthetic Fe(III) precipitates (and their properties) and natural samples. Below are detailed comments.

Note: I prepared this review and afterwards read the excellent, extensive reviews by Susann Henkel and Tom Jilbert. I apologize for a limited degree of overlap.

Key points

General language: unnecessarily verbose and at times rather vague, essentials are buried in winding sentences from which the reader has to deduce the actual information. For a technical note, the experimental section is poor.

There seem to be some errors in the use of % and wt.% which are a bit confusing, please carefully check units for Fe and OC concentrations and extraction efficiencies.

Regarding the choice of organic compounds, I understand why the selection was made to have compounds with different amounts of carboxylic functional groups. It would be good if the authors could also explain the choice for these compounds in general, from a point of view of representing natural organic material in marine sediments. As mentione by Tom Jilbert, the discussion on the impact of type of organic compound (L358 ff) is weak and too dependent on other study by Fisher et al.

Coprecipitation is known to affect the structure of Fe(III) precipitates, was any mineralogical characterization of the Fe(III) precipitate performed? The impact of coprecipitation will perhaps be minor for 2-line ferrihydrite, i.e. a poorly ordered ordered Fe(III) precipitate would likely become a bit more disordered. However, there are indications

that such minor changes in structure can result in large changes in reactivity (and thus solubility). And the very high OC:Fe ratio might in fact result in a Fe(III)/OC coprecipitated that is not 2L ferrihydrite but some organic-rich, amorphous hydrous ferric oxide. There is no mention or discussion on the likely characteristics of the Fe(III) precipitates and their relation to natural counterparts anywhere in the paper.

Also, I wonder why only 2-line ferrihydrite was used? This poorly ordered Fe(III) precipitate often transforms very rapidly into more crystalline Fe(III) precipitates such as lepidocrocite or goethite (for which limited solubility in CBD would be much more relevant?). Overall, it would be good if the authors could spend some words on their exclusive choice for 2-line ferrihydrite (or the Fe(III) precipitate that would actually form under the experimental conditions): is it representative for the Fe-OC pool in soils and/or sediments?

Would have been interesting to see OC extraction efficiency for all treatments with the Fe-OC/sediment mixtures, not sure whether results from the variable dithionite experiment justifies assuming 1:1 relationship between Fe and associated OC extractability across the experiments conducted in this study. In fact, it would have also been very useful to have OC data for the dithionite/FeOC concentration range in the first experiment shown in Fig. 1: currently, we are not really given much to go by to understand how OC extractability varies as function of extraction conditions: one number for OC and then the assumption that Fe extractability is a perfect proxy for Fe-associated OC extractability for synthetic precipitates and natural samples alike. I find this a bit meagre (particularly for a technical note).

The impact of freeze-drying (section 4.2) is not represented properly. It shows a strong negative effect on the extractability of synthetic fresh Fe(III) precipitate, which is to be expected and reported regularly for poorly ordered Fe(III) precipitates (e.g. Kraal et al., Chemosphere, 2019). But is showed to have no effect on Fe extractability from the Arctic sample! However, in the discussion this important observations is ignored and there is a winding, unfocused and partly incorrect (see also Susann Henkel's review)

discussion on the practices and challenges of freeze-drying. I have some reservations about the discussion on the impact of dithionite concentration on extraction efficiency in relation to increasing concentrations of Fe and OC. Firstly (L273-281), the authors present data from sediment/synthetic Fe-OC mixtures with Fe and OC concentrations that are incredibly high and as far as I know definitely not, as the authors claim, common (20-30 wt% OC and > 10 wt% Fe are not representative of normal marine sediments, shallow or deep). Secondly, the authors focus a lot of attention on the relationship between dithionite concentration and OC content (L272-281), while I would expect that it is the ratio between dithionite and Fe (which is discussed later in the discussion, L282-291). I would argue that the bulk OC content of a sediment has relatively little to do with extractability of OC-FeHR, and I think this discussion on the role of dithionite concentration should better reflect the processes by which OC-FeHR is liberated. It actually seems like the "standard" (Lalonde) procedure works well for most sediments. The only adaptation really put forward in the current technical note is increasing the dithionite concentration, with two huge caveats: it seems only necessary for extremely Fe-rich samples (an observation not properly represented, as mentioned above) and the authors suggest that jacking up the dithionite concentration may have negative consequences on the performance of the extraction method, without actually exploring those potential negative impacts. I then wonder what to take away from this technical note?

Detailed

L29. CO2

L87. HCl

L100-115. Would be nice to report the Fe/OC ratios during coprecipitation and the rationale behind choosing the concentrations. Also, how long was the precipitation allowed to occur? L111-112. a factor

L113. binding association? Seems repetitive. It's also not type of binding, as all organic

compounds have carboxyl groups. So... do you mean denticity, i.e. number of groups with which ligand binds to atom?

L112-115. Please rephrase this, it is unclear (what is "weak" binding in this context?) and the link to slurry and dry sample is not explained.

L117. "to explore whether mechanistic trends persisted", please explain what this means here. In other words, use less fancy words and provide more concrete information about what you want to test by varying Fe-OC content

L119. "using the same original carrier sample and similarly treated to liberate OC and inorganic carbon". You mixed marine sediment with your Fe-OC precipitate (I think that 'carrier' terminology is both unnecessary and incorrect as a carrier is strictly something else than a matrix). What does "the same original carrier sample" even mean, you used the same sample as used by others before? And can you please better explain what treatments were applied to the sample, rather than "similarly treated to liberate OC and inorganic carbon", which could mean treatment for IC and OC was similar, or it was similar to treatments by Fisher et al. You are not writing a novel, the reader expects a clear and concise (and correct) description of materials and methods applied to them. So, as matrix (I am not adopting the 'carrier' terminology) you used sediment from which CaCO3 and OC were removed, right? L121. Why "spiking" here and "mixing" before?

L136-137. Was the artificial seawater deoxygenated? Do you expect any Fe precipitation issues when introducing oxygenated seawater into a sample with Fe(II)? Also, what was the composition/recipe of the ASW? Wash steps commonly involve simple strong salt solutions (e.g. 1 M MgCl2), preparing ASW for this seems like an extra step for which I would like to know the rationale.

L140. "Testing the impact of different extraction conditions"

L162. What is freeze thawing? Just thawing?

L174-184. Only after reading section 3.1 does it become clear in which samples OC was measured. Please rewrite/rephrase this section, the whole thing is difficult to understand. In particular, sentences like "Carbon content was not measured for all samples, but was used [used?] during the experiment where Na dithionite concentrations varied (see section 3.1)." and "This was performed to ensure that at the end point samples with incomplete Fe recovery also experienced incomplete OC recovery, as expected due to the <1 OC:Fe molar ratio of our coprecipitates." are awkward because they do not concisely convey the relevant information but rather circle around the relevant information with winding sentences. So, why do you expect correlation between Fe and OC extractability based on the OC:Fe ratio? If it's a coprecipitate, this correlation is expected irrespective of the ratio? To be blunt, the writing style is verbose, rather vague and indirect, which is really taking my attention away from the relevant information (which I have to deduce myself in part). Lastly, in light of the considerable and inconsistent differences between Fe and OC extractabilities, I would actually have been interested in OC extraction efficiency for all the different treatments, I do not really understand why these data were not included, I am also not convinced that the results for one type of experiment (varying dithionite concentration) justify using Fe as only indicator of Fe-OC extraction efficiency across a range of experiments.

L186. So, the first thing I am interested in, is the composition of the Fe-OC precipitates. Is all the OC coprecipitated; did any Fe or OC remain in solution after the precipitation reaction? Also, the 'concentration range' was based on mass% of Fe-OC in the sediment/Fe-OC mixtures, but it is unclear what the concentrations of Fe and OC were in these mixtures (only in L200-201 is this addressed for the first time I think). 20-50 wt% seems biased to really high concentrations of Fe-OC where extraction efficiency is actually low, curious to know why this range was chosen. So, please start the Results section with information on the precipitates that were synthesized, and in the Materials and Methods talk a bit about the choice for the concentration range.

L186. The key is Na dithionite concentrations in solution rather than quantities, I would

say.

L189. Added to the solution, not "our reaction"

L190. Maybe good to immediately emphasize this is a Na dithionite test using the Fe-OC/sediment mixtures in the text, not only in the Fig. 1 caption. Also, mention which OC source (1, 2 or 3 COOH) was in the coprecipitates used for this test. By the way, is the blue axis title correct? "Maximal OC-FeHR extracted (wt%)"? Is the unit wt%, or % of initial OC-FeHR?

L190-191. I object to "this figure can be interpreted as a visualisation of the reduction capacity of Na dithionite relative to initial Fe content". The reduction capacity for me is determined by the cell potential for the redox reaction. The plot shows a decreasing Fe reduction efficiency with decreasing initial dithionite/Fe(III) ratio.

L196. baseline dithionite addition?

L203-205. There is a blue triangle that does not correspond to anything in the legend. Also, using OC-Fe as name for Fe-associated OC is very confusing (to me, OC-Fe would imply OC-associated Fe just like FeS2-Fe means pyrite-iron), please address this and use unambiguous names for the extracted Fe and OC.

L205. "within 10%" is a bit misleading here, because it is the absolute error so in a relative sense if becomes bigger, > 10%, at lower extraction efficiencies, please address this. L206. At which dithionite concentration?

L208. "varying C content" is highly confusing here, you mean three types of OC, right?

L207-212. I am confused again. From the methods (section 2.1), I gather that the Fe-OC coprecipitates used in the dithionite concentration experiment reported in Figure 1 contained hexanedioic acid (' 2 COOH') and was performed with freeze-dried Fe-OC. In Figure 1, it shows a decrease in % Fe (and presumably % OC) extracted with increasing Fe-OC wt% from 90% to 40% between 20wt% to 50 wt% Fe-OC. But in Fig. 2, Fe extraction % decreases from 50% to 30% for 20 wt% to 40 wt% Fe-OC for freeze-

Interactive
comment

dried Fe-OC with 2 COOH as C compound. Am I misunderstanding something, or was there a big difference in Fe extraction % for the two experiments under supposedly similar conditions? As an aside, it would be very useful if the authors could mention at which dithionite concentration the second experiment was conducted (maybe it is tucked away somewhere in the Materials and Methods, please repeat in this section).

L215. With 2 out of 5 treatments not showing this trend, I would consider removing "typically".

L219-220. What is implied here? The sediment contains 20 wt% Fe-OC, that is not a trace amount where analytical limitations would interfere with trends (it is mentioned earlier that this is equivalent to 7 wt% easily reducible Fe). Please explain how the comparatively low (emphasis on comparatively, as the whole range is strongly biased towards really high Fe-OC concentrations) Fe content could obscure a trend.

L224. "were extractable for"? Please rephrase.

L225-227. No doubt this will be dealt with in the discussion (having read the discussion, I now know it is not addressed), but there is a large difference of the impact of freeze-drying on freshly precipitated poorly ordered Fe (for which it is established that freeze-drying decreases reactivity, likely by aggregation) and natural sediment from a depth of 22-23 cm in the sediment in which you will not find any labile, freshly formed Fe unless the sediment forms fresh minerals as an artifact of sample treatment (or am I wrong? There is in fact no information on the chemistry of the studied sediment at all). In this sense, there seems to be a mismatch between the synthetic sample and the natural test material.

L230. "labile" sediment?

L231. Extraction time was extended in 15 min increments

L233. remains constant

L239. So there is the issue of OC hydrolysis at low pH. But to quantify OC in sediments,

they are commonly decalcified to remove inorganic C, using dilute HCl. What do the authors think about this? Does low pH have to be avoided at all costs during CBD extraction, to then submit the sediment to low pH during decalcification?

L239. You targeted physical aspects? Such as... the concentration of dithionite??? This is chemical, surely.

L244-246. As mentioned before, the authors should probably also consider the large difference between the impact of freeze-drying on freshly coprecipitated Fe/OC and the impact of freeze-drying on relatively old and stable sediment. I do not think these are comparable.

L258. Is the unit wt%? The context implies that this is the recovery efficiency (% of added Fe that is subsequently extracted), not the Fe content in the sample. (Should be checked in other instances as well, for instance y-axis of Fig. 1)

L268. "is removable for"?

L271-281. Because Fe and OC co-vary in the treatments, it is hard to judge what factor determines decreased extractability of OC-FeHR: is it the increase in OC or the increase in Fe (up to 24 wt% Fe!)? Merely looking at the trend in OC-Fe extractability in Fig. 1 does not answer that question. I would expect OC-FeHR to be liberated by dithionite because the Fe is reductively dissolved, and so the efficiency of OC liberation scales with efficiency of Fe reduction and not necessarily OC content. This would in fact mean that the bulk OC content of the sample is irrelevant, and the focus is on the wrong parameters in this section. The text in L293-306 supports this, the authors need to rethink their focus and wording here to better capture the chemical processes that occur when treating a sample with dithionite to reductively dissolve FeOx and associated compounds/elements. Again, regarding the units: the concentrations should probably be wt.% rather than % for the reported OC-Fe contents? Also, the authors mention that "many samples exist in the 20-30 [wt.]% range". And "the average value for marine sediment OC-FeHR composition is greater than 20 [wt.]%". These numbers and statements surprise me. Studying coastal and deep-sea sediments myself, I usually find TOC concentrations of 1-10 wt.% in a range of marine environments. The authors seem to claim that extremely organic-rich sediments are in fact very common (even though the phrasing "in many samples" is very different from "in many marine systems"...), and then provide just one example, one value, from the Equatorial Pacific. I would like to have this point discussed a bit more: what kind of depositional environments host these very OC-rich samples, are we talking about modern or ancient (black shales and such, where the issue again appears that testing with labile OC-rich HFO makes little sense)?

L293-306. So, now the authors are saying that the existing method works fine for most samples, that increasing the concentration of CBD could work for some (extreme) samples but may also have negative effects that are not explored... So what are we to do with this technical note?

L307-367. This is an excessively lengthy paragraph on freeze-drying. It is actually strongly detached from the findings and just meanders along various aspects of freeze-drying. The main issue is that the authors do not correctly represent their own results: a decrease in Fe extractability was found in the synthetic samples, but not in the Arctic samples! As I mentioned before, it is to be expected that freeze-drying a much stronger effect on a fresh, poorly ordered Fe(III) precipitate than on a rather old sediment sample. In fact, there are findings that show that freeze-thaw cycles can increase extractability of elements (the authors also touch upon this, and Susann Henkel also hints at some inconsistencies in this section). Note: this is a change in extractability, not content; the text became rather confusing when the authors started to speak about freeze-drying as a treatment that can increase the contents of for instance OC and metals... (L351-353). Overall, this whole section fails to address the key point, i.e. the discrepancy between the results for the synthetic and environmental samples (or, in broader terms, the difference between artefacts in fresh and old Fe minerals, whether they are synthetic or natural), and instead presents long and rather unfocused and at

times confusing literature review on freeze-drying.

Despite my sharp tone (work in progress), I trust this review is fair and constructive.

---

## Author Comment (AC3) · 18 Dec 2020

Response to review by Tom Jilbert.

Original comments in black, responses are in blue.

Fisher et al. present results of experiments into the extractability of OC-FeR (that is to say, sedimentary organic carbon bound to reactive Fe) during treatment with citrate bicarbonate-dithionite solution. CBD extractions are a commonly applied method, either stand-alone or as part of sequential extractions, for investigating elements associated with reducible phases in marine sediments. As the authors state, there is much heterogeneity in the details of applied CBD extraction protocols, even within the narrower context of studies into OC-FeR. This has led to difficulty in comparing results and the possibility that the currently used protocols may be sub-optimal for their stated goals. Therefore there is clearly a need for studies like this one, to eventually improve/harmonize the approaches used in the community.

Overall I had the feeling that the study delivers some interesting results although some of the interpretations are left only lightly justified. This leads to the idea that a more developed set of experiments could have yielded a more useful step forward. For example, the conclusion that complexation by citrate may be limiting the recovery of Fe in the highest-OC-FeR experiment deserves to be tested through a concentration series similar to the dithionite series the authors report. This is especially the case considering the comments of the first reviewer (Henkel) questioning whether citrate limitation is a feasible explanation for the observations in Fig. 1. I do not demand that the authors produce such additional data before publication but it is clear that their conclusion would be more robust if it was available, and therefore the overall impact of the study would be greater. A similar criticism could be leveled at the interpretations of the experiment comparing freeze-dried and wet samples, although I would say this is a complex topic that warrants a separate study.

We thank Dr Jilbert for their review and for seeing the value in our study. In the first review of this manuscript, Dr Henkel did raise the issue of whether citrate limitation could explain the reduced loss of Fe at high wt% Fe contents and in turn directed us to Henkel et al., (2016) which suggested the use of anoxic extractions instead. We happily included this point in our revisions to the manuscript; however, both anoxic extraction and increased citrate address the same issue of Fe precipitation in different ways so there appears to be some consensus on the underlying issue here. It is difficult to think of any other reason why Fe losses would be minimised, as there is nowhere else for the Fe to go besides staying in solution or precipitating.

We did not conduct experiments with increased citrate as the method we test is used for OC extraction and we were therefore very cautious about adding C containing compounds to the reaction. Indeed, we did not have a solution to this problem until Patzner et al., (2020) very recently incorporated a correction for the background DOC levels in the reaction.

We agree there is more work to be done on freeze-dried vs wet samples. Particularly around the effects of freeze-thawing, we unfortunately were unable to conduct these types of analyses as the sediment samples we had access to had all been previously frozen (as is common). However, our study provides a new perspective on this issue; for example, we are not aware of any studies which have quantified the effect of variable water mass on introducing error in determining the dry weight equivalent of sediments, or any studies which have aimed to compare the same synthetic samples under different preparation methods.

We have modified our interpretation of the freeze-drying experiment on the natural samples following the comments raised by Peter Kraal.

Another important point is that it seems that some of the content here may be an overflow from the authors' recent Chem. Geol. paper (cited Fisher et al. 2020), which is not a criticism as such but in some cases I had the feeling the reader is being referred there to explain what is going on in the experiments presented here, which should be avoided as this ms. must also be a stand-alone study, even if it is a Technical Note. I am specifically referring to the interpretation of the results of the experiment in which the degree of carboxylation of OM in synthetic OC-FeR is varied (Fig. 2). The discussion of the mechanisms here (Paragraph from Line 358) is too thin and the reader cannot understand why the degree of carboxylation makes such a difference to Fe extractability without accessing the other paper.

We have expanded our manuscript to provide an explanation for the carboxyl relationship identified in the previous study. We also expanded the methods section in response to a similar comment by a previous reviewer.

*"Trends between Fe extraction and carboxyl content have been previously identified, with an increase in the number of carboxyl groups in an iron bound organic compound resulting in an increased proportion of Fe liberated from the sample (Fisher et al., 2020). This was explained by the greater amorphicity of ferrihydrite when carboxyl rich OC is incorporated into the mineral lattice, i.e., the resultant phase is less crystalline than pure ferrihydrite and therefore easier to reductively dissolve."*

Unfortunately I read Susann's review only after making my own comments on the original ms., then later noticed that she has done a very comprehensive job and found several of the same issues that I wished to highlight. It is good to see that the authors have responded thoroughly to Susann's comments and this will undoubtedly improve the next version. Therefore my list of additional comments is comparatively short.

We are grateful for this response to the changes we have already made to the manuscript and agree it has been significantly improved by the comments we have received.

General (in addition to the above; all Line numbers refer to the original ms):

 - The Introduction can be better worded and arranged: First I suggest to move the para. starting Line 59 to directly above the short para. starting Line 81. This way you first describe the problems with the existing methodologies, then set out how you intend to solve them. Next, check a few key sentences: e.g. Line 26 "Understanding in which environments organic carbon (OC) persists": please clarify that you are referring to preservation of OC in sediments; Line 54 "fully reduce all solid reactive Fe phases and associated carbon"... I could not find this phrase in Lalonde et al. 2012, although it is presented here as a quotation. Please check.

We have moved this paragraph to the suggested location. On line 26 we are discussing the importance of understanding the global carbon cycle, we did not mention sediments here as we acknowledge the fact that OC does not just occur in marine sediments and its preservation is facilitated by many mechanisms (not just reactive iron). In the last line of the abstract, we now also mention that OC-Fe interactions exist in soils. On line 28 we begin the discussion about marine sediments.

The quote from Lalonde et al. (2012) appears on the first page of the PDF of that manuscript. On checking this, the original paper doesn't include the word "fully", so this has

been removed from our quote but this does not change the meaning of the quote as the original article uses "all".

Original (Lalonde): *(The CBD method) "dissolves from the sediment matrix all solid reactive iron phases and the organic carbon associated with these phases (OC-Fe)"*

Our amended Line 54: *These findings contrast the previous understanding of the CBD method performed in an experimental context which states that this extraction will reduce "all solid reactive iron phases and the organic carbon associated with these phases" (Lalonde et al., 2012)*

 - Throughout: the terminology in this field is easily misunderstood. E.g. the first reviewer thought for the whole time that % OC-FeR refers to % of total sediment, when in fact it refers to % of total OC. I also had major difficulties to get this upon first reading. So I suggest to clarify terminology early on, and modify figures and captions to make this easier to follow. E.g. I note that Barber et al. 2017 use more descriptive terminology in tables and figures e.g. "OC bound to Fe (% of total OC)" in their Table 1 and "Fraction of total sediment Fe" in their Fig. 4. Also check that CDB/CBD is used consistently. Both current appear.

We agree that the terminology can be misunderstood in these type of studies. A key difference between our study and that of Barber et al., (2017) is in the total OC content of samples. In our study, the natural sediment matrix has been made OC free, and we have added this aspect to the methods to emphasise how this sediment was prepared in response to the first review. The only OC source in our spiked sediments comes from the coprecipitation of organic acids with ferrihydrite ($OC-Fe_R$), while natural sediments contain various forms of unbound OC. Any addition of $OC-Fe_R$ would represent 100% of the total sample OC pool regardless of how much was added, so this is not a useful metric. When we refer to a sample containing e.g. 20% $OC-Fe_R$ that is not to say 20% of the OC present is Fe bound, but that 20 wt% of that sediment is the $OC-Fe_R$ coprecipitate complex. Following this review and the previous, we have decided to refer to quantities in wt% as much as possible.

We have modified the definition given to $OC-Fe_R$ on line 40 to *"OC bound to reactive iron".*

Additionally we have removed the use of %$OC-Fe_R$ following the previous review which asked us to consider the use of content vs concentration. Therefore, the amount of $OC-Fe_R$ added is now referred to as the $OC-Fe_R$ content in terms of wt%. This has also been changed in the text, e.g. from *"20% $OC-Fe_R$-"* to *"the sample containing 20 wt% $OC-Fe_R$".* Where reference is made to the total OC pool (when referencing other studies) this is explicitly stated; e.g. line 263 has been changed to *"$OC-Fe_R$ has been observed at contents exceeding 40% (of total OC) in terrestrial environments".*

 In table 2, which describes the content of spiked samples, we have added "wt%" to describe the OC-Fe content of the sample. Table 2 also includes the actual composition of the sediment in mg to make it as clear as possible.

The caption to figure 3 has been updated to be more descriptive so they can be understood without reference to table 2.

Figure 3: "The sample used in this experiment was a spiked sediment comprised of 60 wt% carrier sediment and 40 wt% of a 2 COOH $OC-Fe_R$ coprecipitate."

Figures 1 and 2 already use "wt% of $OC-Fe_R$ in sediment" to describe the varying $OC-Fe_R$ contents.

The CBD/CDB interchangeability has been picked up on and corrected.

- Description of Fig. 1 results. The phrase "maximal extraction" is used repeatedly when describing the results, but it is only explained in the Discussion (Line 266). The best place for this description is actually Methods, because you can already state how you intend to use the data to estimate this value. That will make reading the paper a whole lot easier overall.

The definition for maximal extraction has bene moved to the end of section 2.6 in the methods. We have rephrased line 266 to retain the example of maximal extraction and ensure this is not lost.

Specific - Line 67: maybe qualify with 'partial hydrolysis' or similar. For significant digestion of OC from sediments, either very low pH (and use of specific oxidizing acids) or very high pH are required.

Added "partial" to line 67 and subsequent occurrences.

-Line 89: misplaced comma after "suggesting"

Removed in review 1.

-Line 111: "and" should be "a"

Corrected

-Line 120: "varied" in preference to "differed"

Changed.

-Line 192-193: rephrase to "with those samples containing the least Fe showing the greatest proportional/relative extraction of Fe"

Rephrased from the first review to *"with the highest recovery of Fe in samples with low OC-$Fe_R$ contents."*

-Line 203-205: this looks more like part of a caption for Fig 1

Changed the phrasing of this section to refer to Fig.1.

-Line 244-246: Does this mean that the natural sediment samples in these experiments were freeze-thawed before the experiments? If so it will be important to state this in Section 2.2.

This refers to the sediment samples which we performed the extractions on rather than the carrier sediment that was spiked with the coprecipitate. Section 2.5 is the corresponding method section for this and includes the line *"half of which was freeze thawed and half was freeze dried"*

-Line 251: why give the formula? there are many Fe oxides that can be dissolved in dithionite so I suggest just to leave it out

Removed.

Line 259-263: Not clear how XAS can indicate clustering. If you are referring to locally enhanced concentrations ("hotspots"), yes this is a real phenomenon observed by high resolution mapping techniques. Still, I would be surprised if a homogenized sample of 0.25g would have a distinctly different OC-FeR content from the bulk sediment, so the logic of the statement is not clear and the paragraph does not really benefit from it.

Removed this reference, instead replaced with *"spatial and temporal variation in Fe fluxes to the seafloor can result in Fe rich sediments e.g. near hydrothermal vents (Poulton and Canfield, 2006) or in Fe-Mn nodules (Hein et al., 1997)"*

Line 293-294: This is a confusing opening sentence to the paragraph. Rephrase to make more concise.

*Rephrased to: "OC-Fe$_R$ was incompletely recovered for samples treated with a dithionite content equal to that which elicits maximal Fe extraction (Fig. 1)."*

Line 304-305: It is not clear to me how an increase of DOC (citrate) during the extraction would impact on the quantification of OC after the extraction., if this is done on the solid phase. Can the sample not simply be rinsed before the drying and analysis?

A triplicate rinse stage is conducted as citrate is known to be retained in the solid matrix, however, in the Lalonde et al. (2012) study this was not a significant problem (<0.08% of dry weight was citrate). This problem is heightened in synthetic studies such as ours, as the OC/Fe ratio is much lower than for natural samples. Therefore, the ferrihydrite surface has much more space available for sorption of organics such as citrate. Further, citrate has 3 carboxyl groups and we previously showed that 3 COOH containing organics were very difficult to remove from ferrihydrite with the CBD method in Fisher et al., (2020). This would suggest that should citrate become bound, it is unlikely to be removed by the rinse.

To clarify this point we have added a sentence to the section highlighting this is of most concern for synthetic samples. *"This is particularly acute for the reduction of synthetically precipitated samples where low OC/Fe ratios leave more of the mineral surface available for sorption compared to Fe phases associated with natural organic matter."*

Line 326-328: Clauses of the sentence are not well constructed

Restructured into two sentences.

---

## Author Comment (AC4) · 18 Dec 2020

Response to review by Anonymous Reviewer 3.

Original comments in black, response in blue.

The manuscript is an extended footnote to an earlier paper Fisher et al. paper published this year on the effect of organic acids on reactivity and solubility of iron oxides. The manuscript addressed three aspects of a modified citrate-dithionite-bicarbonate (CDB) extraction method used for determining Fe content associated with various iron oxide mineral phases. Variables tested were the dithionite concentration, freeze-drying versus wet extraction, and time of extraction.

We thank the reviewer for their review of our manuscript. On the point that they feel this is an "extended footnote", it may be useful for us to clarify the differences between our previous study and this new submission for any readers unfamiliar with the earlier work. The reviewer is correct to say that it follows an earlier paper we published and which we refer to in this technical note; however, the only similarity is in methodology. In our previous study we established a system by which we could create synthetic OC-Fe compounds with known OC and Fe concentrations, allowing us to determine the true recovery of Fe phases by chemical extractions. We used this system in the previous study to probe chemical interactions related to carboxyl content of organic matter. Here, we utilise this method to investigate the efficiency of the extraction methods. This manuscript contains entirely new data from a series of 4 experiments (time, dithionite concentration, sediment preparation methods for synthetic samples, and extraction of Arctic sediments). We believe this is presents original data and is sufficiently different from our previous work to warrant publication, and we felt the technical note format was most suited to the type of study we conducted.

Starting with the title, the goal of the study and the actual study is mismatched. The paper is about extraction of iron oxides, and the entire discussion revolves around the efficiency of the dithionite method toward the extraction iron oxides at pH, not the organic carbon that is extracted.

While we appreciate that the carrier phase of organic carbon (i.e., ferrihydrite) is dissolved in our experiments, the method we are testing was designed to release organic carbon bound to this carrier phase. We argue that the current title and study goal reflect this fact. In other words, we are not trying to quantitatively dissolve iron oxides, but we are testing possible iterations of a published method designed to quantitatively liberate iron-oxide associated organic carbon.

My reading of the Lalonde et al. 2012 article is that they were employing a more gentle (i.e. circumneutral pH) treatment in order to not overestimate the loss of organic carbon due to hydrolysis. That is perfectly reasonable, as they did not want to overestimate OC losses from the main pool due to hydrolysis. Their goal was not to accurately quantify the Fe content, but to dissolve most of the iron oxide fraction and thereby release iron oxide bound organic matter. Here, the authors imply that this approach is not quantitative. It is not clear at whom or at which samples this study is aimed.

We agree with the reviewer here, the method is aimed at extracting (as quantitatively as possible) the Fe-associated OC pool, not Fe itself - which also addresses the issue raised in the reviewer's previous comment. However, for a system where OC is quantified via the reductive release of Fe, all associated Fe must be reduced to fully liberate the OC pool. We show both in this study (Fig.1) and in Fisher et al. (2020) that full reductive release of OC is not achieved under our experiment conditions. The aim of our Technical Note is to provide colleagues who conduct this type of extraction with an awareness of possible factors (if any) that may affect the efficiency of their extractions. Additionally, given the wide variation in

applied methodologies we wanted to understand whether this was necessarily a problem for reproducibility, i.e. how robust the method is to certain (but admittedly not all the possible) variations in the protocol.

The authors only considered ferrihydrite. So is this applicable only for modern sediments? What about sediments or rocks containing greater concentrations of goethite or hematite?

The benefit of the method we deploy to investigate OC-Fe interactions is that the geochemical system is simplified as far as possible, as such we only consider ferrihydrite here. We realise that this approach also brings limitations, i.e. it does not consider goethite, hematite, magnetite etc. However, research progresses incrementally, and our study does not claim to give all the answers to all the questions in the OC-Fe realm. But regarding the issue of more crystalline iron oxide phases, we show that the tested method cannot fully extract OC and Fe in our ferrihydrite based system, so more crystalline phases are almost certainly going to be even more resistant to the chemical treatment. Indeed, Adhikari and Yang (2015) have already conducted these type of experiments in a hematitie based system (coprecipitated with humic acids) and show incomplete (<50% of Fe liberated) reduction of similar synthetic compounds. As the reviewer suggests, the experiments we conduct are therefore most applicable to modern sediments where ferrihydrite concentrations are high (e.g., near hydrothermal vents, in acid mine drainage deposits), however, this is also where current research suggests the largest fraction of the OC-Fe pool to reside. Recently it was also shown that goethite and hematite likely become less important in terms of OC association due to the desorption of OM during phase transformation (Jelavić, Mitchell and Sand (2020) *Geochem. Persp. Let)*.

The authors claim that no study has thus far has performed a determination of the reductive capacity of the dithionite method (Lines 74-75). The authors, however, also do not clearly provide the criteria for "reductive capacity". It is only implied in their approach of using varying "weight percentages of OC-Fe" that simulates a titration of sorts. And as described below, there are methodological problems with this approach. It is not clear that this a substantial step beyond the Fisher et al., 2020 paper.

In the Fisher et al., (2020) paper a dithionite addition of 0.25g was used throughout, following Lalonde et al., (2012), so no investigation into the reductive capacity was made or can be determined from that study. The purpose here was to take account of the fact that many different iterations of this method have used different masses of dithionite to extract the OC-Fe pool, and a compilation of these is shown in Table 1. None of these studies, to the best of our knowledge, have quantified how much Fe can be extracted based on the varying amount of dithionite they use, but simply recommend a mass of sample which can obviously vary widely in Fe content, especially given the range of samples this method is applied to. We accept the reviewer's comment that we have not provided a definition for reductive capacity and have amended this in the abstract.

"We provide an assessment of the reducing capacity of Na dithionite in the CBD method *(the amount of Fe reduced by a fixed amount of dithionite)"*

The authors pose the question of whether there is a one-size-fits-all solution (line 95), or should the extraction be adjusted to fit the set of samples and exact research question. But they do not really answer this question. For instance, the effects of freeze drying on wet chemical extractions of sediments as extensively discussed in Section 4.2, has long been known (e.g., Rapin et al., 1986, ES&T; and more recently Raiswell et al., 2004, Chem Geol.). This discussion is superfluous. The authors point out the problems faced by every sediment biogeochemist, but offer no new insights of their own, or at least none that have not been

already considered by other studies. They propose no solutions to any of these aspects, except to say that methods employed should be rigorously documented. As the authors point out, analyzing freshly collected wet sediments is not practical for most studies. One has to ask if the efforts to improve the dithionite method are even worth the effort, if freeze-drying is out of the question. The topic of study is an important one in sediment biogeochemistry, especially in how do we deal with examining organic carbon concentrations and speciation in complex matrices. But does this paper bring about a consensus on how to proceed? Unfortunately, I have to answer, no it does not, outside of stating that when using wet chemical sediment extraction methods, that geochemists should carefully consider the type of sediment being analyzed, the amounts of reactants in the methods, sample storage and the exact question being investigated.

The reviewer raises an important point here about the overall value of the study, and we are glad that in principle they agree that research in this area is important. We accept that some of the implications made in the introduction about our aim and the overall conclusions do not match up, particularly about whether the method should be improved/replaced by a new method. We wish to be clear that this is not a 'modified protocol' study; our aim here is not to replace that set out by Lalonde et al., 2012.

This has now been clarified by a restructure of the introduction with a new clear focus *"We aimed to establish whether widespread methodological variations in the CBD extraction protocol for OC-Fe$_R$ extraction influenced the efficiency, and therefore comparability, of this characteristic for marine sediment samples".* We have also modified the discussion to be less "superfluous" by concentrating around the newly outlined focus. We have added clarity in the introduction that this is not designed to be an all-encompassing study, and yes other studies have in isolation considered some of (not all) these factors for their influence in various extraction protocols but rarely do studies such as ours pull these variable parameters together or make it the focus of a discussion. We hope in creating this synthesis this results in a valuable technical paper for colleagues utilising the CBD method, increasing the awareness of pitfalls in reproducibility, particularly given that OC-Fe studies continue to evolve their methodologies.

The new scope of this study is formed on the basis that ,both historically and continually, a vast range of modifications have been made to the CBD method for OC-Fe and these are largely uncalibrated and difficult to compare to one another. Within this scope, rather than suggesting a replacement method, the discussion is now framed around understanding whether the variability in these pre-existing methods is a barrier to comparability and reproducibility between data sets. In other words, does it matter if different extraction times/ dithionite strengths/ sample preparation methods are used? We fully appreciate that we do not present all possible variations of all parameters, but clearly a few of the most critical ones.

We believe by narrowing the focus of the study we can retain its technical importance but avoid the apparent disappointment arising from comments regarding lack of completeness from reviewers who have, fairly, expected that this study is designed to replace or modify an existing method.

In addition, as pointed out in review 2 by Tom Jilbert, some of the topics we pick up on could be papers in their own right, e.g. freeze drying vs wet sediments for OC extraction, and we hope that this can lay the groundwork for future studies.

Methodology

The high iron oxide contents used in these experiments are problematic. First of all, it is not entirely clear what exactly is being measured (see comments on the term "OCFe"). Let's assume that it is %Fe.

Correct, %Fe is measured. Other reviewers raised a similar point and in response, we have added to section 2.6 of the methods to clarify that Fe was measured and how this is calculated.

*"The recovery of Fe following the extractions was calculated by subtracting the control corrected loss of Fe from the initial sample Fe content. Maximal extraction of Fe is defined as the point from which further addition of Na dithionite does not further increase the extraction of Fe."*

For example, the 20% OC-Fe sample contains 0.2x 0.25 g artificial sediment, or 0.05 g Fe.

In the 15 mL of reaction solution, this gives a. 0.06 mole/L Fe. The dithionite solution of 0.25 g $Na_2S_2O_4$ in 15 mL yields 0.093moles $S_2O_4^{2-}$ anion per L. Assuming that upon dispropotionation of the dithionite inwater yields two reducing equivalents, which is probably overestimated due to sidereactions with oxygen and other S decomposition products, we would have <0.18 mol/Ldithionite reducing capacity. Dithionite is barely in excess of the reactive iron fraction,which is a poor starting point for a quantitative extraction. It certainly becomes worseor untenable at 50% or 75% or 100% OC-Fe contents. If the OC-Fe weight% refers to FeOOH, things improve. But only by a third. It is not surprising that the method failsmto reduce these high Fe oxide containing slurries. These high Fe concentrations are actually not realistic (see also my comment below). As the authors point out in Line 259, wt% Fe contents in most sediments are usually less than 10%.

These details can be found in Table 2, where the terminology used has been simplified in response to previous reviewer comments – we apologise for the confusion that has led the reviewer to conduct these calculations however, they are incorrect. The 20% OC-Fe sample contains 0.2 g of artificial sediment and 0.05 g of the OC-Fe$_R$ coprecipitate. That is to say 0.05 g of the resultant complex from OC coprecipitation with ferrihydrite, not 0.05 g of pure Fe. We give the Fe value for this sediment sample as ~7 wt%, equivalent to 0.0175 g in a total mass of 0.25 g, which is less than a third of the amount assumed/calculated by the reviewer and very much in line with common Fe contents in natural sediments. Hopefully this removes some of the concern the reviewer had regarding samples at the 20% OC-Fe$_R$ content.

The reviewer is correct to say that the Fe contents become very high in the 50%+ OC-Fe samples (roughly equivalent to 17wt%+ of Fe), and it is for this reason we did not use these high Fe samples in the dithionite concentration experiments or time experiments. The 50-100% OC-Fe$_R$ samples are only used in the experiment to determine whether sample preparation methods affect Fe extraction from these samples. These were suitable samples to use for this experiment as this is an intracomparison of dried vs wet samples; the actual Fe extraction values are not of critical importance, but mainly the different between dried vs wet sample aliquots. Further, by performing an experiment on samples containing 100% OC-Fe$_R$ (i.e. only the coprecipitate) this serves as a control to ensure that interactions between the coprecipitate and the added sediment are not responsible for the differences observed here.

We are grateful that the reviewer has acknowledged that we explicitly give a comparison of the true Fe content of sediments in the paper. In response to a comment from Dr Jilbert we

have further reinforced this point by removing a reference to Fe clustering increasing sample Fe content. We have made an effort to be as transparent as possible in which samples relate to environmental content; however, the purpose, and benefit, of experimental studies such as this is to be able to create artificial conditions way beyond what is found in natural samples, providing a better understanding of fundamental chemical mechanisms.

Lastly, we would like to highlight that there are numerous instances where Fe contents of a sediment exceed the value for "normal" marine sediments, we have added detail of this in section 4.1 *"While reactive Fe content in bulk natural sediments is usually below 7 wt% Fe (e.g. Canfield, 1989;Raiswell and Canfield, 1998), spatial and temporal variation in Fe fluxes to the seafloor can result in Fe rich sediments e.g. near hydrothermal vents (Poulton and Canfield, 2006) or in Fe-Mn nodules (Hein et al., 1997)."*

The high iron concentrations used in these experiments exacerbate another problem with the experimental set-up. As far as I can tell, the samples were not shaken. I assume that the precipitates sank to the bottom of the reaction vessel (details on the reaction vial type and geometry are missing). Over time the reaction rates will become diffusion limited without shaking. This also renders the results of the time-course experiment somewhat difficult to interpret.

We apologise for this omission. The samples were in fact vortexed following the addition of dithionite (and likewise for the controls), and this important detail has been added to the methods. Shaking throughout the extraction for this extraction protocol is impossible because the samples are incubated in a heated water bath. However, for the time experiments these were manually shaken every 15 minutes, as each sample was removed. We thank the reviewer for noticing this omission in the methods and have corrected accordingly. Reaction vial type was a centrifuge tube (see line 127).

There appear to be no replicates for each dithionite addition. This makes interpretation of the results, especially in Figure 2, difficult.

This is discussed in line 153-157. Essentially, the synthetic OC-Fe$_R$ complexes are limited by production yields with a 5 L reaction volume producing only 5 g of precipitate. This was already scaled up from a usual 2 L reaction volume, but it is not logistically possible to conduct larger scale reactions, and recover the products, without industrial scale laboratory facilities. Therefore, careful choices were made over how this finite amount of sample should be used.

Firstly, behind every data point there are two samples given that a control extraction is conducted alongside each reduction. Secondly, as stated in the text *"Repeats of samples across the content gradient are in lieu of direct replicates for each unique sample condition."* Performing direct replicates from multiple precipitations is problematic as the actual amount of ferrihdyrite precipitated, and the amount of OC coprecipitated, varies between precipitations, i.e. you get a slightly different product each time you precipitate. So while the trends within each series of dilutions from the same precipitate are the same, averaging multiple different samples to represent one data point is likely to introduce an artificial error given we are not experimenting with absolute values. In the context of Figure 2, to maximise the data set, "replicates" occur at different dilutions of OC-Fe$_R$ (y axis) as opposed to individual point replicates. In this figure we see the same trend (= Fe recovery in wet extractions exceeds that of dry extraction) across all samples. In addition, the reviewer can be further assured by the inclusion of three different organic acid based precipitates, and the trend between wet and dry samples persists across three independent precipitations. Finally, while the experiment in Figure 2 is not designed to investigate the role of different organic

acids (due to inclusion of high Fe contents as noted by the reviewer), there is strong agreement in the overall trends between the results shown here and those in our previous study (which does investigate the effect of organic acids). This provides a good check and balance that the system here is performing as expected based on earlier experiments. We give an expanded explanation of the comparison between the two studies in an addition to section 4.2 following a previous reviewer's comment.

Style and Readability

The manuscript would be better served by a radical reduction in length. This is a technical note describing three relatively short comparison experiments that are an extension of the Fisher et al. 2020 paper. For instance, the first two sentences of the manuscript (lines 25 to 27) are obvious to readers of Biogeosciences. There are details (Lines 166- 174) about diluting samples for AAS analysis that do not need to be repeated in such detail. The reader assumes that the authors have a basic understanding concerning the basics of the instrumental analysis. Section 2.7 appears superfluous because there is no where in the Results where organic C is discussed.

Following our response to the reviewer's previous comment, where we suggested to refocus the scope of the manuscript, this will certainly shorten the introduction and discussion. However, both the other reviewers and this reviewer (see next comment) have asked for expansion in places to remove reliance on the Chem Geol paper which we agree is important and have been happy to fulfil (see response to Reviewer 1 and 2). In addition, we think that it is the purpose of a technical note in particular to report all technical aspects of the analysis in detail, to leave no doubt about the procedures.

The reviewer is mistaken to say section 2.7 is unnecessary, as organic C measurements are included in Fig.1 and directly discussed in the results; please see line 203-205 in the original manuscript. It is possible that the reviewer did not notice this was a commentary on the OC results as it is referred to as OC-Fe$_R$, (organic carbon bound to reactive iron). In response to reviewer 1 we have added a more explicit definition of this term at the start of the manuscript which we hope will remove any misunderstandings regarding terminology.

On the other hand, the any clear description of the carrier material was lacking, and I had to read the Fisher et al. 2020 Chem Geology article to understand how this key component had been treated.

This point was similarly made by reviewer 1 and 2, we have expanded the methods section relating to treatment of the carrier material to fully describe this.

I am confused by the use of the term "OC-FeR". What exactly is this? Organic C associated with iron oxides, as per Lalonde et al., 2012? Or is this Fe that is somehow made unreactive by Organic Carbon? Or is this simply the total iron oxide content? Or perhaps, the reactive iron content, whatever that may be? Are they referring to %dry weight Fe? Or are they referring to %dry weight FeOOH, or perhaps Fe2O3?,or perhaps %weight of whatever happens to precipitate including the organic fraction added?

Yes, this links to the point made above, and we apologise for the confusing terminology. We have added a definition *"OC bound to reactive iron (OC-Fe$_R$)"* in the introduction following comments made by previous reviewers. We have also made a substantial effort to reduce terminology where possible, including a much wider use of wt% for dry masses and removal of unnecessary formulae such as $Fe_2O_3$.

Further Comments

Line 183 The clause in the first line of the Results has no meaning. The manuscript is plagued by ill-defined discussion of reactivity. There are sentences such as "associated OC has a large influence on Fe reactivity." Towards what?

References to reactivity in this manuscript refer to the reactivity of Fe when reduced by dithionite, and this has been clarified for all in text mentions of reactivity. We have removed the first part of line 183 to make this sentence more concise.

*"The contemporary CBD method of Lalonde et al. (2012) requires a 0.25 g addition per relative to 0.25 g of dried sediment sample. Here, the mass of dithionite added to our reaction was adjusted (0.125 g, 0.375 g, 0.500 g, 0.625 g) while the sediment mass remained at 0.25 g"*

Line 65: This sentence is misleading. Many permutations, improvements and evaluations to and of the dithionite method have been made, particularly with respect to marine sediments. See for instance Lord 1982 (J Sed Petr.), Kostka and Luther 1994 (GCA) and Raiswell et al. 1994 (Chem Geol). The authors must be referring to the extraction of organic matter.

Correct, this is a reference to OM extraction, supported by the statement on line 66 *"constraints associated with trying to qualitatively extract both OC and Fe."* We have added to the sentence to further reinforce this so it now reads *"Systematic improvement to the CBD method **for the extraction of OC-Fe$_R$** have not yet been attempted."*

Line 46: This is not surprising as hematite has been shown to be only partially dissolved by CDB method (see Kostka and Luther, GCA, 1994).

The method used in the study cited by the reviewer is a pH 4.8 dithionite extraction; we acknowledge the role of pH 4.8 extractions on lines 68-69 in the original study. However, these extractions are fundamentally different to the neutral pH OC-Fe extraction we conduct. In the Lalonde et al. (2012) iteration of the dithionite extraction (which we are working from), the neutral approach is said to reduce *"all solid reactive iron phases and the organic carbon associated with these phases"* (line 53). Therefore, while we agree with the reviewer that the outcome is not necessarily surprising, it is notable that other studies have come to other conclusions when applying this specific extraction to hematite samples.

Line 48 What do the authors mean be Fe reactivity here? Is this the goal of the study? Or the extractability of organic compounds.

See response to the first comment replied to in the "Further comments" section.

Line 54 "developed" knowledge?

Removed developed

Line 120 This is not a concentration gradient. First of all, the authors are referring to contents, not concentrations (there seems to be confusion about the terms concentration and content throughout this manuscript). Secondly, a gradient implies a change in concentration over some property (e.g. depth, distances, density, etc..)

We have checked and modified our use of content vs concentration throughout the manuscript in response to a similar point by reviewer 1. Gradient has been substituted for series.

Line 122 Confusing. Was the carrier material freeze-dried before or after mixing (or not all)?

*Before mixing, clarified in the expanded methods section on carrier treatment. "This sample was freeze-dried then ashed (650 °C, 12 hrs) to remove OC, and fumigated with HCl vapour to remove inorganic carbon."*

Line 140 This is repetition of the Lines 80 and following.

Lines 80 onwards discuss how other studies have varied methodological parameters; Lines 140 onwards discuss the specific changes we make in our method,

Line 170 Samples that were highly concentrated were diluted only 10 times while the more dilute samples were diluted 20 times?

To clarify, the samples diluted 10 times (or not at all) were the supernatants from the seawater rinses which follow collection of the extraction supernatant. The highly concentrated samples from the extraction were strongly diluted as appropriate to fit the AAS calibration window. The seawater rinses have very low concentrations of Fe and hence do not need much (if any) dilution. We have clarified this in text.

*"Dilutions of initial samples, in addition to the extraction supernatants, were conducted using MilliQ water, to produce a subsample within the calibrated concentration range of the AAS (1–10 ppm Fe)."*

Line 180 It's not clear that the authors differentiate here between a standard that is used for calibration and a secondary standard used as control.

For this purpose, they are the same thing, i.e., there is no control of accuracy with an international reference material. The machine is calibrated by sampling different masses of the manufacturer's standard. Then, during the sample run, this standard is resampled to check for drift. This is an important check in our method due to the use of decarbonated samples where drift can be a sign that the filter requires changing due to chloride accumulation. For analysing Fe concentrations by AAS at such high levels in experimental samples, the use of international reference materials is not common practise.

Line 190: This does not show the reductive capacity of the dithionite. If, for instance, dithionite is in excess, then 100% Fe extraction cannot show the reductive capacity of the dithionite.

The reviewer is correct here, but since we never reach 100% Fe extraction, this is not an issue in this application.

Line 224 Freeze-thawed samples? This experiment is not mentioned in the methods. Furthermore, this sentence (which is also discussion/interpretation) does not make sense. What "previous one". The sentence refers to Figure 1. There are no freezethaw or freeze-dry samples in Figure 1.

Please see Line 160-2 of the methods *"Arctic Ocean seafloor sediment was collected (…), of which half was freeze thawed and half was freeze dried."*

The sentence refers to the experiment between slurry and freeze-dried samples (shown in Fig. 2). We have added a clarification.

*"This experiment only differed from that shown in Fig.2 by comparing freeze dried vs freeze thawed (not slurry) samples".*

Figure 1: What do the fits represent and how is the fitting done? It looks to me like if you added more dithionite, eventually the %Fe recovery would start to decrease at some point.

Also, the blue symbols representing maximal Fe extraction do not match the corresponding curves for the black symbols.

The trend shown is a second order polynomial nonlinear fit. We address the issue of decreased Fe reduction as being a feature of Fe precipitation, an area of the manuscript we have expanded on in response to the review by Dr Henkel. The blue symbols represent maximal OC extraction rather than Fe, the OC/Fe ratio of <1 explains why they do not line up. A revised version of Figure 1 can be found in the response to Dr Henkel.

Figure 2 is difficult to interpret. Firstly, the dependent variable is plotted on the x-axis, which is confusing for the bar chart depiction. Secondly, outside of the observation that freeze-dried sediments tend exhibit lower extractability than the fresh samples at high Fe contents, it is difficult to ascertain any kind of trend. Given the lack of replicates for each sample, and the large degree of variability in extractabilities, I find it difficult to be able to say anything concrete about these results.

We refer the reviewer to the improved Figure 2 created following constructive input from Dr Henkel which can be found in our reply. The issue of replicates has been replied to in detail earlier in this response.

Line 215 : "Typically"?

Removed

Line 251: Repetition.

Removed

Line 261. This is a red herring type of argument. One of the reasons that sediments are dried or freeze-dried and ground, is to avoid the problem that very small sample sizes and heterogeneity incur in solid phase analytical chemistry, when comparing average samples within a study. If one is interested in very small scale Fe-C heterogeneity, then a wet chemical extraction method is not the right approach.

This section was removed in an earlier edit in response to comments from Dr Jilbert.

Line 284: Confounded?

Substituted with "reinforced"

Line 284: Just state that the reagents were no longer in excess (see comments above).

Unclear what this comment refers to.

Line 294: I don't believe that the authors mean to say that organic carbon is reduced and released into the solution phase. Interestingly, sulfite incorporation into carbonyl groups may promote organic carbon solubility.

Replaced *"released from the sediment"* with *"released from the coprecipitate complex".*

Line 303: If increasing reagent additions make more problems, then what is the point?

We have addressed this point in detail thanks to a suggestion from Dr Henkel about the potential of incorporating anoxic methods as an alternative to deal with Fe precipitation.

Line 399: There is no "standard" method against which to calibrate

This line no longer stands part of the revised manuscript following earlier edits.

---

## Author Comment (AC5) · 18 Dec 2020

Response to review by Peter Kraal (Reviewer 4)

Original comments in black, responses in blue.

With interest I have read this manuscript, in which the authors explore the impact of adaptations to an established chemical method (reductive dissolution of Fe(III) by dithionite) to extract OC associated with Fe oxide minerals. This topic is of interest, because the impact of Fe-OC interactions have a bearing on the environmental fate (and possible global budgets) of both Fe and C. However, the manuscript seems to imply that no adaptation is necessary for the majority of marine sediment samples, and the full analytical impact of some suggested changes was in fact not explored. As such, in my honest opinion, I do not really see the added value of this "technical note". I do not think that the rather loose suggestion, that increasing dithionite concentration during OCFe extraction might be useful with the caveat that it may have negative consequences that were not investigated, is particularly useful to the geochemical community. Furthermore, I think that the authors overlook some key points that can be taken from the data and do not properly consider the relationships between the findings for the poorly ordered synthetic Fe(III) precipitates (and their properties) and natural samples.

We thank Dr Kraal for their review. In response to the overall value of the study, reviewer 3 raised a similar point which we responded to in detail in that review. Essentially, we have reframed the introduction and discussion around addressing the issue of comparability between extractions conducted by differing methods which we felt provided a better focus. We also think that within this context the synthesis we provide is more useful to colleagues utilising this methodology. The citrate issue is somewhat of a red herring and we detail why any potential implications of increasing citrate are likely to be unproblematic in response to Tom Jilbert, we also include an anoxic alternative to citrates increases thanks to suggestions by Susann Henkel. We are grateful for the comments around how we can better discuss the data we present; this was particularly useful for the discussion on freeze drying of the Arctic sediments which we have adapted substantially.

Below are detailed comments. Note: I prepared this review and afterwards read the excellent, extensive reviews by Susann Henkel and Tom Jilbert. I apologize for a limited degree of overlap.

Key points General language: unnecessarily verbose and at times rather vague, essentials are buried in winding sentences from which the reader has to deduce the actual information. For a technical note, the experimental section is poor. There seem to be some errors in the use of % and wt.% which are a bit confusing, please carefully check units for Fe and OC concentrations and extraction efficiencies.

This has been corrected following previous reviews; the use of "content" and "concentration" has been modified, and wt% has been used much more widely to avoid confusion.

Regarding the choice of organic compounds, I understand why the selection was made to have compounds with different amounts of carboxylic functional groups. It would be good if the authors could also explain the choice for these compounds in general, from a point of view of representing natural organic material in marine sediments. As mentioned by Tom Jilbert, the discussion on the impact of type of organic compound (L358 ff) is weak and too dependent on other study by Fisher et al.

The compounds weren't chosen to represent marine OM, but to be the simplest types of compounds capable of bonding with reactive iron. We have expanded on the selection of

these compounds in the methods (line 110 in the original ms) through explicit mention of the bonding mechanism.

*"These acids differ in their carboxyl group content (pentanoic- 1 COOH, hexanedioic- 2 COOH, 1,2,4-Butanetricarboxylic- 3 COOH), a factor thought to influence their binding to $Fe_R$ **via bonding between carboxyl groups and mineral hydroxyls** (Karlsson and Persson, 2010, 2012;Mikutta, 2011)."*

Additionally, as Tom Jilbert also commented on this matter, for the discussion in line 358, we added the extra section (below) to expand on the role of OC compounds here and remove the reliance on the previous publication.

*"Trends between Fe extraction and carboxyl content have been previously identified, with an increase in the number of carboxyl groups in an iron bound organic compound resulting in an increased proportion of Fe liberated from the sample (Fisher et al., 2020). This was explained by the greater amorphicity of ferrihydrite when carboxyl rich OC is incorporated into the mineral lattice, the resultant phase is less crystalline than pure ferrihydrite and therefore easier to reduce."*

Coprecipitation is known to affect the structure of Fe(III) precipitates, was any mineralogical characterization of the Fe(III) precipitate performed? The impact of coprecipitation will perhaps be minor for 2-line ferrihydrite, i.e. a poorly ordered ordered Fe(III) precipitate would likely become a bit more disordered. However, there are that such minor changes in structure can result in large changes in reactivity (and thus solubility). And the very high OC:Fe ratio might in fact result in a Fe(III)/OC coprecipitated that is not 2L ferrihydrite but some organic-rich, amorphous hydrous ferric oxide. There is no mention or discussion on the likely characteristics of the Fe(III) precipitates and their relation to natural counterparts anywhere in the paper. Also, I wonder why only 2-line ferrihydrite was used? This poorly ordered Fe(III) precipitate often transforms very rapidly into more crystalline Fe(III) precipitates such as lepidocrocite or goethite (for which limited solubility in CBD would be much more relevant?). Overall, it would be good if the authors could spend some words on their exclusive choice for 2-line ferrihydrite (or the Fe(III) precipitate that would actually form under the experimental conditions): is it representative for the Fe-OC pool in soils and/or sediments?

Mineralogical characterisation was performed on these same samples in the previous study; we include the XRD analysis shown in sup fig. 1 of that paper below. The reviewer is correct that the mineral becomes more disordered and this is now mentioned in text (see response to previous comment). This XRD analysis shows that the resultant sample remains as ferrihydrite but the peaks soften with the decreasing crystallinity of the structure. Following the review by Susann Henkel, where the suggestion that the ferrihydrite may be transformed during freeze-drying was raised, we added a reference to this XRD data in text (end of line 334 in the original ms).

*"An alternate hypothesis to describe the reduced Fe recovery for dried sediment, that transformation of ferrihdyrite occurred during freeze drying, was ruled out by x-ray diffraction (XRD) characterisation of the freeze dried phase as 2-line ferrihydrite (Fisher et al., 2020)."*

We have added a section to expand on the use of 2-line over 6-line ferrihydrite. For the production of ferrihydrite we worked from the method of Schwertmann and Cornell (2000) *Iron Oxides in the Laboratory.* They describe the difference in production of 2-line vs 6-line ferrihydrite as

"Rapid, forced hydrolysis of $Fe^{III}$ salt solutions under very acidic conditions at elevated temperatures (e.g. 80 "C) for a short period of time leads to 6-line ferrihydrite whereas rapid hydrolysis at RT and close to neutral pH produces 2-line ferrihydrite (Chukhrov et al., 1973; Schwertmann and Fischer, 1973)". Additionally, 2-line ferrihydrite acts as a precursor for hematite and goethite, mirroring mineralogical transformations in sediment.

Regarding the choice of ferrihydrite, we know that in most natural environments the association of organic matter with Fe occurs during $Fe^{2+}$ oxidation (e.g. Sodano et al., 2017, Riedel et al., 2013), and often (i.e., in many soils and most marine sediments) under low temperature, near netural pH conditions. These conditions are therefore much closer to the synthesis of 2-line than 6-line ferrihydrite, hence the choice. Additionally, 2-line ferrihydrite is used exclusively for OM coprecipitation as the high temperature, low pH conditions needed to precipitate 6-line ferrihydrite would make sorption of OM impossible due to hydrolysis. It would, without doubt, be interesting to observe what happens to the ferrihydrite-OC coprecipitate during further aging/crystallisation, but that is beyond the scope of this Technical Note.

In section 2.1 of the Methods, we added: *"2-line ferrihydrite was chosen as it is readily precipitated in the low temperature, circumneutral pH, aqueous conditions of marine surface sediments and has an established ability to be experimentally coprecipiated with organic matter (e.g. Eusterhues et al., 2008;Eusterhues et al., 2011;Eusterhues et al., 2014)."*

[Figure]

**X-Ray Diffraction Analysis**

Figure S.1- Stacked XRD of coprecipitates with increasing carboxyl rich organic content.

Would have been interesting to see OC extraction efficiency for all treatments with the Fe-OC/sediment mixtures, not sure whether results from the variable dithionite experiment justifies assuming 1:1 relationship between Fe and associated OC extractability across the experiments conducted in this study. In fact, it would have also been very useful to have OC data for the dithionite/FeOC concentration range in the first experiment shown in Fig. 1: currently, we are not really given much to go by to understand how OC extractability varies as function of extraction conditions: one number for OC and then the assumption that Fe extractability is a perfect proxy for Fe-associated OC extractability for synthetic precipitates and natural samples alike. I find this a bit meagre (particularly for a technical note).

Unfortunately, it was not possible to produce OC data for all the points. The take away from Fig.1 is that even when Fe extraction is stretched to its maximum (by alteration of dithionite content), OC recovery is still incomplete. Therefore, at any point below this dithionite concentration, OC is certainly also incompletely extracted. While we do not state that Fe is a perfect proxy for OC loss, a strong similarity between the two values is to be expected given the 0.7-1:1 molar ratio between OC and Fe.

The impact of freeze-drying (section 4.2) is not represented properly. It shows a strong negative effect on the extractability of synthetic fresh Fe(III) precipitate, which is to be expected and reported regularly for poorly ordered Fe(III) precipitates (e.g. Kraal et al., Chemosphere, 2019). But is showed to have no effect on Fe extractability from the Arctic sample! However, in the discussion this important observations is ignored and there is a winding, unfocused and partly incorrect (see also Susann Henkel's review) discussion on the practices and challenges of freeze-drying.

We added to this section following Susann Henkel's review to include the suggestion of anoxic conditions. The inclusion of the Arctic sample is not meant for direct comparison with the artificial precipitate, but shows the effects of different dithionite additions on a natural sample representing what is often available to these types of studies. . It is also worth noting that our synthetic system is not meant as a direct analogy for real marine sediments, but plenty of studies perform mineral synthesis and utilise these samples for extractions, so even in synthetic samples this is an important difference. We did expand on the practicalities of marine sediment treatment in new sections added to 4.2

e.g.

*"Wet thawed samples have been used more widely in the sequential extraction of Fe (e.g. Laufer et al., 2020;Riedinger et al., 2017;Wehrmann et al., 2014); additionally the Arctic sediment sample used in our analysis was similarly subject to freeze-thawing. However, our freeze-thawed sample showed no difference in its recovery for Fe compared to the dried variant of this sample indicating a potential interference from the thawing stage."*

*"However, the use of wet sediments is likely to be inappropriate for some analyses or sample sites either due to practical considerations such as the difficulty in transporting heavy wet sediments or when there is a need to preserve the sediment profile, for example protecting anoxic sediments from oxic redox transformations."*

I have some reservations about the discussion on the impact of dithionite concentration on extraction efficiency in relation to increasing concentrations of Fe and OC. Firstly (L273-281), the authors present data from sediment/synthetic Fe-OC mixtures with Fe and OC concentrations that are incredibly high and as far as I know definitely not, as the authors claim, common (20-30 wt% OC and > 10 wt% Fe are not representative of normal marine sediments, shallow or deep).

We do not claim that we are trying to replicate normal marine sediment conditions here as this is a mechanistic study, see response to reviewer 3 for our discussion on high Fe contents.

Secondly, the authors focus a lot of attention on the relationship between dithionite concentration and OC content (L272-281), while I would expect that it is the ratio between dithionite and Fe (which is discussed later in the discussion, L282-291). I would argue that the bulk OC content of a sediment has relatively little to do with extractability of OC-FeHR, and I think this discussion on the role of dithionite concentration should better reflect the processes by which OC-FeHR is liberated. It actually seems like the "standard" (Lalonde) procedure works well for most sediments.

We agree that the ratio between dithionite and Fe is the most important fact; we do not discuss bulk sedimentary Fe since our samples contain only OC-Fe$_R$. The Lalonde method does likely work well for most sediments so we removed the direct criticism here, instead saying *"This finding demonstrates for these OC-Fe$_R$ rich sediments an increase in the dithionite content would aid a more accurate content determination."* However, many alternate methods have differing dithionite contents, and we show dithionite content can be an important factor Fe extraction.

The only adaptation really put forward in the current technical note is increasing the dithionite concentration, with two huge caveats: it seems only necessary for extremely Fe-rich samples (an observation not properly represented, as mentioned above) and the authors suggest that jacking up the dithionite concentration may have negative consequences on the performance of the extraction method, without actually exploring those potential negative impacts. I then wonder what to take away from this technical note?

As expanded on in the response to prior comments, the discussion around the increase of dithionite has been expanded to include a non-damaging anoxic method. The overall take away from this paper should be that methods which have already used differing dithionite conditions in the past are unlikely to be comparable in their results. That increasing extraction time, as done by recent methods, is largely pointless and that the important differences noted in drying of sediment is potentially important and requires further consideration. We hope this will become more apparent by the reframing of the discussion.

Detailed

L29. CO2

Changed

L87. HCl

Changed

 L100-115. Would be nice to report the Fe/OC ratios during coprecipitation and the rationale behind choosing the concentrations. Also, how long was the precipitation allowed to occur?

Added: *"The resultant slurry was rinsed 5 times in 5 L of DI water **over 4 days**"*

Added: *"The mass of organic acids used was determined through batch coprecipitations with varying organic contents, the values used here represent the saturation point where a greater addition of organic does not result in increased association of the organic with ferrihydrite."*

Added: *"The coprecipitations produced three complexes with an increasing number of carboxyl groups resulting in an increased molar C/Fe ratio of (0.04, 0.25 and 0.70:1)."*

L111-112. a factor

Changed

L113. binding association? Seems repetitive. It's also not type of binding, as all organic compounds have carboxyl groups. So. . . do you mean denticity, i.e. number of groups with which ligand binds to atom?

Changed to "strength of OC-Fe$_R$ association".

L112-115. Please rephrase this, it is unclear (what is "weak" binding in this context?) and the link to slurry and dry sample is not explained.

Changed to: *"For example, if it were shown that the OC-Fe$_R$ containing was removable for a greater proportion of its Fe content in a slurried sate than as a freeze dried solid, is this related to the strength of the OC-Fe$_R$ association or the physical difference (e.g. aggregation, surface area) of the dried vs slurried samples."*

L117. "to explore whether mechanistic trends persisted", please explain what this means here. In other words, use less fancy words and provide more concrete information about what you want to test by varying Fe-OC content

Changed to: *"The OC-Fe$_R$ content of synthetic samples was varied to explore whether mechanistic any trends in Fe extractability for each of our experiments persisted at environmentally relevant OC-Fe$_R$ contents."*

The actual parameters we changed are detailed in their own methods section (2.4)

L119. "using the same original carrier sample and similarly treated to liberate OC and inorganic carbon". You mixed marine sediment with your Fe-OC precipitate (I think that 'carrier' terminology is both unnecessary and incorrect as a carrier is strictly something else than a matrix). What does "the same original carrier sample" even mean, you used the same sample as used by others before? And can you please better explain what treatments were applied to the sample, rather than "similarly treated to liberate OC and inorganic carbon", which could mean treatment for IC and OC was similar, or it was similar to treatments by Fisher et al. You are not writing a novel, the reader expects a clear and concise (and correct) description of materials and methods applied to them. So, as matrix (I am not adopting the 'carrier' terminology) you used sediment from which CaCO3 and OC were removed, right?

Deleted "carrier"

We expanded this section following similar comments from an earlier reviewer to read:

*"The OC-Fe$_R$ content of synthetic samples was varied to explore whether any trends in Fe extractability for each of our experiments persisted at environmentally relevant OC-Fe$_R$ contents. To achieve this we spiked the precipitate into a marine sediment sample from the Barents Sea (water depth 141 m; sediment core depth, 33.5 cm; station B6, E40; cruise JR16006). This sediment was freeze-dried then ashed (650 °C, 12 hrs) to remove OC and fumigated with HCl vapour to remove inorganic carbon."*

L121. Why "spiking" here and "mixing" before?

Mixing removed, see above.

L136-137. Was the artificial seawater deoxygenated? Do you expect any Fe precipitation issues when introducing oxygenated seawater into a sample with Fe(II)? Also, what was the composition/recipe of the ASW? Wash steps commonly involve simple strong salt solutions (e.g. 1 M MgCl2), preparing ASW for this seems like an extra step for which I would like to know the rationale.

The artificial seawater was made with 35g/L of NaCl. It was not deoxygenated; Fe precipitation was avoided through acidification of the supernatants. The rationale for this is to remove residual citrate and bicarbonate as detailed in the Lalonde et al., (2012) method.

L140. "Testing the impact of different extraction conditions"

Changed subheading.

L162. What is freeze thawing? Just thawing?

Changed to "half was thawed following prior freezing at the point of sampling"

L174-184. Only after reading section 3.1 does it become clear in which samples OC was measured. Please rewrite/rephrase this section, the whole thing is difficult to understand. In particular, sentences like "Carbon content was not measured for all samples, but was used [used?] during the experiment where Na dithionite concentrations varied (see section 3.1)." and "This was performed to ensure that at the end point samples with incomplete Fe recovery also experienced incomplete OC recovery, as expected due to the 10%, at lower extraction efficiencies, please address this.

Rephrased the first part of this paragraph to make it easier to understand. The two quoted sentences have been removed.

*"Carbon content was measured for the synthetic samples used in the experiment investigating the influence of Na dithionite content on Fe extractability (section 3.1). Here, C content was determined for all OC-Fe$_R$ contents (20-50%) both before and after Fe extraction to determine whether OC-Fe$_R$ was completely recovered; given that the Fe this OC is bound to was incompletely reduced across the series."*

 L206. At which dithionite concentration?

Added: "at the maximum dithionite content each sample was subjected to."

L208. "varying C content" is highly confusing here, you mean three types of OC, right?

Changed to "varying by organic acid"

 L207-212. I am confused again. From the methods (section 2.1), I gather that the FeOC coprecipitates used in the dithionite concentration experiment reported in Figure 1 contained hexanedioic acid (' 2 COOH') and was performed with freeze-dried FeOC. In Figure 1, it shows a decrease in % Fe (and presumably % OC) extracted with increasing Fe-OC wt% from 90% to 40% between 20wt% to 50 wt% Fe-OC. But in Fig. 2, Fe extraction % decreases from 50% to 30% for 20 wt% to 40 wt% Fe-OC for freeze dried Fe-OC with 2 COOH as C compound. Am I misunderstanding something, or was there a big difference in Fe extraction % for the two experiments under supposedly similar conditions?

This is broadly correct, they are similar precipitates in that they were made in the same way, however, they were the product of separate precipitations (as one precipitation does not give a large enough yield). This results in variable Fe and C contents in the product of each precipitate, and the variation becomes stark when percentages are implicated (particularly at

lower contents such as 20 wt%) due to the small numbers produced. So a relatively small absolute change can lead to a large % change. This illustrates why independent repeats would not be useful, information should be taken from the trends within each figure rather than a cross comparison between two different products. As noted, the relative trend (decrease in Fe recovery with increase in OC-Fe$_R$ content) is the same, even if the scale differs.

 As an aside, it would be very useful if the authors could mention at which dithionite concentration the second experiment was conducted (maybe it is tucked away somewhere in the Materials and Methods, please repeat in this section).

Yes, everything outside of the varying dithionite concentrations experiment was conducted according to the Lalonde method so the dithionite content returns to 0.25 g. Added *"Recovery of Fe following extraction with 0.25 g of Na dithionite"* to methods section 3.2

L215. With 2 out of 5 treatments not showing this trend, I would consider removing "typically".

Removed in earlier review.

L219-220. What is implied here? The sediment contains 20 wt% Fe-OC, that is not a trace amount where analytical limitations would interfere with trends (it is mentioned earlier that this is equivalent to 7 wt% easily reducible Fe). Please explain how the comparatively low (emphasis on comparatively, as the whole range is strongly biased towards really high Fe-OC concentrations) Fe content could obscure a trend.

It is correct to say these are not trace amounts, the issue with low concentrations in in the quantification of Fe. At 20 wt% OC-Fe, the Fe supernatant requires a 400x dilution to measure Fe in the 1-10 ppm range. Therefore, any small difference in the absolute amount of Fe measured would be multiplied by a large factor which results in the bias shown. This serves as an additional explanation for the difference in % terms referred to in the earlier comment.

L224. "were extractable for"? Please rephrase.

"Extraction of Fe from freeze-dried samples was…"

L225-227. No doubt this will be dealt with in the discussion (having read the discussion, I now know it is not addressed), but there is a large difference of the impact of freezedrying on freshly precipitated poorly ordered Fe (for which it is established that freezedrying decreases reactivity, likely by aggregation) and natural sediment from a depth of 22-23 cm in the sediment in which you will not find any labile, freshly formed Fe unless the sediment forms fresh minerals as an artifact of sample treatment (or am I wrong? There is in fact no information on the chemistry of the studied sediment at all). In this sense, there seems to be a mismatch between the synthetic sample and the natural test material.

We are soon to publish the chemistry of the sediment used here *(Faust et al., Nat. Comms, Accepted)*, hence why we did not include the analyses within this manuscript. A truncated sequential extraction was performed on this core up to 20 cm, at which depth we can confirm a low presence of iron oxyhydroxides (<0.15%), although this still represents 23.44% of the Fe$_R$ pool, similar to the ~22% value we obtained. We have added this reference and a summary of sediment chemistry to the methods.

*"This core has been analysed to a depth of 20 cm (see Faust et al., xxxx), where a reactive Fe content of 0.64 wt% was determined, of which 23.44% could be extracted by chemical treatments targeting labile and poorly crystalline Fe oxyhydroxides."*

We appreciate the overall point about the difference in Fe content of deeper sediment samples compared to surface sediments and have adjusted wording to remove any idea that the sediment sample is comparable to the synthetic minerals.

L230. "labile" sediment?

Removed

 L231. Extraction time was extended in 15 min increments

Changed

L233. remains constant

Changed

 L239. So there is the issue of OC hydrolysis at low pH. But to quantify OC in sediments, they are commonly decalcified to remove inorganic C, using dilute HCl. What do the authors think about this? Does low pH have to be avoided at all costs during CBD extraction, to then submit the sediment to low pH during decalcification?

This is a good point that we agree with, for that reason in this paper we chose to decalcify the samples by HCl fumigation as oppose to HCl rinses (as done in other studies). Since nothing is discarded from the sample when it is treated by fumigation then we reduce any possibility of OC loss.

L239. You targeted physical aspects? Such as. . . the concentration of dithionite??? This is chemical, surely.

Changed

L244-246. As mentioned before, the authors should probably also consider the large difference between the impact of freeze-drying on freshly coprecipitated Fe/OC and the impact of freeze-drying on relatively old and stable sediment. I do not think these are comparable.

See response to the final comment on the same point.

L258. Is the unit wt%? The context implies that this is the recovery efficiency (% of added Fe that is subsequently extracted), not the Fe content in the sample. (Should be checked in other instances as well, for instance y-axis of Fig. 1)

Checked and changed, Y axis of Fig. 1 doesn't include wt%.

L268. "is removable for"?

Changed to liberates

L271-281. Because Fe and OC co-vary in the treatments, it is hard to judge what factor determines decreased extractability of OC-FeHR: is it the increase in OC or the increase in Fe (up to 24 wt% Fe!)? Merely looking at the trend in OC-Fe extractability in Fig. 1 does not answer that question. I would expect OC-FeHR to be liberated by dithionite because the Fe is reductively dissolved, and so the efficiency of OC liberation scales with efficiency of Fe reduction and not necessarily OC content. This would in fact mean that the bulk OC content

of the sample is irrelevant, and the focus is on the wrong parameters in this section. The text in L293-306 supports this, the authors need to rethink their focus and wording here to better capture the chemical processes that occur when treating a sample with dithionite to reductively dissolve FeOx and associated compounds/elements. Again, regarding the units: the concentrations should probably be wt.% rather than % for the reported OC-Fe contents?

We are unclear exactly what this comment is getting at, we suspect this is an issue around phrasing which we hope to have resolved following earlier comments. We fully agree that Fe extraction is the critical parameter here, however in our experiments Fe is intimately linked to OC following coprecipitation. The bulk OC of the samples is the same as the amount of OC bound to $Fe_R$, which the comment agrees "scales with efficiency of Fe reduction".

wt% has been now used throughout to refer to OC-$Fe_R$ contents.

Also, the authors mention that "many samples exist in the 20-30 [wt.]% range". And "the average value for marine sediment OC-FeHR composition is greater than 20 [wt.]%". These numbers and statements surprise me. Studying coastal and deep-sea sediments myself, I usually find TOC concentrations of 1-10 wt.% in a range of marine environments. The authors seem to claim that extremely organic-rich sediments are in fact very common (even though the phrasing "in many samples" is very different from "in many marine systems". . .), and then provide just one example, one value, from the Equatorial Pacific. I would like to have this point discussed a bit more: what kind of depositional environments host these very OC-rich samples, are we talking about modern or ancient (black shales and such, where the issue again appears that testing with labile OC-rich HFO makes little sense)?

The samples we are referring to here as having over 20% OC-$Fe_R$ contents mean that 20% of the TOC pool of a sediment is bound to reactive iron, not that 20 wt% of the sediment is OC-$Fe_R$.

L293-306. So, now the authors are saying that the existing method works fine for most samples, that increasing the concentration of CBD could work for some (extreme) samples but may also have negative effects that are not explored. . . So what are we to do with this technical note?

This narrative can be shifted by changing the focus of the paper slightly. We can instead say that the wide range of dithionite concentrations used in previous methods (Table 1) can have an effect on Fe extractability, and therefore studies with different dithionite masses can produce non-reproducible results, rather than focusing on achieving maximal extraction. The point about citrate as a negative was addressed in response to Susann Henkel and Tom Jilbert, essentially we now include the prospect of anoxic conditions for high Fe samples (which Henkel et al., (2016) shows to work). The issue with citrate contamination is also of minor importance for natural samples, it is only synthetic ferrihydrite which has a much greater free surface (less OM coverage) where citrate contamination can become a big problem. Lalonde et al., (2012) showed that citrate contamination for natural sediments is at most 0.08% of dry sediment weight so increasing it is unlikely to significantly change this. We have toned down the caution given to citrate in the manuscript and included the anoxic alternative.

L307-367. This is an excessively lengthy paragraph on freeze-drying. It is actually strongly detached from the findings and just meanders along various aspects of freezedrying. The main issue is that the authors do not correctly represent their own results: a decrease in Fe extractability was found in the synthetic samples, but not in the Arctic samples! As I mentioned before, it is to be expected that freeze-drying a much stronger effect on a fresh, poorly ordered Fe(III) precipitate than on a rather old sediment sample. In fact, there are

findings that show that freeze-thaw cycles can increase extractability of elements (the authors also touch upon this, and Susann Henkel also hints at some inconsistencies in this section). Note: this is a change in extractability, not content; the text became rather confusing when the authors started to speak about freeze-drying as a treatment that can increase the contents of for instance OC and metals. . . (L351-353). Overall, this whole section fails to address the key point, i.e. the discrepancy between the results for the synthetic and environmental samples (or, in broader terms, the difference between artefacts in fresh and old Fe minerals, whether they are synthetic or natural), and instead presents long and rather unfocused and at times confusing literature review on freeze-drying.

Despite my sharp tone (work in progress), I trust this review is fair and constructive.

We have rewritten this part of the manuscript to remove the extended discussion on freeze-drying. We state that the dominant Fe phases here are crystalline phases unlikely to be extracted by CBD treatment under any circumstance. We also consider the fact that in deeper sediments such as ours the freeze drying effect we observe for freshly precipitated ferrihydrite has occurred through natural aging of the sediment (due to transformation and aggregation with diagenesis). This allows us to conclude that extraction of surface sediments with high ferrihydrite contents is likely to be improved through use of non-freeze dried sediments while for those deeper in the sediment profile these effects are less important.

*"Wet thawed samples have been used more widely in the sequential extraction of Fe (e.g. Laufer et al., 2020;Riedinger et al., 2017;Wehrmann et al., 2014), additionally the Arctic sediment sample used in our analysis was similarly thawed following freezing on collection. However, our thawed sample showed no difference in its recovery for Fe compared to the dried variant of this sample. Natural aging processes within the sediment could describe this lack of freeze-drying effect in deeper sediments (here 22-23cm) with the transformation of labile poorly crystalline ferrihydrite to more stable phases which have increased resistance to CBD extraction. The loss of bound OM through biological degradation or loss in mineral transformation (Jelavić et al., 2020), as well as the decrease in redox potential at depth effect mineral stability beyond what is represented by synthetic precipitation of fresh ferrihydrite complexes. "*

*"We conclude that freeze-drying of fresh or synthetic sediments containing poorly crystalline iron is likely to negatively bias the quantification of these phases by chemical extractions. This may be of particular interest where synthetic iron minerals are used to calibrate extraction protocols. In deeper sediments where the iron content is dominated by more crystalline phases, the effect of sample preparation is no longer significant. We expect that the scale of the freeze-drying effect decreases in line with the decreasing oxyhydroxide content of the uppermost part of the sediment profile, however, it may be useful to determine the threshold at which sample preparation no longer imparts a quantification bias on Fe in future work."*

We have removed the section relating to changes in absolute C/metal contents. We hope the changes we have made in this section satisfy the request for a clearer discussion on the differences between synthetic/environmental samples.

Additional references:

Sodano, M., Lerda, C., Nisticò, R., Martin, M., Magnacca, G., Celi, L. and Said-Pullicino, D., 2017. Dissolved organic carbon retention by coprecipitation during the oxidation of ferrous iron. *Geoderma*, 307, pp.19-29.

Riedel, T., Zak, D., Biester, H. and Dittmar, T., 2013. Iron traps terrestrially derived dissolved organic matter at redox interfaces. *Proceedings of the National Academy of Sciences*, 110(25), pp.10101-10105.

Faust, J.C., Tessin, A., Fisher, B.J., Zindorf, M., Papadaki, S., Hendry, K.R., Doyle, K.A., März, C. Accepted. Millennial scale persistence of organic carbon bound to iron in Arctic marine sediments. Nature Communications.

Laufer, K., Michaud, A. B., Røy, H., and Jørgensen, B. B.: Reactivity of Iron Minerals in the Seabed Toward Microbial Reduction – A Comparison of Different Extraction Techniques, Geomicrobiology Journal, 37, 170-189,

Riedinger, N., Brunner, B., Krastel, S., Arnold, G. L., Wehrmann, L. M., Formolo, M. J., Beck, A., Bates, S. M., Henkel, S., Kasten, S., and Lyons, T. W.: Sulfur Cycling in an Iron Oxide-Dominated, Dynamic Marine Depositional System: The Argentine Continental Margin, Frontiers in Earth Science, 5

Wehrmann, L. M., Formolo, M. J., Owens, J. D., Raiswell, R., Ferdelman, T. G., Riedinger, N., and Lyons, T. W.: Iron and manganese speciation and cycling in glacially influenced high-latitude fjord sediments (West Spitsbergen, Svalbard): Evidence for a benthic recycling-transport mechanism, Geochimica et Cosmochimica Acta, 141, 628-655

Henkel, S., Kasten, S., Poulton, S. W., and Staubwasser, M.: Determination of the stable iron isotopic composition of sequentially leached iron phases in marine sediments, Chemical Geology, 421, 93-102

Jelavic, S., Mitchell, A.C., Sand, K.K. Fate of organic compounds during transformation of ferrihydrite in iron formations. Geochemical Perspectives Letters, 15, 25-29.

---

## Referee Report (RR1)

The Fisher et al. revised manuscript examines the recovery of organic-associated iron oxides using synthetic ferrihydrite with co-precipitated model organics. The justification for the research is the potential of such complexes to enhance the burial of OC by diminishing organic matter mineralization though adsorption or co-precipitation. The title suggests that this paper could lead to a better tool for determining the pool of organic matter associated with Fe, when in fact the paper is about the efficiency of Fe oxide recovery. Much of the interest in these extractions originally stems from the soil extraction of Fe-bound phosphate (i.e. Chang and Jackson 1957), the lacustrine Fe-P work of JDH Williams and colleagues, trace metal associations (i.e. Tessier) and more recent work on Fe forms relative to sulfidization (i.e. Canfield) and marine phosphates (i.e. Ruttenberg). All such extractions, regardless of the Fe association of interest, are necessarily operationally defined. Iron phases other than the target phase can lead to over-estimation, poor recovery of target phases can be an issue (i.e. the topic of this paper), sediment handling (oxidation/drying) can lead to widely varying results, and for some associations, resorption to remaining sediments can decrease efficiencies. Low yields can be remedied by multiple extractions or changes in the strength of the extractant, but with a risk of increasing matrix effects on the final analysis or affecting non-target phases.

Multiple reviewers ahead of this review have provided the authors with a detailed evaluation of the papers merit's, numerous details on literature context, critique of the overall experimental and measurement scheme, and paper organization. In view of this, my critique focus on the revised product and its overall merit.

The use of model phases to assist development of an adequate test of the extraction, is a strong part of this effort – recognizing that the material used is a subset of the forms of iron and organic matter expected in the field. The recovery of poorly crystalline Fe oxides generally is much lower than crystalline forms, with effects on extractability. The lower extractability of OC-FeR at high Fe concentrations, relative to the lower Fe concentrations, is problematic for assessing reactive Fe oxides in Fe-enriched sediment horizons. Repeated extraction could be one solution?

Considering the value of this paper to the literature, its strength is primarily as a "cautionary tale", meaning that the geochemical practitioner examining Fe forms in Fe-rich sediments needs to be aware of the poorer yield from Fe with adsorbed or co-precipitated organic matter. In this regard, the manuscript is useful and in some cases, important. I'm in agreement with the abstract sentence: *While our study is not an all-inclusive method comparison and is not aimed at delivering the "perfect" extraction setup, our findings provide a collected summary of critical factors which influence the efficiency of the CBD extraction for OC-FeR.* The lack of a solution to these disappointing yields makes this paper perhaps less interesting than it might be. However, the observations of the limits of CDB extractions make it a useful and likely valuable contribution to our understanding of coastal sediment organic matter and Fe geochemistry.

---

## Author Response (AR2)

Reviewer comments in black, author response in blue.

We thank both new reviewers for their positive and helpful comments on our manuscript. Since Jeffery Cornwell had no further specific changes, the below responses relate to the comments raised in Bo Thamdrup's review.

The manuscript presents a methodological study of a widely used wet chemical extraction method for determining "reactive iron" and associated organics (and phosphate) in soils and sediments – the dithionite-citrate-bicarbonate (CBD) extraction. It demonstrates how the selectivity of the method might deviate strongly from its intended target by leaving relatively large proportions of the reactive iron pool untouched and how this is affected by associated organics, and it provides guidelines for how to use the extraction to minimize these issues.

Wet chemical extractions for Fe, organics and P are a jungle with the exact protocols varying between labs, and calibrations with standard materials frequently being neglected. In this perspective, the present study has merit, although it also fails to some extent with respect to standards by only testing a homemade iron mineral. While this suffices to demonstrate the issues with the method, and OC-Fe materials might not be available, tests with some iron phases would have been useful.

The experiments have been carried out with care, the results are of good quality, and the manuscript is generally well written.

We thank the reviewer for their comments. As suggested, OC-Fe standard materials do not exist to the best of our knowledge. Synthetic ferrihydrite was used for this study because it is well established as a suitable sample material for both comparison of Fe extractions (e.g., Thompson et al., 2019, Poulton and Canfield, 2005) and OC-Fe interactions, as seen in the Eusterhues studies cited within this manuscript.

My greatest concern with the paper relates to a mixing up of Fe and organic carbon (OC) extractability. It starts with the title, which suggests that the subject is the extractability or Fe-bound OC, but really 90% of the results are about the extractability of Fe. It might be that the authors equate Fe and OC extractability, but then this needs to be discussed and justified.

The CBD extraction we use operates at circumnetural pH to prevent hydrolysis of OC. Well established, efficient methods already exist for the extraction of the target Fe phases, e.g., the pH 4.8 acetic acid buffered dithionite extraction of Poulton and Canfield (2005). Therefore, the extraction we employ is exclusively used for the extraction of iron bound organic carbon, hence the title. Reframing the title to suggest we are investigating dithionite based Fe extraction risks causing confusion since nobody uses this method solely for Fe extraction.

As the reviewer suggests, Fe and OC extractability are strongly linked, since the reductive dissolution of the Fe phases liberates associated OC. We have added a new sentence and reference to clarify the relative release of OC and Fe during the extraction. This extraction in particular results in a correlated release of OC and Fe because it targets the highly reactive Fe

fraction to which OC associates. This is because overtime Fe phases become more crystalline and reactivity and surface sorption capacity decreases (Lalonde et al., 2012).

**On line 42-43** we now emphasise this by stating *"The CBD extraction for OC-Fe$_R$ operates on the principle that reductive dissolution of reactive Fe phases with sodium (Na) dithionite exclusively and quantitatively liberates Fe$_R$-bound OC from the sediment matrix. This extraction can be considered to target OC-Fe$_R$ since the vast majority of iron bound OC is associated with the highly reactive (Fe$_R$) fraction, dissolved by CBD, since more crystalline Fe phases have both reduced surface reactivity (Lalonde et al., 2012) and smaller specific surface area (Jelavić, Mitchell, Sand, 2020) for OC sorption. The reductive release of OC from an OC-Fe$_R$ complex has been shown to occur asynchronously and OC is mobilised to the dissolved phase at a greater rate than Fe (Adhikari et al., 2016)."*

Indeed, Fig. 1 demonstrates that these two parameters do not go hand in hand. In the introduction, l. 39-68, I recommend a clearer separation of studies focusing at Fe, OC, and P extraction with DCB, and I would like to see some discussion of the expected congruency between the extraction of these substances. Fe extraction would seem the natural starting point in this introduction, with OC-Fe and P-Fe extraction being later adaptations. Now it seems the other way around.

We have split the section into two paragraphs to create a greater distinction between historical use of CBD methods (Fe, OC and P) and the recent findings which drive our study. We have additionally changed the format of the start of the first paragraph to begin with Fe before moving on to OC and P as suggested. This paragraph now ends with a comparison between the Fe-P and OC-Fe methods, as we note it is difficult to compare the relative efficiency of these methods because of the variation in the reductive strength. **Line 51**: *"While Ruttenberg (1992) and Thompson et al. (2019) report 90-100% of synthetic ferrihydrite is extracted by the CBD method for Fe$_P$, the dithionite-to-sample ratio in their studies was more than double the ratio used in the OC-Fe$_R$ extraction by Lalonde et al. (2012)"*. We further compare the Fe-P study of Thompson et al., (2019) to an OC-Fe study of Adhikari and Yang (2015) and Fisher et al., (2020) in the new subsequent paragraph.

For instance, I am relatively sure (though I couldn't find my copy and check this) that Mehra and Jackson did NOT use CBD for OC-bound to reactive iron (as claimed in l. 39-40) but for the reactive iron itself.

Corrected:

**Line 40-42**: *"The method was originally applied to the extraction of iron oxides from soils (Deb, 1950; Mehra and Jackson, 1958) before being adapted for OC-Fe$_R$ quantification in marine sediments by Lalonde et al. (2012)"*

Specific comments

24: Please leave it to the reader to decide on the value of your work.

Removed "valuable"

61: Did they really say that CBD will REDUCE the Fe-associated organic C?

Changed to "dissolve".

64: Most of the tests concern Fe extraction, not OC-Fe

Changed to "$Fe_R$ associated with OC"

86: Same as above, most tests concern Fe extraction, not OC-Fe

**Line 90**: Expanded for clarity: "To address the question of how methodological variation affects OC-$Fe_R$ extraction, **due to variable dissolution of the associated $Fe_R$ phase**"

Methods, sect. 2.1 and 2.2: Are the procedures here not same as in Fisher et al. 2020? Or how do the procedures in l. 117-26 and 128-37 differ from the previous paper? If procedures for synthesis differ, how was the structure verified? If they don't, please make this clear and shorten the methods here.

The original version of this manuscript had a shorter methods section. In our first round of review we were asked by multiple reviewers to expand on the methods so readers did not need to consult Fisher et al., (2020) to understand what we did. In addition we were also asked to justify the use of 2-line ferrihydrite which added a new paragraph to section 2.1. The current methods section is as short as we can make it while retaining enough detail to reproduce the experiment. We have decided on balance it is better to retain this expanded methods section in line with the previous reviewers comments.

Results: Please use past tense for your results as customary.

Corrected.

Section 3.1: The first halves of each of the two paragraphs repeat results and should be strongly abbreviated.

This comment refers to repeating results, however we believe it likely refers to repeating methods (since the paragraph starts with an overview of what was done rather than results). We have shortened the overview in the first paragraph down to one sentence.

244: "incomplete Fe extractions" is difficult to understand, please rephrase.

Rephrased: "where the chosen method resulted in <100% extraction of the targeted phases."

248: 40% and 50% of total OC are confusing coming and not comparable to 7 wt% Fe in the previous sentence. These are not useful values here without information about the total OC content. If the systems were OC poor, the OC-Fe content could be low.

We have added two sentences to clarify the purpose of these values and put them into a better context within the paragraph. Line 248: *"Our high Fe content is driven by low C/Fe ratios since only short chain organic compounds are associated with the $Fe_R$ phases, and are therefore designed to be a mechanistic model rather than to simulate the types of compounds which occur naturally. Both the amount of reactive Fe and the amount of OC associated with Fe are highly variable, and the factors which control the OC-$Fe_R$ interaction remain poorly understood."*

**Line 260:** *"This large variability in environments containing OC-Fe$_R$, and the composition of such compounds, highlights the need to examine their extractability across a wide matrix."*

257-279: I suggest merging and condensing these two paragraphs. The discussion of 30-40% OC-Fe first and then 50% OC-Fe seems too repetitive and does not convey the big picture very well.

We have considered this suggestion but do not think it would improve the readability of the manuscript. The trends shown in 30-40% OC-Fe are very similar whereas 50% OC-Fe shows a different trajectory (Fig 1). It is important to consider these separately because there are specific factors which may be responsible for the limitation in OC-Fe and Fe extraction observed at 50%, as was raised in detail by Susann Henkel in the first round of review. We state this difference on **line 274** *"This differs from the previous compositions in reaching a maximum at ~60% Fe, as opposed to the ~90% achieved for 20-40 wt% OC-Fe$_R$."*. We appreciate the desire to present the big picture but feel this is best left to the conclusion while the more technical points of this technical note are presented in the discussion.

Fig. 1: The curves in the figure are not explained and look like parabolas with decreasing values at the highest dithionite additions. I recommend use of hyperbolas or other asymptotic functions instead to reflect the idea that the extraction saturates.

We have changed the curve fit for this figure to reflect a levelling off, reflecting saturation, as oppose to a decrease.

In addition we have changed all figures to remove error bars since these related to a systematic instrumental error yet gave the impression of independent repeats which could not be performed for reasons given in the manuscript. Figure captions have been adapted accordingly and instrument error is now provided in the open access data asset.